# Global glacier volume projections under high-end climate change scenarios

Sarah Shannon[1,2], Robin Smith[3], Andy Wiltshire[4], Tony Payne[2], Matthias Huss[5,6], Richard Betts[1,4], John Caesar[4], Aris Koutroulis[7], Darren Jones[8] and Stephan Harrison[8]

[1]School of Geography, The University of Exeter, The Queen's Drive, Exeter, Devon, EX4 4QJ, UK
[2]Bristol Glaciology Centre, Department of Geographical Science, University Road, University of Bristol, BS8 1SS, UK
[3]NCAS-Climate, Department of Meteorology, University of Reading, Reading, RG6 6BB, UK
[4]Met Office, Fitzroy Road, Exeter, Devon, EX1 3PB, UK
[5]Department of Geosciences, University of Fribourg, Fribourg, Switzerland
[6]Laboratory of Hydraulics, Hydrology and Glaciology, ETH Zurich, Zurich, Switzerland
[7]School of Environmental Engineering, Technical University of Crete, Akrotiri, 73100 Chania, Greece
[8]University of Exeter, Penryn Campus, Treliever Road, Penryn, Cornwall, TR10 9FE, UK

*Correspondence to*: S. R. Shannon (sarah.shannon@bristol.ac.uk)

**Abstract.**

The Paris agreement aims to hold global warming to well below 2ºC and to pursue efforts to limit it to 1.5ºC relative to the pre-industrial period. Recent estimates based on population growth and intended carbon emissions from participant countries, suggest global warming may exceed this ambitious target. Here we present glacier volume projections for the end of this century, under a range of high-end climate change scenarios, defined as exceeding +2 ºC global average warming relative to the preindustrial period. Glacier volume is modelled by developing an elevation-dependent mass balance model for the Joint UK Land Environmental Simulator (JULES). To do this, we modify JULES to include glaciated and un-glaciated surfaces that can exist at multiple heights within a single grid-box. Present day mass balance is calibrated by tuning albedo, wind speed, precipitation and temperature lapse rates to obtain the best agreement with observed mass balance profiles. JULES is forced with an ensemble of six Coupled Model Intercomparison Project Phase 5 (CMIP5) models which were downscaled using the high resolution HadGEM3-A atmosphere only global climate model. The CMIP5 models use the RCP8.5 climate change scenario and were selected on the criteria of passing 2°C global average warming during this century. The ensemble mean volume loss at the end of the century plus/minus one standard deviation is, -64±5% for all glaciers excluding those on the peripheral of the Antarctic ice sheet. The uncertainty in the multi-model mean is rather small and caused by the sensitivity of HadGEM3-A to the boundary conditions supplied by the CMIP5 models. The regions which lose more than 75% of their initial volume by the end of the century are; Alaska, Western Canada and US, Iceland, Scandinavia, Russian Arctic, Central Europe, Caucasus, High Mountain Asia, Low Latitudes, Southern Andes and New Zealand. The ensemble mean ice loss expressed in sea-level equivalent contribution is 215.2 ± 21.3 mm. The largest contributors to sea-level rise are Alaska (44.6 ± 1.1mm), Arctic Canada North and South (34.9 ± 3.0mm), Russian Arctic (33.3 ± 4.8mm), Greenland (20.1±4.4), High Mountain Asia (combined Central Asia, South Asia East and West), (18.0 ± 0.8mm), Southern Andes (14.4 ± 0.1mm) and Svalbard (17.0 ±

4.6mm). Including parametric uncertainty in the calibrated mass balance parameters, gives an upper bound global volume loss of 281.1 mm, sea-level equivalent by the end of the century. Such large ice losses will have inevitable consequences for sea-level rise and for water supply in glacier-fed river systems.

## 1 Introduction

Glaciers act as natural reservoirs by storing water in the winter and releasing it during dry periods. This is particularly vital for seasonal water supply in large river systems in South Asia (Immerzeel, 2013;Lutz et al., 2014;Huss and Hock, 2018) and Central Asia (Sorg et al., 2012) where glacier melting contributes to streamflow and supplies fresh water to millions of people downstream. Glaciers are also major contributors to sea-level rise, despite their mass being much smaller than the Greenland and Antarctic ice sheets (Kaser et al., 2006;Meier et al., 2007;Gardner et al., 2013). Since glaciers are expected to lose mass
into the twenty first century (Radić et al., 2014;Giesen and Oerlemans, 2013;Slangen et al., 2014;Huss and Hock, 2015), there is an urgent need to understand how this will affect seasonal water supply and food security. To study this requires a fully integrated impacts model which includes the linkages and interactions between glacier mass balance, river runoff, irrigation and crop production.

The Joint UK Land Environment Simulator (JULES) (Best et al., 2011) is an appropriate choice for this task because it models
these processes, but is currently missing a representation of glacier ice. JULES is the land surface component of the Met Office Global Climate Model (GCM), which is used for operational weather forecasting and climate modelling studies. JULES was originally developed to model vegetation dynamics, snow and soil hydrological processes within the GCM but now has a crop model to simulate crop yield for wheat, soybean, maize and rice (Osborne, 2014), an irrigation demand scheme to extract water from ground and river stores and two river routing schemes; Total Runoff Integrating Pathways (Oki, 1999)(TRIP) and the
RFM kinematic wave model (Bell et al., 2007). The first objective of this study is to add a glacier ice scheme to JULES to contribute to the larger goal of developing of a fully integrated impacts model.

The second objective is to make projections of glacier volume changes under high-end climate change scenarios, defined as exceeding 2 °C global average warming relative to the preindustrial period (2017). The Paris agreement aims to hold global warming to well below 2ºC and to pursue efforts to limit it to 1.5ºC relative to the pre-industrial period (UNFCCC 2015),
however, there is some evidence that this target may be exceeded. Revised estimates of population growth suggests there is only a 5% chance of staying below 2 °C and that the likely range of temperature increase will be 2.0–4.9 °C (Raftery et al., 2017). A global temperature increase of 2.6–3.1 °C has been estimated based on the intended carbon emissions submitted by the participant countries for 2020 (Rogelj et al., 2016). Therefore, in this study we make end of the century glacier volume projections, using a subset of downscaled Coupled Model Intercomparison Project phase 5 (CMIP5) models which pass 2ºC
and 4ºC global average warming. The CMIP5 models use the Representative Concentration Pathways (RCP) RCP8.5 climate change scenario for high greenhouse gas emissions.

The paper is organised as follows; In Section 2 we describe the glacier ice scheme implemented in JULES and the procedure for initialising the model. Section 3 describes how glacier mass balance is calibrated and validated for the present day. Section 4 presents future glacier volume projections, a comparison with other studies and a discussion on parametric uncertainty in the calibration procedure. Section 5 discusses the results, the model limitations and areas for future development. In Section 6, we summarise our findings with some concluding remarks.

## 2 Model description

JULES (described in detail by Best et al. (2011)) characterises the land surface in terms of sub-grid scale tiles representing natural vegetation, crops, urban, bare soil, lakes and ice. Each grid box is comprised of fractions of these tiles with the total tile fraction summing to 1. The exception to this, is the ice tile which cannot co-exist with other surface types in a grid box. A grid box is either completely covered in ice or not. All tiles can be assigned elevation offsets from the grid box mean which is typically set to zero as default.

To simulate the mass balance of mountain glaciers more accurately we extend the tiling scheme to flexibly model the surface exchange in different elevation classes in each JULES gridbox. We have added two new surface types, glaciated and unglaciated elevated tiles to JULES (version 4.7) to describe the areal extent and variation in height of glaciers in a gridbox (Fig. 1). Each of these new types, at each elevation, has its own bedrock sub-surface with a fixed heat capacity. These sub-surfaces are impervious to water, and have no carbon content, so have no interaction with the complex hydrology or vegetation found in the rest of JULES. Because glaciated and unglaciated elevated tiles have their own separate bedrock sub-surface they are not allowed to share a gridbox with any other tiles. For instance, gridboxes cannot contain partial coverage of elevated glacier ice and vegetated tiles.

JULES is modified to enable tile heights to be specified in meters above sea level, as opposed to the default option, which is to specify heights as offsets from the gridbox mean. This makes it easier to input glacier hypsometry into the model and to compare the output to observations for particular elevation bands. To implement this change, the gridbox mean elevation associated with the forcing data, is read in as an additional ancillary file. Downscaling of the climate data, described in Section 2.1, is calculated using the difference between the elevation band ($z_{band}$) and the gridbox mean elevation ($z_{gbm}$)

$$\Delta z = z_{band} - z_{gbm} \tag{1}$$

For the purposes of this study JULES is set up with a spatial resolution of 0.5-degree and 46 elevation bands ranging from 0 - 9000m in increments of 250m. The horizontal resolution of 0.5-degree is used because it matches the forcing data used to drive the model. The vertical resolution of 250m was used based on computational cost. The vertical and horizontal resolutions of the model can be modified for any setup.

Each elevated glacier tile has a snowpack which can gain mass through accumulation and freezing of water and lose mass through sublimation and melting. JULES has a full energy balance multi-level snowpack scheme which splits the snowpack into layers each having a thickness, temperature, density, grain size (used to determine albedo), and solid ice and liquid water

contents. The initialisation of the snowpack properties and the distribution of the glacier tiles as a function of height is described in section 2.3. Fresh snow accumulates at the surface of the snowpack at a characteristic low density and compacts towards the bottom of the snowpack under the force of gravity. When rain falls on the snowpack, water is percolated through the layers if the pore space is sufficiently large, while any excess water contributes to the surface runoff. Liquid water below the melting

temperature can refreeze. A full energy balance model is used to calculate the energy available for melting. If all the mass in a layer is removed within a model timestep then removal takes place in the layer below. The temperature at each snowpack level is calculated by solving a set of tridiagonal equations for heat transfer with the surface boundary temperature set to the air temperature and the bottom boundary temperature set to the sub-surface temperature.

A snowpack may exist on both glaciated and unglaciated elevated tiles if there is accumulation of snow.

The elevation-dependent mass balance ($SMB_{z,t}$) is calculated as the change in the snowpack mass ($S$) between successive time steps

$$SMB_{z,t} = S_{z,t} - S_{z,t-1} \qquad (2)$$

The scheme assumes that the snowpack can grow or shrink at elevation bands depending on the mass balance, but that tile fraction (derived from the glacier area) is static with time. The ability to grow or shrink the snowpack at elevation levels means

that the model includes a simple elevation feedback mechanism. If the snowpack shrinks to zero at an elevation band, then the terminus of the glacier moves to the next level above. On the other hand, if the snowpack grows at an elevation band it just continues to grow and there is no process to move the ice from higher elevations to lower elevations. Typically, in an elevation feedback, when a glacier grows the surface of the glacier will experience a cooler temperature, however in this case, the snowpack surface experiences the temperature of the elevation band.

## 2.1 Downscaling of climate forcing on elevations

Both glaciated and unglaciated elevated tiles are assigned heights in meters above sea level and the following adjustments are made to the surface climate in gridboxes where glaciers are present.

### 2.2.1 Air temperature and specific humidity

Temperature is adjusted for elevation using a dry and moist adiabatic lapse rate depending on the dew point temperature. First the elevated temperature follows the dry adiabat

$$T_z = T_0 - \gamma_{dry}\Delta z \qquad (3)$$

where $T_0$ is the surface temperature, $\gamma_{dry}$ is the dry adiabatic temperature lapse rate (°Cm$^{-1}$) and $\Delta z$ is the height difference between tile elevation and the gridbox mean elevation associated with the forcing data.

If the $T_z$ is less than the dew point temperature $T_{dew}$ then the temperature adjustment follows the moist adiabat. A moist adiabatic lapse rate is calculated using the surface specific humidity from the forcing data

$$\gamma_{moist} = \frac{\left(\frac{g(1+lc.q_0)}{r.T_v(1-q_0)}\right)}{\left(\frac{C_p+lc.2.q_0.R}{r.T_v2.(1-q_0)}\right)} \quad \text{(4)}$$

$q_0$ is the surface specific humidity, $l_c$ is the latent heat of fusion of water at 0°C ($2.501 \times 10^6$ J kg$^{-1}$), g is the acceleration due to gravity (9.8 ms$^{-2}$), r is the gas constant for dry air (287.05 kg K$^{-1}$) R is the ratio of molecular weights of water and dry air (0.62198) and $T_v$ (K) is the virtual dew point temperature

$$T_v = T_{dew}\left(1 + \left(\frac{1}{R} - 1.0\right).q_0\right) \quad \text{(5)}$$

The height at which the air becomes saturated $z$ is

$$z = \frac{T_0 - T_{dew}}{\gamma_{dry}} \quad \text{(6)}$$

The elevated temperature following the moist adiabat is then

$$T_z = T_{dew} - (\Delta z - z)\gamma_{moist} \quad \text{(7)}$$

Additionally, when $T_z < T_{dew}$, the specific humidity is adjusted for height. The adjustment is made using the elevated air temperature and surface pressure from the forcing data using a lookup table based on Goff-Gratch formula (Landolt-Bornstein, 1987). The adjusted humidity is then used in the surface exchange calculation.

### 2.2.2 Longwave radiation

Downward longwave radiation is adjusted by assuming the atmosphere behaves as a black body using Stefan-Boltzmann's

law. The radiative air temperature at the surface $T_{rad,0}$ is calculated using the downward longwave radiation provided by the forcing data $LW_{\downarrow z0}$

$$T_{rad,0} = \left(\frac{LW_{\downarrow z0}}{\sigma}\right)^{\frac{1}{4}} \quad \text{(8)}$$

Where $\sigma$ is the Stefan-Boltzmann constant ($5.67 \times 10^{-8}$ W m$^{-2}$ K$^{-4}$). The radiative temperature at height is then adjusted

$$T_{rad,z} = T_{rad,0} + T_z - T_0 \quad \text{(9)}$$

Where $T_0$ is the grid box mean temperature from the forcing data and $T_z$ is the elevated air temperature.  This is used to calculate the downward longwave radiation $LW_{\downarrow z}$ at height

$$LW_{\downarrow z} = \sigma T_{rad,z}^4 \quad \text{(10)}$$

An additional correction is made to ensure that the gridbox mean downward longwave radiation is preserved

$$LW_{\downarrow z} = LW_{\downarrow z} - \sum_{z=1}^{n} LW_{\downarrow z}.frac(z)\frac{|LW_{\downarrow z}|}{\sum_{i=1}^{z}|LW_{\downarrow z}|.frac(z)} \quad \text{(11)}$$

where frac is the tile fraction.

### 2.2.3 Precipitation

To account for orographic precipitation, large scale and convective rainfall and snowfall are adjusted for elevation using an annual mean precipitation gradient (%/100m)

$$P_z = P_0 + P_0\,\gamma_{precip}(z - z_0) \tag{12}$$

where $P_0$ is the surface precipitation, $\gamma_{precip}$ is the precipitation gradient and $Z_0$ is the grid box mean elevation. Rainfall is also converted to snowfall when the elevated air temperature $T_z$ is less the melting temperature (0ºC). The adjusted precipitation fields are input into the snowpack scheme and the hydrology subroutine. When calibrating the present-day mass balance, we needed to lapse rate correct the precipitation to get sufficient accumulation in the mass balance compared to observations. The consequence of this, is that the gridbox mean precipitation is no longer conserved. We tested scaling the precipitation, in a way that conserves the gridbox mean by reducing the precipitation near the surface and increasing it at height, but this did not yield enough precipitation to get a good agreement with the mass balance observations. If the model is being used to simulate river discharge in glaciated catchments, then the precipitation lapse rate could be used as a parameter to calibrate the discharge.

### 2.2.4 Wind speed

A component of the energy available to melt ice, comes from the sensible heat flux which is related to the temperature difference between the surface and the elevation level and the wind speed. Glaciers often have katabatic (downslope) winds which enhance the sensible heat flux and increase melting (Oerlemans and Grisogono, 2002). It is important to represent the effects of katabatic winds on the mass balance when trying to model glacier melt, particularly at lower elevations where the katabatic winds speed is highest.

To explicitly model katabatic winds would require knowledge of the gridbox mean slope at elevation bands, so instead a simple scaling of the surface wind speed is used to represent katabatic winds. Over glaciated grid boxes the wind speed is

$$u_z = u_0 \gamma_{wind} \tag{13}$$

where $\gamma_{wind}$ is a wind speed scale factor and $u_0$ is the surface wind speed. The simple scaling increases the wind speed relative to the surface forcing data and assumes that the scaling is constant for all heights.

Although our approach is rather crude, we found that scaling the wind speed was necessary to get reasonable values for the sensible heat flux. This is seen when we compare the modelled energy balance components to observations from the Pasterze glacier in the Alps (Greuell and Smeets, 2001). The measurements consist of incoming and outgoing short and long wave radiation, albedo, temperature, wind speed and roughness length at five heights between 2205m-3325m meters above sea level on the glacier. Table S6 in the Supplementary Material lists the observed and modelled energy balance components and meteorological data, for experiments with and without wind speed scaling. The comparison shows that JULES underestimates the sensible heat flux by at least one order of magnitude and the modelled wind speed is four times lower than the observations. When we increase the wind speed to match the observations there is a better agreement with the observed sensible heat flux. The surface exchange coefficient which is used to calculate the sensible heat flux is a function of the wind speed in the model.

## 2.2 Glacier ice albedo scheme

The existing spectral albedo scheme in JULES simulates the darkening of fresh snow as it undergoes the process of aging (Warren and Wiscombe, 1980). The growth rate of the grain is an empirically derived function of the snowpack temperature. The snow aging scheme does not reproduce the low albedo values typically observed on glacier ice, therefore a new albedo scheme is used. The new scheme is a density-dependent parameterization which was developed for the implementation in the Surface Mass Balance and Related Sub-surface processes (SOMARS) model (Greuell and Konzelmann, 1994). The scheme linearly scales the albedo from the value of fresh snow, to the value of ice, based on the density of the snowpack surface. The new scheme is used when the surface density of the top 10cm of the snowpack ($\rho_{surface}$) is greater than the firn density (550 kgm$^{-3}$) and the original snow aging scheme is used when ($\rho_{surface}$) is less than the firn density.

$$\alpha_\lambda = \alpha_{\lambda,ice} + \left(\rho_{surface} - \rho_{ice}\right)\left(\frac{\alpha_{\lambda,snow} - \alpha_{\lambda,ice}}{\rho_{snow} - \rho_{ice}}\right) \tag{14}$$

$\alpha_{\lambda, snow}$ is the maximum albedo of fresh snow, $\alpha_{\lambda, ice}$ is the albedo of melting ice, $\rho_{snow}$ is the density of fresh snow (250 kgm$^{-3}$) and $\rho_{ice}$ is the density of ice (917 kgm$^{-3}$). The albedo scaling is calculated separately in two radiation bands; visible wavelengths $\lambda = 0.3$–$0.7\mu m$ (VIS) and near-infrared wavelengths $\lambda = 0.7$–$5.0\mu m$ (NIR). The parameters, $\alpha_{vis, ice}$, $\alpha_{vis\ snow}$, $\alpha_{nir, ice}$, $\alpha_{nir, snow}$, $\gamma_{temp}$, $\gamma_{precip}$ and $\gamma_{wind}$ are tuned to obtain the best agreement between simulated and observed surface mass balance profiles for the present-day (see section 3).

## 2.3 Initialisation

The model requires initial conditions for (1) the snowpack properties (2) glaciated and unglaciated elevated tile fractions within a gridbox. The location of glacier grid points, the initial tile fraction and the present-day ice mass is set using data from the Randolph Glacier Inventory Version 6 (RGI6) (RGI Consortium, 2017). This dataset contains information on glacier hypsometry and is intended to capture the state of the world's glaciers at the beginning of the 21st century. A new feature of the RGI6 is a 0.5-degree gridded glacier volume and area datasets, produced at 50m elevation bands. Volume was constructed for individual glaciers using an inversion technique to estimate ice thickness created using glacier outlines, a digital elevation model and a technique based on the principles of ice flow mechanics (Farinotti et al., 2009;Huss and Farinotti, 2012). The area and volume of individual glaciers have been aggregated onto 0.5-degree grid boxes. We bin the 50m area and volume into elevations bands varying from 0m to 9000m in increments of 250m to match the elevation bands prescribed in JULES.

### 2.3.1 Initial tile fraction

The elevated glaciated fraction is

$$frac_{ice(n)} = \frac{RGI\_area(n)}{gridbox\_area(n)} \tag{15}$$

where *RGI_area* is the area (km$^2$) at height from the RGI6, *n* is the tile elevation and *gridbox_area* (km$^2$) is the area of the gridbox. In this configuration of the model, any area that is not glaciated is set to a single unglaciated tile fraction (*frac_rock*) with a gridbox mean elevation. It is possible to have an unglaciated tile fraction at every elevation band, but since the glaciated tile fractions does not grow or shrink, we reduce our computation cost by simply putting any unglaciated area into a single tile fraction.

$$frac_{rock} = 1 - \sum_{n=1}^{n=nBands} frac_{ice}(n) \tag{16}$$

nBands = 37 is the number of elevation bands.

### 2.3.2 Initial snowpack properties

The snowpack is divided into ten levels in which the top nine levels consist of 5m of firn snow with depths [0.05m, 0.1m, 0.15m, 0.2m, 0.25m, 0.5m, 0.75m, 1m, 2m] and the bottom level has a variable depth. For each snowpack level the following properties must be set; density (kgm$^{-3}$), ice content (kgm$^{-2}$), liquid water content (kgm$^{-2}$), grain size (µm) and temperature ($^o$K). We assume there is no liquid content in the snowpack by setting this to zero. The density at each level is linearly scaled with depth, between the value for fresh snow at the surface (250kgm$^{-3}$), to the value for ice at the bottom level (917kgm$^{-3}$).

For the future simulations the thickness and ice mass at the bottom of the snowpack comes from thickness and volume data in the RGI6. The data is based on thickness inversion calculations from Huss and Farinotti (2012) for individual glaciers which are consolidated onto 0.5-degree gridboxes. The ice mass is calculated from the RGI6 volume assuming an ice density 917 kgm$^{-3.}$ For the other layers the ice mass is calculated by multiplying the density by the layer thickness which is prescribed above. For the calibration period, the ice mass at the start of the run (1979) is unknown. In the absence of any information about this, a constant depth of 1000m is used which is selected to ensure that the snowpack never completely depletes over the calibration period. This consists of 995m of ice at the bottom level of the snowpack and 5m of firn in the layers above. The ice content of the bottom level is the depth (995m) multiplied by the density of ice.

The snow grain size used to calculate spectral albedo (see section 2.2) is linearly scaled with depth and varies between 50µm at the surface for fresh snow to 2000µm at the base for ice. The snowpack temperature profile is calculated by spinning the model up for 10 years for the calibration period and 1 year for the future simulations. The temperature at the top layer of the snowpack is set to the January mean temperature and the bottom layer and subsurface temperature is set to the annual mean temperature. For the calibration period the monthly and annual temperature comes from the last year of the spin-up. Setting the snowpack temperature this way gives a profile of warming towards the bottom of the snowpack representative of geothermal warming from the underlying soil. The initial temperature of the bedrock before the spin up is set to 0$^o$C but this adjusts to the climate as the model spins up. We use these prescribed snowpack properties as the initial state for the calibration and future runs.

# 3 Mass balance calibration and validation

## 3.1 Model calibration

Elevation-dependent mass balance is calibrated for the present-day by tuning seven model parameters and comparing the output to elevation-band specific mass balance observations from the World Glacier Monitoring Service ((2017). Calibrating mass balance against in-situ observations is a technique which has been used by other glacier modelling studies (Radić and Hock, 2011;Giesen and Oerlemans, 2013;Marzeion et al., 2012). For the calibration, annual elevation-band mass balance observations are used because there is data available for sixteen of the eighteen RGI6 regions. For validation, winter and summer elevation-band mass balance is used because there is less data available.

The tuneable parameters for mass balance are; visible snow albedo ($\alpha_{vis, snow}$), visible melting ice albedo ($\alpha_{vis, ice}$), near-infrared snow albedo ($\alpha_{nir, snow}$), near-infrared melting ice albedo ($\alpha_{nir, ice}$), orographic precipitation gradient ($\gamma_{precip}$), temperature lapse rate ($\gamma_{temp}$) and wind speed scaling factor ($\gamma_{wind}$).

Random parameter combinations are selected using Latin Hyper Cube Sampling (McKay et al., 1979) between plausible ranges which have been derived from various sources outlined below. This technique randomly selects parameter values; however, reflectance in the VIS wavelength is always higher than in the NIR. To ensure the random sampling does not select NIR albedo values that are higher or unrealistically close to the VIS albedo values, we calculate the ratio of VIS to NIR albedo using values compiled by compiled by Roesch et al (2002) . The ratio VIS/NIR is calculated as 1.2 so any albedo values that exceed this ratio are excluded from the analysis. This reduces the sample size from 1000 to 198 parameter sets.

In the VIS wavelength the fresh snow albedo is tuned between 0.99 - 0.7 where upper bound value comes from observations of very clean snow with little impurities in the Antarctic (Hudson et al., 2006). The lower bound represents contaminated fresh snow and comes from taking approximate values from a study based on laboratory experiments of snow, with a large grain size (110 µm) containing 1680 parts per billion of black carbon (Hadley and Kirchstetter, 2012). Visible snow albedos of approximately 0.7 have also been observed on glaciers with black carbon and mineral dust contaminants in the Tibetan Plateau (Zhang et al., 2017). In the NIR wavelength the fresh snow albedo is tuned between 0.85 – 0.5 where the upper bound comes from spectral albedo observations made in Antarctica (Reijmer et al., 2001). We use a very low minimum albedo for melting ice (0.1) the VIS and NIR wavelengths to capture dirty debris covered ice.

The temperature lapse rate is tuned between values of $4.0 – 10^\circ C$ $km^{-1}$ where the upper limit is determined from physically realistic bounds and lower limit is from observations based at glaciers in Alps (Singh, 2001). The temperature lapse rate in JULES is constant throughout the year and assumes that temperature always decreases with height.

The wind speed scaling factor $\gamma_{wind}$ is tuned within the range 1-4 to account for an increase in wind speed with height and for the presence of katabatic winds. The upper bound is estimated using wind observations made along the profile of the Pasterze glacier in the Alps during a field campaign (Greuell and Smeets, 2001). Table S6 in the Supplementary Material contains the wind speed observations on the Pasterze glacier. The maximum observed wind speed was 4.6 $ms^{-1}$ (at 2420 meters above sea-

level) while the WATCH-ERA Interim dataset (WFDEI) (Weedon et al., 2014) surface wind speed for the same time period was 1.1ms$^{-1}$ indicating a scaling factor of approximately 4.

The orographic precipitation gradient $\gamma_{precip}$ is tuned between 5-25%/100m. This parameter is poorly constrained by observations therefore a large tuneable range is sampled. Tawde et al. (2016) estimated a precipitation gradient of 19%/100m

for 12 glaciers in the Western Himalayas using a combination of remote sensing and in-situ meteorological observations of precipitation. Observations show that the precipitation gradient can be as high as 25%/100m for glaciers in Svalbard (Bruland and Hagen, 2002) while glacier-hydrological modelling studies have used much smaller values 4.3%/100m (Sorg et al., 2014) and 3%/100m  (Marzeion et al (2012). The tuneable parameters and their minimum and maximum ranges are listed in Table 1.

The model is forced with daily surface pressure, air temperature, downward longwave and shortwave surface radiation, specific humidity, rainfall, snowfall and wind speed from the WATCH-ERA Interim dataset (WFDEI) (Weedon et al., 2014).  To reduce the computation time, only grid points where glacier ice is present are modelled.  An ensemble of 198 calibration experiments are run.  For each simulation the model is spun up for 10 years and the elevation-dependent mass balance is compared to observations at 149 fields sites over the years 1979-2014.

The elevation-dependent mass balance observations come from stake measurements taken every year at different heights along the glaciers.  Many of the mass balance observations in the WGMS are supplied without observational dates. In this case, we assume the mass balance year starts on the 1$^{st}$ October to ends on the 30th September with the summer commencing on the 1$^{st}$ May. Dates in the Southern hemisphere are shifted by six months.   The observations are grouped according to standardised regions defined by the RGI6 (Fig. 2). The best regional parameter sets are identified by finding the minimum root mean square

error between the modelled mass balance and the observations.

Figure 3 shows the modelled mass balance profiles plotted against the observations using the best parameter set for each region. The best regional parameter sets are listed in Table 2 and the root mean square error, correlation coefficient, Nash–Sutcliffe efficiency coefficient and mean bias are listed in Table 3. Nine out of the sixteen regions have a negative bias in the annual mass balance.  Notably Svalbard, Southern Andes and New Zealand underestimate mass balance by 1 m.w.eq.yr$^{-1}$.  The

negative bias is also seen in the summer and winter mass balance and discussed in Section 3.2. The model performs particularly poorly for the Low Latitude region which has a large RMSE (3.02 m.w.eq.yr$^{-1}$). This region contains relatively small tropical glaciers in Colombia, Peru, Ecuador, Bolivia and Kenya. Marzeion et al (2012) found a poor correlation with observations in the low latitude region when they calibrated their glacier model using CRU data. They attributed that to the fact that sublimation was not included in their model, a process which is important for the mass balance of tropical glaciers. Our mass balance

model includes sublimation, so it is possible the WFDEI data over tropical glaciers is too warm. The WFDEI data is based on ERA-interim reanalysis where air temperature has been constrained using CRU data. The CRU data comprises of temperature observations which are sparse in regions where tropical glaciers are located. Furthermore, the quality of the WFDEI data will depend on the performance of the underlying ECMWF model. In Central Europe some of the poor correlation with observations is caused by the Maladeta glacier in the Pyrenees (Fig. 3) which is a small glacier with an area of 0.52 km$^2$ (WGMS, 2017).

When this glacier is excluded from the analysis the correlation coefficient increases from 0.26 to 0.35 and the RMSE decreases from 2.03 to 1.73 meters of water equivalent per year.

## 3.2 Model validation

The calibrated mass balance is validated against summer and winter elevation-band specific mass balance for each region
where data is available (Fig. 4). For all regions, except Scandinavia in the summer, negative Nash-Sutcliffe numbers are calculated for winter and summer elevation-dependent mass balance (Table 4). The negative numbers arise because the bias in the model is larger than the variance of the observations. There are negative biases for nearly all regions implying that melting is overestimated in the summer and accumulation is underestimated in the winter. This means that future projections of volume loss presented in section 4.2 might be overestimated.
The reason for the negative bias is because the model underestimates the precipitation and therefore the accumulation part of the mass balance is underestimated. This is because our approach to correcting the coarse scale gridded precipitation for orographic effects is simple. We use a single precipitation gradient for each RGI6 region and do not apply a bias correction. A bias correction is often recommended because precipitation is underestimated in coarse resolution datasets. Gauging observations are sparse in high mountains regions and snowfall observations can be susceptible to undercatch by 20–50%
(Rasmussen et al., 2012). Our precipitation rates are generally too low because we do not bias correct the precipitation.

Other studies use a bias correction that varies regionally (Radić and Hock, 2011;Radić et al., 2014;Bliss et al., 2014). In those studies, the precipitation at the top of the glacier was estimated using a bias correction factor $k_p$. The decrease in precipitation from the top of the glacier to the snout was calculated using a precipitation gradient. To account for the fact that the mass balance of maritime and continental glaciers respond differently to precipitation changes $k_p$ was related to a continentality
index. Our motivation for using a single precipitation gradient for each RGI6 region, and no bias correction was to test the simplest approach first, however the resulting biases suggest that this approach could be improved.

The impact of underestimating the precipitation is that we simulate negative mass balance in winter at some observational sites (Fig 5(A) and Fig 4.). To demonstrate this, we compare the mass balance components for two glaciers; the Leviy Aktru in the Russian Altai Mountains which has negative mass balance in the winter and Kozelskiy glacier in North Eastern Russia which
has no negative mass balance in the winter (See Fig. S9). Both glaciers are in the North Asia RGI6 region, so have the same tuned parameters for mass balance. The simulated winter accumulation rates are much lower at Leviy Aktru glacier than Kozelskiy glacier leading to negative mass balance at the lowest 3 model levels below 2750m.

The simplistic treatment of the precipitation lapse rate also leads to instances where the model simulates positive mass balance in the summer at some locations (Fig 6 (A) and Fig 4.). We show the summer mass balance components for the same two
glaciers in Fig S10. Positive mass balance is simulated at Kozelskiy glacier because accumulation exceeds the melting. This suggests that the precipitation gradient (19% per 100m for North Asia) is overly steep in the summer at this location.

Another reason we underestimate the accumulation is due to the partitioning of rain and snow based on an air temperature threshold of 0°C. The 0°C threshold is likely too low, resulting in an underestimate of snowfall. When precipitation falls as rain or snow it adds liquid water or ice to the snowpack. The specific heat capacity of the snowpack is a function of the liquid water ($W_k$) and ice content ($I_k$) in each layer ($k$)

$$C_k = I_k C_{ice} + W_k C_{water} \tag{17}$$

where $C_{ice} = 2100$ JK$^{-1}$kg$^{-1}$ and $C_{water} = 4100$ JK$^{-1}$kg$^{-1}$. The liquid water content is limited by the available pore space in the snowpack, therefore changes in the snowfall (ice content) control the overall heat capacity. The underestimate in the ice content reduces the heat capacity which causes more melting than observed.

Other modelling studies have used higher air temperature thresholds; 1.5°C (Huss and Hock 2015, Giesen and Oerlemans 2012), 2°C (Hirabayashi et al 2010) and 3°C (Marzeion et al 2012). An improved approach would use the wet-bulb temperature to partition rain and snow which would include the effects of humidity on temperature. Alternatively, a spatially varying threshold based on precipitation observations could be used. Jennings et al (2018) showed by analysing precipitation observations, that the temperature threshold varies spatially and generally higher for continental climates than maritime climates.

Winter mass balance is simulated better than summer mass balance, which is seen by the lower root mean square errors for winter in Table 4. Furthermore, the biases are larger in the summer than in the winter (Table 4). It is likely that the simple albedo scheme, which relates albedo to the density of the snowpack surface, performs better in the winter when snow is accumulating than in summer when there is melting. Fig 5 (B-D) and Fig. 6 (B-D) shows the winter and summer mass balances for all observation sites when area thresholds of 100km$^2$, 300km$^2$ and 500km$^2$ are applied to the validation. There is an improvement in the simulated summer mass balance when the glaciated area increases. This is seen by the improved correlation in Fig. 6(D) in which the validation is repeated but only grid boxes with a glaciated area greater than 500km$^2$ are considered. This indicates the model is better a simulating summer melting over regions with a large ice extent, than over regions with a small glaciated area.

## 4 Glacier volume projections

### 4.1 Downscaled climate change projections

Glacier volume projections are made for all regions, excluding Antarctica, for a range of high-end climate change scenarios. This is defined as climate change that exceeds 2°C and 4°C global average warming, relative to the pre-industrial period (Gohar et al., 2017). Six models fitting this criterion were selected from the Coupled Model Intercomparison Project Phase 5 (CMIP5). A new set of high resolution projections were generated using the HadGEM3-A Global Atmosphere (GA) 6.0 model (Walters et al., 2017). The sea surface temperature and sea-ice concentration boundary conditions for HadGEM3-A are supplied by the CMIP5 models. All models use the RCP8.5 'business as usual scenario' and cover a wide range of climate sensitivities, with

some models reaching 2°C global average warming relative to the pre-industrial period, quickly (IPSL-CM5A-LR) or slowly (GFDL-ESM2M) (Table 5). The models also cover a range of extreme wet or dry climate conditions. This is important to consider for glaciers in the central and eastern Himalaya which accumulate mass during the summer months due to monsoon precipitation (Ageta and Higuchi, 1984) and because future monsoon precipitation is highly uncertain in the CMIP5 models (Chen and Zhou, 2015).

The HadGEM3-A data are bias corrected using a trend preserving statistical bias method that was developed for the first Inter-Sectoral Impact Model Intercomparison Project (ISI-MIP) (Hempel et al., 2013). This technique uses WATCH forcing data (Weedon et al., 2011) to correct offsets in air pressure, temperature, longwave and shortwave downward surface radiation, rainfall, snowfall and wind speed but not specific humidity. The method adjusts the monthly mean and daily variability in the GCM variables but still preserves the long-term climate signal. The HadGEM3-A was bi-linearly interpolated from its native resolution of N216 (~60km), onto a 0.5-degree grid, to match the resolution of the WATCH forcing data which was used for the bias correction. The daily bias corrected surface fields from the HadGEM3-A are used to run JULES offline to calculate future glacier volume changes. The bias correction was only applied to data up until the year 2097, which means the glaciers projections terminate at this year. A flow chart of the experimental set up is shown in Fig. 7. The HadGEM3-A climate data was generated and bias corrected for the High-End cLimate Impact and eXtremes (HELIX) project.

## 4.2 Regional glacier volume projections 2011-2097

Glaciated areas are divided into 18 regions defined by the RGI6 with no projections made for Antarctic glaciers because the bias correction technique removes the HadGEM3-A data from this region. JULES is run for this century (2011 to 2097) using the best regional parameter sets for mass balance found by the calibration procedure (Table 2). No observations were available to determine the best parameters for Iceland and the Russian Arctic, therefore global mean parameter values are used for these regions. End of the century volume changes (in percent) are found by comparing the volume at end of the run (2097) to the initial volume calculated from the RGI6. Regional volume changes expressed in percent for low (0-2000m), medium (2250m-4000m) and high (4250m-9000m) and all elevation ranges (0-9000m) is listed in Table 6. The total volume loss over all elevation ranges is also listed in mm of sea-level equivalent in Table 6 and plotted in Fig. 10. Maps of the percentage volume change at the end of the century, relative to the initial volume are contained in the Supplementary Material in Figs. S1-S7.

A substantial reduction in glacier volume is projected for all regions (Fig. 8). Global glacier volume is projected to decrease by 64±5% by end of the century, where the value corresponds to the multi-model mean ± one standard deviation. The regions which lose more than 75% of their volume by the end of the century are; Alaska (-89 ± 2%), Western Canada and US (-100 ± 0%), Iceland (-98 ± 3%), Scandinavia (-98 ± 3%), Russian Arctic (-79 ± 10% ), Central Europe (-99 ± 0 %), Caucasus (-100 ± 0 %), Central Asia (-80 ± 7%), South Asia West (-98 ± 1%), South Asia East (-95 ± 2 %), Low Latitudes (100 ± 0 %), Southern Andes (-98 ± 1%) and New Zealand (-88±5%). The HadGEM3-A forcing data shows these regions experience a strong warming. In most regions this is combined with a reduction in snowfall relative to the present day, which is drives the mass loss (Fig. 9). Regions most resilient to volume losses are Greenland (-31 ± 5%) and Arctic Canada North (-47 ± 3%). In the

case of Arctic Canada North snowfall increases relative to the present day which helps glaciers to retain their mass. There is a rapid loss of low latitude glaciers which has also been found by other global glaciers models (Marzeion et al., 2012;Huss and Hock, 2015). Our model overestimates the melting of these glaciers for the calibration period (Fig. 3), so this result should be treated with a degree of caution. Some of the high latitude regions particularly Alaska, Western Canada & US, Svalbard and North Asia experience very large volume increases at their upper elevation ranges. This would be reduced if the model included glacier dynamics, because ice would be transported from higher elevations to lower elevations. The ensemble mean global sea level equivalent contribution is 215.2 ± 21.3 mm. The largest contributors to sea-level rise are Alaska (44.6 ± 1.1mm), Arctic Canada North and South (34.9 ± 3.0mm), Russian Arctic (33.3 ± 4.8mm), Greenland (20.1±4.4), High Mountain Asia (combined Central Asia, South Asia East and West), (18.0 ± 0.8mm), Southern Andes (14.4 ± 0.1mm) and Svalbard (17.0 ± 4.6mm). These are the regions which have been observed by the Gravity Recovery and Climate Experiment (GRACE) satellite to have lost the most mass in the recent years (Gardner et al., 2013).

To investigate which parts of the energy balance are driving the future melt rates, we show the energy balance components averaged over all regions and all elevation levels in Figure 11. Future melting is caused by a positive net radiation of approximately 30 $Wm^{-2}$ that is sustained throughout the century. This is comprised of 18 $Wm^{-2}$ net shortwave, 3 $Wm^{-2}$ net longwave, 5 $Wm^{-2}$ latent heat flux and 4 $Wm^{-2}$ sensible heat flux. The largest component of the radiation for melting comes from the net shortwave radiation. The upward shortwave radiation comprises of direct and diffuse components in the visible and near infrared wavelengths. The visible albedo deceases because melting causes the ice surface to darken. In contrast, the near infrared albedo increases because the ice is heating up emitting radiation in the infrared part of the spectrum.

The downward and upward longwave radiation are increasing in future however, the net longwave radiation contribution to the melting is small. The downward longwave radiation increases because of the $T^4$ relationship with air temperature, whereas the upward longwave radiation increases because the glacier surface is warming. The latent heat flux from refreezing of melt water and the sensible heat from surface warming, are also small components of the net radiation balance.

## 4.3 Mass Balance Components

In this section we examine how the surface mass balance components vary with height and how this will change in the future. Figure 12 shows the accumulation, refreezing and melting contributions to mass balance averaged over low, medium and high elevations ranges for the period 1980-2000. Sublimation is excluded because its contribution to mass balance is relatively small. As expected, there is more melting in the lower elevation ranges and more accumulation at the higher elevation ranges. The refreezing component, which includes refreezing of melt water and elevated adjusted rainfall, shows no clear variation with height. This is because the refreezing component can both increase and decrease with height. Refreezing can increases towards lever elevations because there is more rain and melted water. It can also decrease if the snowpack is depleted or if there is not enough pore space to hold water because previous refreezing episodes have converted the firn into solid ice. The largest accumulation rates occur in Alaska (5.3 m.w.eq.yr$^{-1}$) and Western Canada and US (7.3 m.w.eq.yr$^{-1}$) between 4250m-

9000m and the largest melt rates are found in the Caucasus and Middle East (-7.4 m.w.eq.yr$^{-1}$) and the Low Latitudes (-7.6 m.w.eq.yr$^{-1}$).

Figure 13 shows how the global annual mass balance components vary with time for low, medium and high elevations ranges. There is a reduction in accumulation and refreezing at all elevation ranges towards the end of the century. Melt rates decreases at medium and high elevation ranges because glaciers mass is lost at these altitudes, therefore less ice is available to melt (see Fig. S8 for the future cumulated mass balances as a function of height). Melt rates are constant at the low elevation ranges because there remains substantial quantities of ice available to melt at the end of the century in Greenland, Arctic Canada North and South, Svalbard, Russian Arctic. At high elevations mass balance is reduced from -2.2 m.w.eq.yr$^{-1}$ (-177 Gtyr$^{-1}$) during the historical period (1980-2000) to -0.35 m.w.eq. yr$^{-1}$ (-28 Gtyr$^{-1}$) by the end of the century (2080-2097). Similarly, for the medium elevation ranges mass balance reduces from -0.56 m.w.eq.yr$^{-1}$ (-26 Gtyr$^{-1}$) to -0.24 m.w.eq.yr$^{-1}$ (-11 Gtyr$^{-1}$).

## 4.4 Parametric uncertainty analysis

The standard deviation in the volume losses presented above are relatively small. This is because only a single GCM was used to downscale the CMIP5 data (HadGEM3-A). The uncertainty in the ensemble mean reflects the impact of the different sea-surface temperature and sea-ice concentration boundary conditions, provided by the CMIP5 models, on the HadGEM3-A climate. Other sources of uncertainty in the projections can arise from the calibration procedure, observational error, initial glacier volume and area, and structural uncertainty in the model physics. It is beyond the scope of this paper to investigate all the possible sources of uncertainty on the glacier volume losses. Instead we discuss the impact of parametric uncertainty in the calibration procedure in the following section.

In the calibration procedure the mass balance was tuned to obtain an optimal set of parameters for each RGI6 region, however, there may be other plausible parameter sets that perform equally well (i.e. for which the RMSE between the observations and the model is small). The principle of 'equifinality', in which the end state can be reached by many potential means, is important to explore because some parameters may compensate for each other. For example, the same mass balance could be reached by increasing the wind scale factor which enhances melting or decreasing the precipitation gradient which would reduce accumulation. To identify the experiments that perform equally well, we identify where there is a step change in the gradient of the RMSE for each RGI6 region. A similar approach was used by Stone et al. (2010) to explore the uncertainty in the thickness, volume and areal extent of the present-day Greenland ice sheet from an ensemble of Latin Hypercube experiments. The step change in the RMSE is identified using the changepoint detection algorithm called findchangepts (Rebecca et al., 2012;Lavielle, 2005) from the MATLAB signal processing toolbox. The algorithm is run to find where the mean of the top ten experiments (excluding the optimal experiment) changes the most significantly. For each RGI6 region the step changes in the RMSE are shown in Fig. 14.

JULES is re-run for each of the downscaled CMIP5 experiments and for each parameter set that is defined as performing equally well (See Table S1 in the Supplementary Material for a list of the parameters sets that perform equally well). The volume losses expressed in mm of sea-level equivalent are shown in Fig. 15. The effects of the parametric uncertainty on the

volume losses varies regionally, with the largest impact found for Central Europe and Greenland. Regional volume loses including parametric uncertainty in the calibration are summarised in Table 7. Including calibration uncertainty in this way gives an upper bound of 247.3 mm sea-level equivalent volume loss by the end of the century.

Another way to explore the uncertainty in the volume projections caused the calibration procedure, is to use different performance metrics to identify best parameters sets. In addition to using RMSE, we calculate best parameter sets by (1) minimising the absolute value of the bias and (2) maximizing the correlation coefficient. The best regional parameter sets are different depending on the choice of performance metric used (See Tables S2 and S3 in the Supplementary material). For twelve regions, minimising the bias results in higher precipitation lapse rates, than when RMSE values are used to select parameters. This suggests the bias in many regions is caused by underestimating the precipitation lapse rates. As discussed above, this could be due to the fact the gridbox mean WFDEI precipitation was not bias corrected. Glacier volume projections are generated by repeating the simulations using these two additional performance metrics to identify best parameter sets. The uncertainty in the global volume loss when the extra performance metrics are used, is approximately double the uncertainty arising from the different climate forcings (Fig. 16, Table 7). When extra performance metrics are used, the upper bound volume loss increases to 281.1 mm sea-level equivalent by the end of the century.

## 4.5 Comparison with other studies

We compare our end of the century volume changes (excluding parametric uncertainty), to two other published studies which used the CMIP5 ensemble under the RCP8.5 climate change scenario (Huss and Hock, 2015;Radić et al., 2014). Other studies exist, but these include the volume losses from Antarctic glaciers which makes a direct comparison difficult (Marzeion et al., 2012;Slangen et al., 2014;Giesen and Oerlemans, 2013;Hirabayashi et al., 2013). Huss and hock (2015) listed regional percentage volume change and sea-level equivalent values in their study while Radić et al. (2014) listed sea-level equivalent values only (See the comparison Tables S4 and S5 in the Supplementary material).

Our end of the century percentage volume losses compare reasonably well to Huss and Hock (2015) for Central Europe, Caucasus, South Asia East, Scandinavia, Russian Arctic, Western Canada and US, Arctic Canada North, North Asia, Central Asia, Low Latitudes and New Zealand but are significantly higher in the Southern Andes, Alaska, Iceland and Arctic Canada South. The uncertainty in our percentage volume losses are smaller than Huss and Hock (2015) because we only use a single GCM to downscale the CMIP5 experiments while Huss and Hock (2015) use 14 CMIP5 GCMs.

We estimate the end of century global sea-level contribution, excluding Antarctic glaciers, to be $215 \pm 20$mm which is higher than 188mm (Radić et al., 2014) and $136\pm23$mm (Huss and Hock, 2015) caused mainly by greater contributions from Alaska, Southern Andes and the Russian Arctic. These three regions are discussed in turn.

For the Southern Andes our estimates are approximately double (14.4mm) that of the other studies (5.8mm (Huss and Hock, 2015), 8.5mm (Radić et al., 2014)). This region has the largest negative bias in the calibrated present-day mass balance (-2.87 m.w.eq.yr$^{-1}$ see Table 3). To explore the effects of correcting the calibration bias on the ice volume projections, we subtract the bias values listed in Table 3 from the future annual mass balance rates. Each gridbox is assumed to have the same regional

mass balance bias. The bias corrected volume losses are listed in Table S5 in the supplementary material. For the Southern Andes, the volume losses are much closer to the other studies (7.6mm) when the bias is corrected. The impact is less for the other regions where the biases are smaller. For the Russian Arctic our volume losses are higher than the other studies but that should be interpreted with caution because there were no observations available in this region to get a tuned parameter set (global mean parameters where used instead). In Alaska the bias in annual mass balance is small (0.06 m.w.eq.yr$^{-1}$) so correcting the bias has little effect on the volume loss projection for this region. Applying the bias correction increases the global volume loss from $215 \pm 20$mm to $223 \pm 20$ mm, therefore the difference between our model and the other studies cannot be explained by the bias in the calibration.

It is likely that our SLE contributions are higher than the other studies because the climate forcing data is different. The HadGEM3-A model uses boundary conditions from a subset of RCP8.5 CMIP5 models with the highest warming levels. Furthermore, the HadGEM3-A data has a higher resolution (approximately 60km) than the CMIP5 data which was used by the other two studies. This means our model should, in theory, be more accurate at reproducing regional patterns in precipitation and temperature over complex terrain which is important for calculating mass balance.

**5 Discussion**

The robustness of the glacier projections depends on how well the model can reproduce present-day glacier mass balance. Our calibrated seasonal mass balance contains a negative bias (accumulation is underestimated, and melting is overestimated) which suggests the projections of volume loss might be overestimated. One of the main shortcomings of the calibration and validation of mass balance is that only a single type of observations is used. This data was used because we wanted to ensure the model could reproduce variations in accumulation and ablation with height when the elevated tiling scheme was introduced. Point mass balance observations are affected by local factors such as aspect, avalanching, debris cover and there is a possibility that these local factors affect parameter sets chosen for entire RGI region. This could be improved by using observations from satellite gravimetry and altimetry, such as that described by Gardner et al (2013) to get a quantitative estimate of the model performance at the regional scales.

One of the notable differences between our study and other global glacier models is that our tuned precipitation lapse rates are very high, for example, 24%/100m for South Asia West and 19%/100m for Central Asia. Other models have used lower precipitation lapse rates (1-2.5%/100m (Huss and Hock, 2015), 3%/100m (Marzeion et al., 2012)), but they also bias correct precipitation by multiplying by a scale factor. This scaling factor can be considerably high. Giesen and Oerlemans (2012) found that precipitation needed to be multiplied by a factor of 2.5 to get good agreement with mass balance observations. Radić and Hock (2011) derived, through calibration of present-day mass balance, a precipitation scale factor of as high as 5.6 for Tuyuksu and Golubina glaciers in the Tien Shan. Our lapse rates are high because we do not bias correct the precipitation using a multiplication factor for the present-day. For the future GCM data the gridbox mean precipitation was bias corrected using the ISI-MIP technique. The negative bias that we get when validating the present-day mass balance suggests that snowfall

is underestimated in our model. A future study using this model could test whether bias correcting the precipitation before applying the lapse rate correction improves the simulated mass balance.

This is the first attempt to implement a glacier scheme into JULES and so the model has many limitations. One of the key shortcomings is that glacier dynamics is not included (glacier area does not vary). The transport of ice from higher elevations to lower elevations is not included. This process could be included in future work by adding a volume-area scaling scheme Bahr (1997) or a thickness parameterisation based on glacier slope (Marshall and Clarke, 2000). Volume-area scaling has been used to model glacier dynamics in coarse resolution (0.5-degree) models where all glaciers in a gridbox are represented by a single ice body (Kotlarski et al., 2010;Hirabayashi et al., 2010). The current configuration of elevated glaciated and unglaciated tiles in JULES makes it well suited to a volume-area scaling model. This would be implemented by growing (shrinking) the elevated glaciated tiles if mass balance is positive (negative) at each elevation band. In the case where the elevated ice tile grows the unglaciated tile would shrink at that elevation band or vice versa.

The volume-area scaling law has been used successfully to model the dynamics of individual glaciers (Radić et al., 2014;Giesen and Oerlemans, 2013;Marzeion et al., 2012;Slangen et al., 2014) but has some limitations when applied to coarse models where glaciers are consolidated into a single gridbox. The volume-area scaling law, relates volume to area using a constant scaling exponent which is typically derived from a small sample of glacier observations (Bahr et al., 1997). One of the draw backs is that the law is non-linear, meaning the exponent derived from individual glaciers would overestimate the volume of a large ice grid box such as in our model (Hirabayashi et al., 2013). Furthermore, the exponent may not accurately represent the volume-area relationship in other geographical regions. To overcome these issues a spatially variable scaling exponent could be created using the newly available 0.5-degree data on volume and area contained in the RGI6.

Another limitation of the model, which may be problematic for same applications, is that the gridbox mean precipitation is not conserved when precipitation is adjusted for elevation. This correction was necessary to get enough accumulation in the mass balance at high elevations. One way to conserve water mass would be to reduce the precipitation onto the non-glaciated area of the grid cell. This would represent horizontal mass movement within the grid box from wind-blown snow and avalanching.

A further limitation of the model is the simple treatment of katabatic winds, which is modelled by scaling the synoptic wind speed. This could be improved by parameterising katabatic winds based on the gridbox slope and the temperature difference between the glacier surface and the air temperature using the Prandtl model (Oerlemans and Grisogono 2002). Another drawback of the model is the coarse resolution of the gridboxes which make it unfeasible to include some process which affect local mass balance such as hillside shading, avalanching, blowing snow and calving. The model could, however, be run on a finer resolution using higher resolution climate forcing data.

While this modelling projects considerable reduction in glacier mass for all mountain ranges by the end of this century, it is clear that many of the world's mountain glaciers will evolve in ways that are currently difficult to model. For instance, paraglacial processes during deglaciation lead to enhanced rock falls and debris flows from deglaciating mountain slopes and these deliver rock debris to glacier surfaces. This produces debris-covered glaciers and these are common in many mountain regions, including in Alaska, arid Andes, central Asia and in the Hindu Kush-Himalaya. Thick debris cover (decimetres to

metres) limits ice ablation, (e.g., (Lambrecht et al., 2011;Pellicciotti et al., 2014;Lardeux et al., 2016;Rangecroft et al., 2016)) and reverses the mass balance gradient, with comparatively higher ablation rates up glacier than at the debris-covered terminus. This significantly influences glacier dynamics (Benn et al., 2005), and with inefficient sediment evacuation eventually leads to the transition from debris-covered glaciers to rock glaciers (e.g (Monnier and Kinnard, 2017). In the context of continued

climate warming, the transition from ice glaciers to rock glaciers may enhance the resilience of the mountain cryosphere (Bosson and Lambiel, 2016). As a result, better assessment of the response of the mountain cryosphere to climate warming will depend on a clearer understanding of glacier-rock glacier relationships.

There are three key strengths to the JULES glacier model. Firstly, we include variations in orography within a climate gridbox which is important to calculate elevation-dependent glacier mass balance. Kotlarski et al (2010) developed a glacier scheme

for the REMO regional climate model by lumping glaciers into 0-5-degree gridboxes in a similar approach to us, but they did not have a representation of subgrid orography. Instead glacier gridboxes received double the gridbox mean snowfall, glacier ice had a fixed albedo and a constant lapse rate was applied to adjust temperatures. They concluded that to reproduce mass balance trends over the Alps, the scheme needed to include subgrid variability of atmospheric parameters within a gridbox. Secondly, the model uses a full energy balance scheme to calculate glacier melting. This is a more physically based approach

than the widely used temperature index models, which relate melting to temperature using a degree day factor (DDF). The DDF lumps all the energy balance components into a single number meaning that the effects of changing wind speed, cloudiness and radiation on melt rates cannot be considered. Changes in solar radiation can be an important driver of melting. Huss et al (2009) studied long term mass balance trends for a site in the Alps and showed that melting was stronger during the 1940's than in recent years despite more warming. This was because summer solar radiation was higher during the 1940s.

Moreover, temperature index models have been found to be less accurate with increasing temporal resolution (for example on daily time steps) (Hock, 2005). In this paper, we present a brief analysis of the future global energy balance fluxes, but how the fluxes vary for individual regions and elevation levels could be investigated further. Finally, the glacier scheme is coupled to a land surface model, which presents opportunities for further studies. For instance, the model could be used to investigate the impact of climate change on river discharge in glaciated catchments in Asia, South America or the Arctic.

**6 Conclusions**

The first aim of this study was to add a glacier component to JULES to develop a fully integrated model, to simulate the interactions between glacier mass balance, river runoff, water abstraction by irrigation and crop production. To do this we added two new surface types to JULES; elevated glaciated and unglaciated tiles. This allows us to the calculate elevation-dependent mass balance which can be used to study the response of glaciers to climate change. Glacier volume was modelled

by growing or shrinking the snowpack, using the elevation-dependent mass balance, but glacier dynamics was not included. Present-day mass balance was calibrated by tuning albedo, wind speed, temperature and precipitation lapse rates to obtain a set of regionally tuned parameters which are then used to model future mass balance. Winter and summer mass balances are

reproduced reasonably well for regions where the glaciated area is large, however, the model performs poorly for small glaciers particularly in the summer. The fully integrated model is potentially a useful tool for the scientific community to study the impact of climate change on food and water resources.

The second aim of this study was to make glacier volume projections for the future under a range of high-end climate change scenarios. The ensemble mean volume loss ± one standard deviation is, -64±5% for all glaciers excluding those on the periphery of the Antarctic ice sheet. The small uncertainties in the multi-model mean are caused by the sensitivity of HadGEM3-A to the boundary conditions supplied by the CMIP5 models. Our end of the century global volume loss is 215 ± 20mm, which is higher than values reported by other studies. This is because we used a subset of CMIP5 models with the highest warming levels to drive the model and glacier dynamics is not included which results in more mass loss than other studies that include dynamics. Including parametric uncertainty in the calibration procedure results in an upper bound global volume loss of 281.1 mm sea-level equivalent by the end of the century. The projected ice losses will have an impact on sea-level rise and on water availability in glacier-fed river systems.

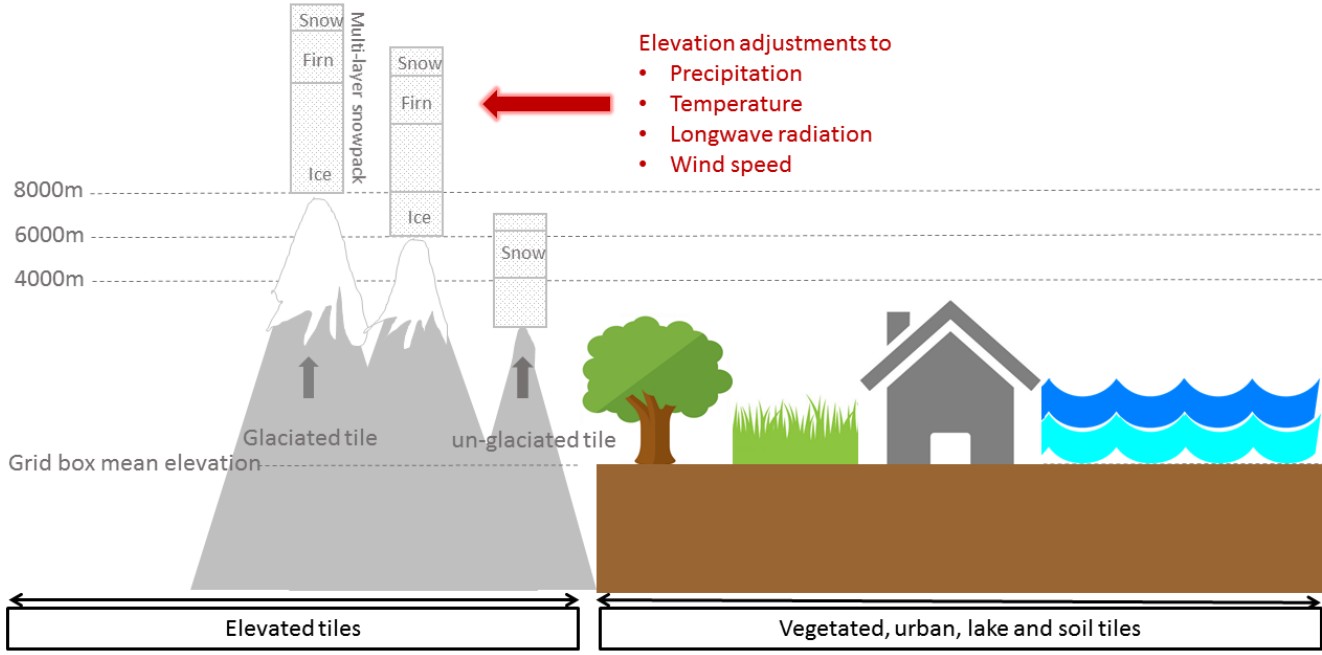

**Figure 1 Schematic of JULES surface types inside a single gridbox. The new elevated glaciated and unglaciated tiles are shown on the left-hand side. Note that elevated glaciated and unglaciated tiles are not allowed to share a gridbox with the other tiles.**

**Table 1 Tuneable parameters for mass balance calculation and their ranges from the literature**

| Parameter | Range of values | Symbol | Units |
|---|---|---|---|
| *Fresh snow albedo (VIS)* | 0.99 - 0.7 | $\alpha_{vis,\ snow}$ | - |
| *Fresh snow albedo (NIR)* | 0.85 - 0.5 | $\alpha_{nir\ snow}$ | - |
| *Ice albedo (VIS)* | 0.7 – 0.1 | $\alpha_{vis,\ ice}$ | - |
| *Ice albedo (NIR)* | 0.6 – 0.1 | $\alpha_{nir,\ ice}$ | - |
| *Temperature lapse rate* | 4 – 9.8 | $\gamma_{temp}$ | $^{\circ}K\ km^{-1}$ |
| *Orographic precipitation gradient* | 5 – 25 | $\gamma_{precip}$ | %/100m |
| *Wind speed scale factor* | 1 - 4 | $\gamma_{wind}$ | - |

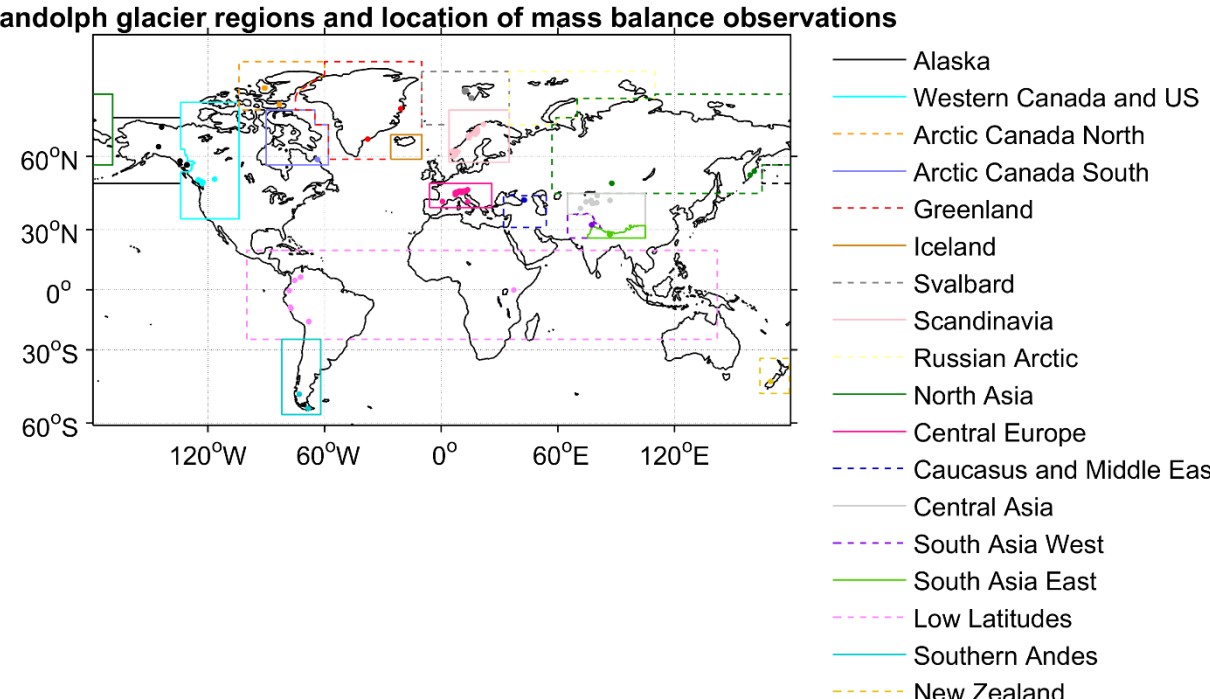

**Figure 2: The location of mass balance profile observations glaciers from the World Glacier Monitoring Service and the Randolph Glacier Inventory regions (version 6.0)**

5  **Table 2 Best parameter sets for each RGI6 region. The regions are ranked from the lowest to the highest RMSE. There are no observed profiles for Iceland and Russian Artic, so the global mean parameters values are used (bold) for the future simulations**

| Region | $\alpha_{vis,snow}$ | $\alpha_{nir,snow}$ | $\alpha_{vis,ice}$ | $\alpha_{nir,ice}$ | $\gamma_{temp}$ $^oK\ km^{-1}$ | $\gamma_{precip}$ $\%/100m$ | $\gamma_{wind}$ |
|---|---|---|---|---|---|---|---|
| Arctic Canada South | 0.94 | 0.77 | 0.68 | 0.53 | 8.3 | 16 | 2.15 |
| Arctic Canada North | 0.96 | 0.70 | 0.49 | 0.12 | 4.2 | 7 | 1.10 |
| Greenland | 0.95 | 0.72 | 0.41 | 0.19 | 8.0 | 15 | 1.07 |
| Alaska | 0.88 | 0.65 | 0.56 | 0.27 | 8.2 | 16 | 1.32 |
| South Asia East | 0.91 | 0.73 | 0.67 | 0.56 | 5.3 | 9 | 1.55 |
| South Asia West | 0.99 | 0.73 | 0.60 | 0.30 | 4.0 | 24 | 1.69 |
| Western Canada and US | 0.97 | 0.64 | 0.45 | 0.26 | 9.3 | 8 | 2.29 |

| | | | | | | | |
|---|---|---|---|---|---|---|---|
| *Central Asia* | 0.94 | 0.74 | 0.69 | 0.50 | 8.1 | 19 | 1.40 |
| *North Asia* | 0.94 | 0.74 | 0.69 | 0.50 | 8.1 | 19 | 1.40 |
| *Central Europe* | 0.83 | 0.63 | 0.59 | 0.35 | 5.8 | 7 | 1.83 |
| *Svalbard* | 0.95 | 0.76 | 0.54 | 0.35 | 9.0 | 14 | 1.02 |
| *Caucasus and Middle East* | 0.90 | 0.71 | 0.53 | 0.28 | 8.3 | 5 | 3.32 |
| *Scandinavia* | 0.95 | 0.76 | 0.54 | 0.35 | 9.0 | 14 | 1.02 |
| *New Zealand* | 0.94 | 0.74 | 0.69 | 0.50 | 8.1 | 19 | 1.40 |
| *Low Latitudes* | 0.94 | 0.74 | 0.69 | 0.50 | 8.1 | 19 | 1.40 |
| *Southern Andes* | 0.95 | 0.76 | 0.54 | 0.35 | 9.0 | 14 | 1.02 |
| ***Mean*** | **0.93** | **0.72** | **0.58** | **0.37** | **7.55** | **14** | **1.56** |

**Table 3 Root mean square error (RMSE), correlation coefficient (r), Nash–Sutcliffe efficiency coefficient (NS), mean bias (BIAS) and the number of elevation-band mass balance observations (No Obs) for RGI6 regions. The regions are ranked from the lowest to the highest RMSE.**

| Region | RMSE $m.w.eq.yr^{-1}$ | r | NS | Bias $m.w.eq.yr^{-1}$ | No Obs |
|---|---|---|---|---|---|
| *Arctic Canada South* | 0.96 | 0.61 | 0.11 | 0.10 | 72 |
| *Arctic Canada North* | 1.06 | 0.19 | -0.44 | 0.52 | 1,332 |
| *Greenland* | 1.09 | 0.66 | 0.14 | 0.14 | 90 |
| *Alaska* | 1.36 | 0.65 | 0.38 | 0.06 | 217 |
| *South Asia East* | 1.41 | 0.15 | -0.34 | -0.19 | 81 |
| *South Asia West* | 1.53 | 0.62 | 0.38 | -0.09 | 168 |
| *Western Canada and US* | 1.73 | 0.69 | 0.41 | -0.40 | 916 |
| *Central Asia* | 1.81 | 0.22 | -1.15 | -0.51 | 2,519 |
| *North Asia* | 1.95 | 0.45 | -0.04 | -0.21 | 1,335 |
| *Central Europe* | 2.03 | 0.26 | -0.65 | 0.30 | 9,561 |
| *Svalbard* | 2.16 | 0.36 | -6.86 | -1.21 | 1,647 |
| *Caucasus and Middle East* | 2.23 | 0.30 | -0.89 | 0.33 | 687 |
| *Scandinavia* | 2.40 | 0.53 | 0.20 | 0.67 | 10,617 |
| *New Zealand* | 2.57 | 0.58 | -0.30 | -1.09 | 45 |
| *Low Latitudes* | 3.06 | 0.36 | -0.71 | -0.88 | 1,016 |
| *Southern Andes* | 3.33 | 0.26 | -12.33 | -2.87 | 118 |
| *Global* | 2.16 | 0.40 | -0.11 | 0.19 | 30,421 |

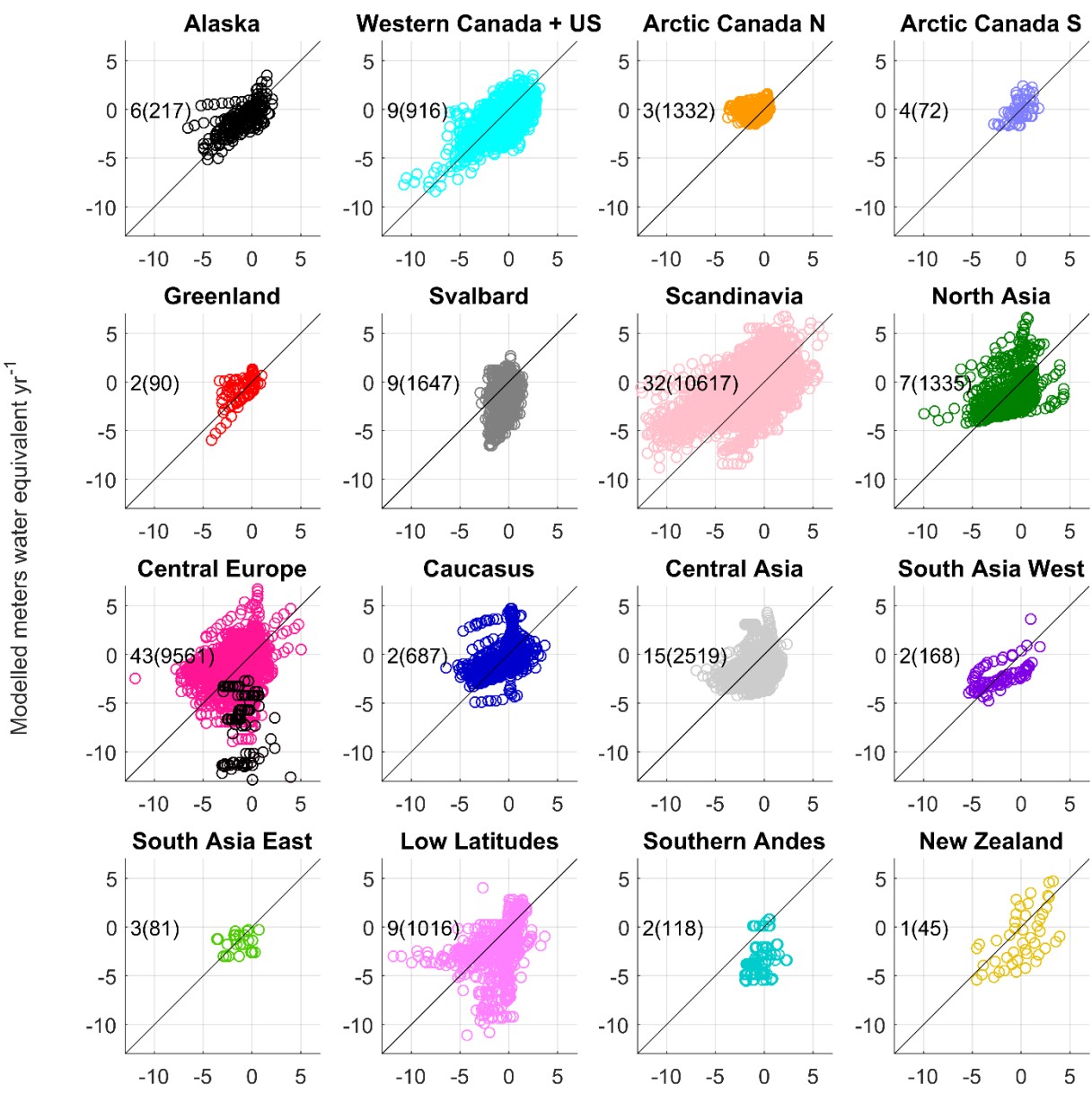

**Figure 3 Modelled annual elevation-dependent specific mass balance against observations from the WGMS. The modelled mass** **balance is simulated on a 0.5-degree grid resolution at 250m elevation bands and the observations are for individual glaciers at**

elevation levels specific to each glacier. The observed mass balance is interpolated onto the JULES elevation bands. If only a single observation exists, then mass balance for the nearest JULES elevation band is used. The number of glaciers is shown in the top left-hand corner and the number of observation points in brackets. In central Europe mass balance for the Maladeta glacier in the Pyrenees is shown in black circles.

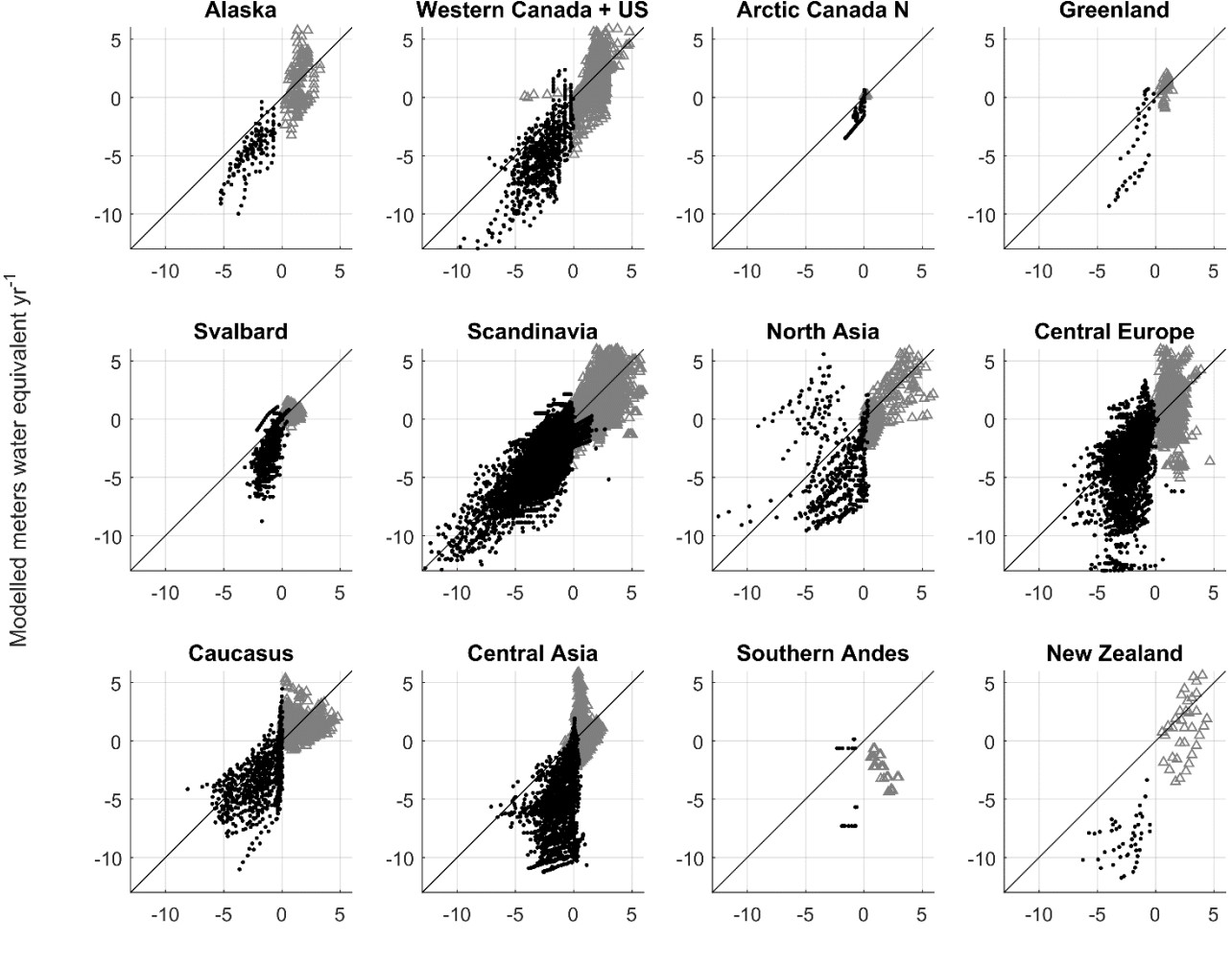

Figure 4 Comparison between modelled and observed elevation-band specific mass balance for winter (grey triangles) and summer (black dots). The modelled mass balance is calculated using the tuned regional parameters from the calibration procedure.

**Table 4 Winter (bold) and summer (italics) number of elevation-band mass balance observations (No obs), root mean square error (RMSE), correlation coefficient (r), Nash–Sutcliffe efficiency coefficient (NS) and mean bias (BIAS).**

| Region | No obs | | RMSE $m.w.eq.yr^{-1}$ | | r | | NS | | BIAS $m.w.eq.yr^{-1}$ | |
|---|---|---|---|---|---|---|---|---|---|---|
| *Alaska* | **127** | 127 | **1.82** | 2.43 | **0.38** | 0.76 | **-7.54** | -2.88 | **-0.29** | -2.09 |
| *Western Canada and US* | **767** | 729 | **1.76** | 2.96 | **0.53** | 0.72 | **-2.68** | -2.25 | **-0.34** | -2.28 |
| *Arctic Canada North* | **49** | 50 | **0.08** | 1.09 | **0.09** | 0.86 | **-0.94** | -5.01 | **0.04** | -0.79 |
| *Greenland* | **28** | 36 | **0.78** | 3.45 | **0.33** | 0.81 | **-11.31** | -11.13 | **-0.11** | -2.40 |
| *Svalbard* | **1,122** | 1,126 | **0.61** | 2.25 | **0.18** | 0.66 | **-3.90** | -12.59 | **-0.38** | -1.84 |
| *Scandinavia* | **5,347** | 10,679 | **1.52** | 1.69 | **0.61** | 0.78 | **-0.78** | 0.32 | **-0.68** | -0.77 |
| *North Asia* | **854** | 828 | **1.54** | 4.15 | **0.71** | 0.20 | **-0.40** | -3.81 | **-1.08** | -2.63 |
| *Central Europe* | **5,496** | 4,804 | **1.21** | 2.77 | **0.12** | 0.33 | **-5.83** | -4.63 | **-0.02** | -1.11 |
| *Caucasus + Middle East* | **602** | 677 | **1.39** | 2.30 | **-0.12** | 0.55 | **-1.15** | -0.94 | **-0.23** | -1.18 |
| *Central Asia* | **1,778** | 1,751 | **1.34** | 4.87 | **0.21** | 0.31 | **-10.57** | -16.92 | **-0.19** | -4.23 |
| *Southern Andes* | **34** | 22 | **4.19** | 4.11 | **-0.81** | -0.08 | **-36.73** | -55.59 | **-3.81** | -2.36 |
| *New Zealand* | **45** | 45 | **3.37** | 6.17 | **0.42** | 0.32 | **-10.63** | -17.82 | **-0.01** | -5.87 |
| *Global* | **16,249** | 20,874 | **1.38** | 2.16 | **0.49** | 0.78 | **-1.16** | 0.11 | **-0.37** | -0.92 |

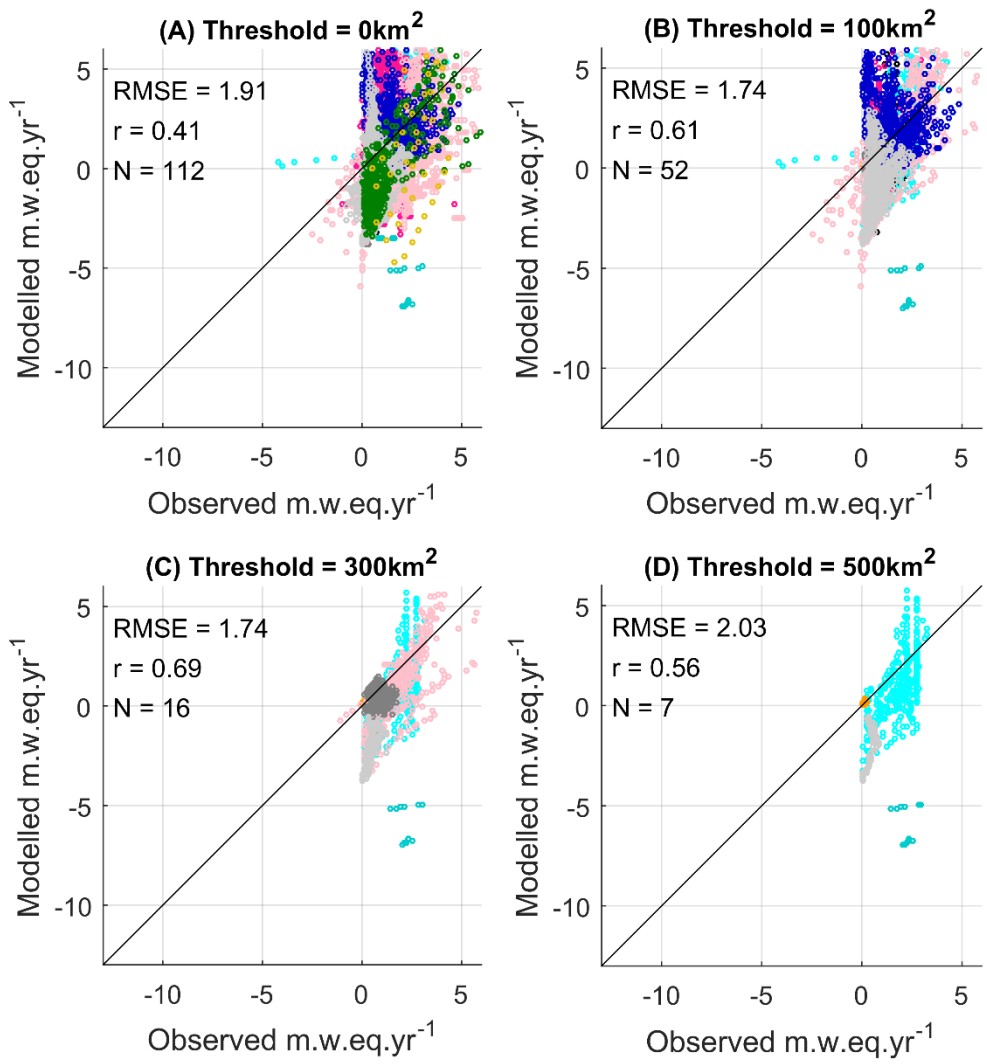

**Figure 5 Simulated and observed elevation-dependent winter mass balance when gridboxes with a glacier area of less than 100km², 300km² and 500km² are excluded. The colour identifies the RGI6 regions shown in Figure 2. The RMSE, correlation coefficient and number of glaciers are listed.**

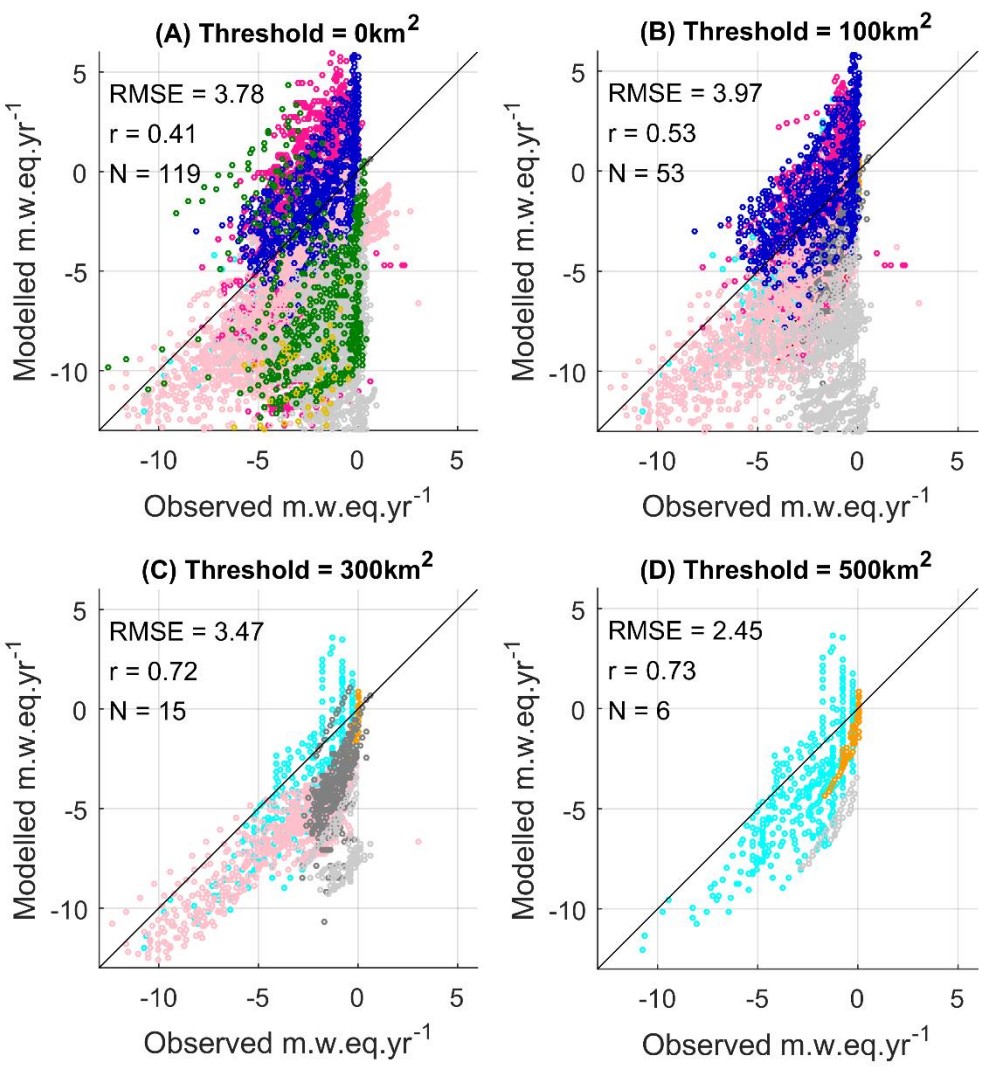

**Figure 6 Simulated and observed elevation-dependent summer mass balance when gridboxes with a glacier area of less than 100km², 300km² and 500km² are excluded. The colour identifies the RGI6 regions shown in Figure 2. The RMSE, correlation coefficient and number of glaciers are listed.**

**Table 5 List of high-end climate change CMIP5 models that are downscaled using HadGEM3-A. The years when the CMIP5 models pass +1.5ºC, +2ºC and +4ºC global average warming relative to the pre-industrial period are shown. *No data is available for 2113 because the bias corrected data ends at 2097.**

| CMIP5 model | Ensemble member | + 1.5ºC | + 2ºC | + 4 ºC |
|---|---|---|---|---|
| IPSL-CM5A-LR | r1i1p1 | 2015 | 2030 | 2068 |
| GFDL-ESM2M | r1i1p1 | 2040 | 2055 | 2113* |
| HadGEM2-ES | r1i1p1 | 2027 | 2039 | 2074 |
| IPSL-CM5A-MR | r1i1p1 | 2020 | 2034 | 2069 |
| MIROC-ESM-CHEM | r1i1p1 | 2023 | 2035 | 2071 |
| HELIX ACCESS1-0 | r1i1p1 | 2034 | 2046 | 2085 |

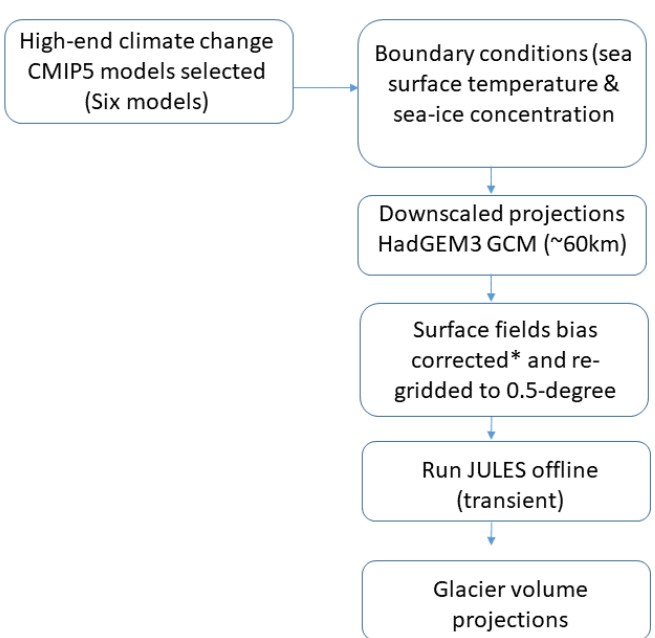

**Figure 7 Flow chart showing the experimental set up to calculate future glacier volume. *The bias correction method is described by Hempel et al. (2013b)**

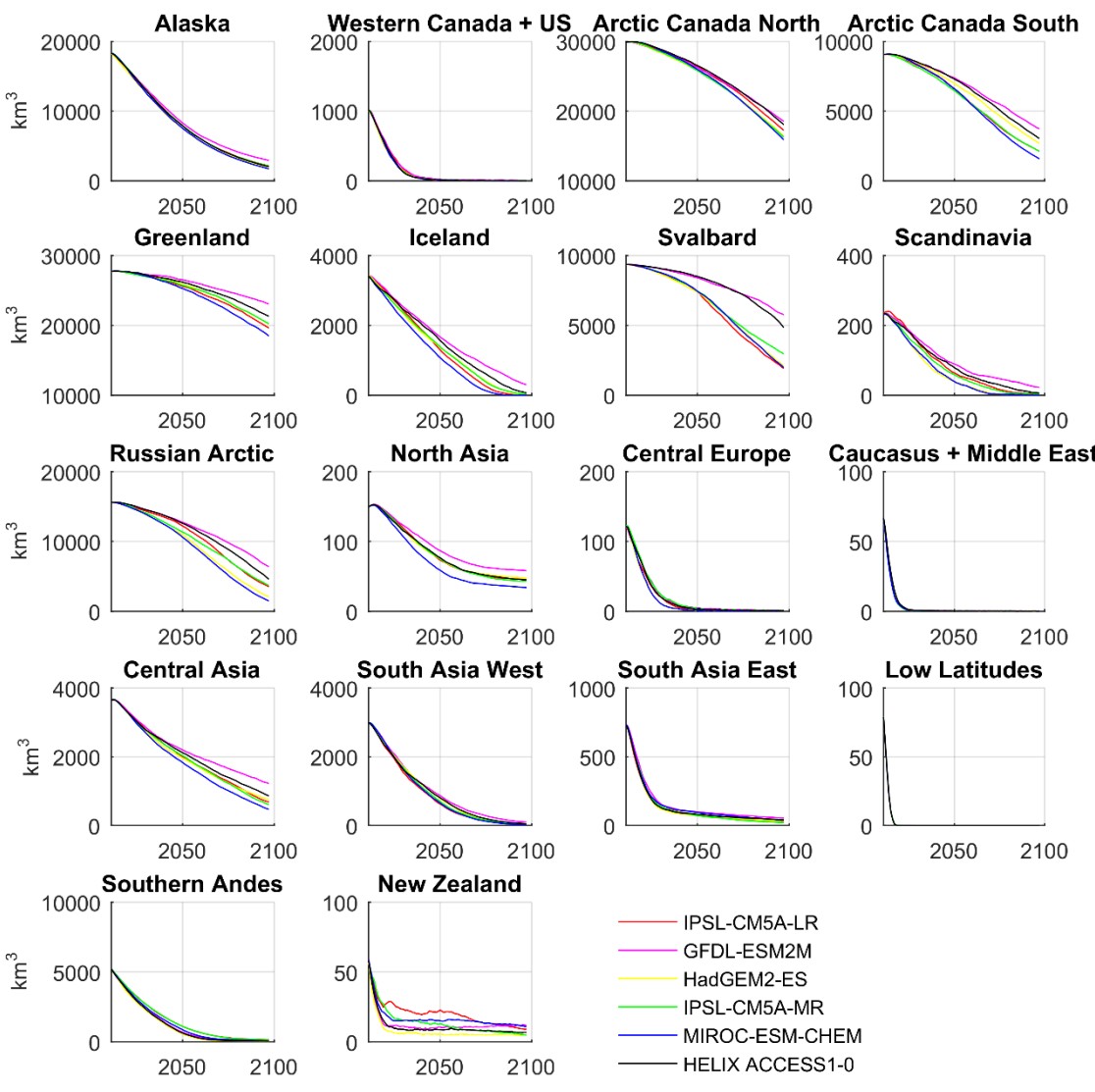

**Figure 8 Regional glacier volume projections using the HadGEM3-A ensemble of high-end climate change scenarios.**

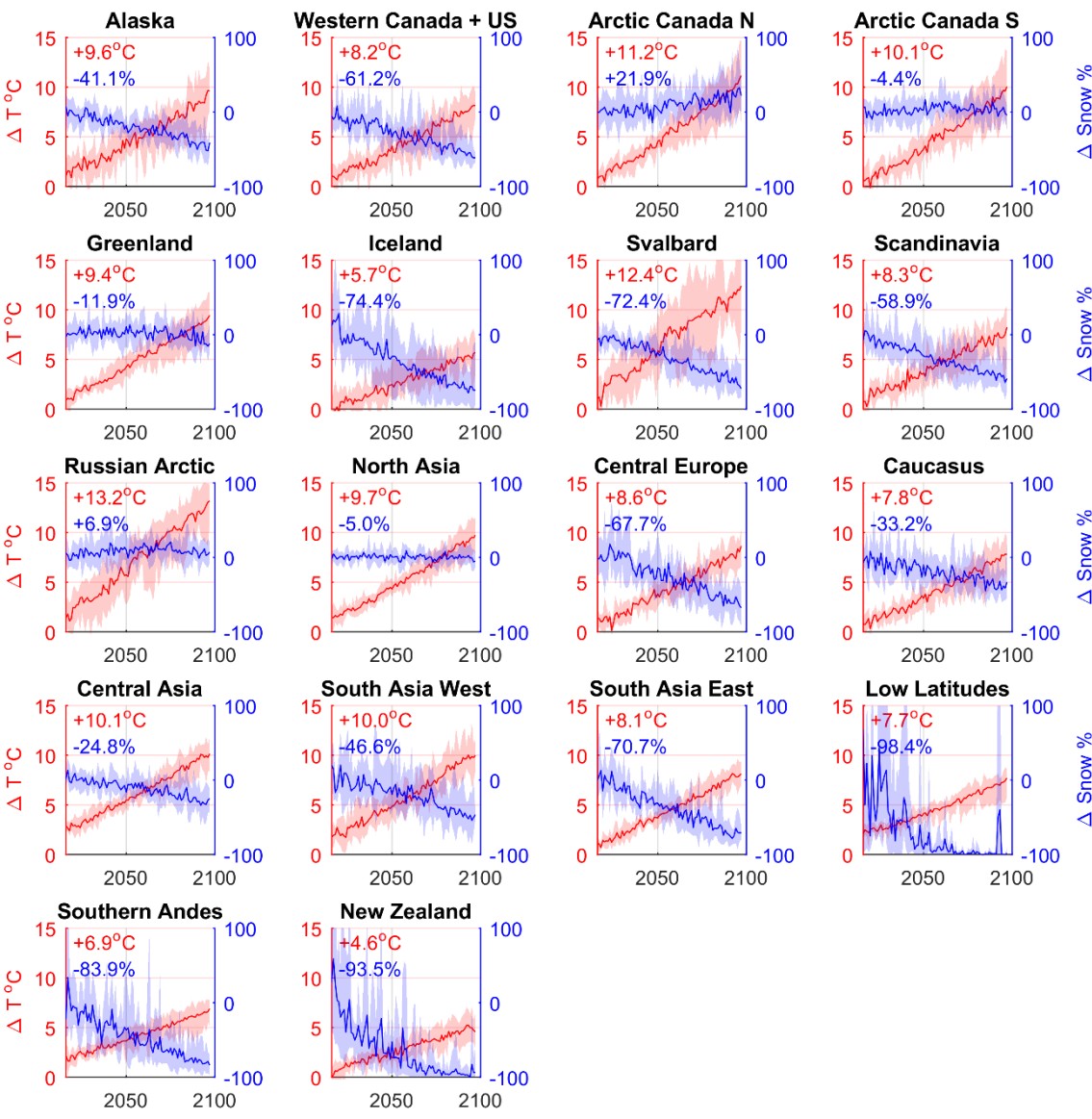

**Figure 9 Regional temperature and snowfall changes relative to present day (2011-2015) from the HadGEM3-A ensemble over glaciated grid points. The ensemble mean is shown in the solid line and the range of model projections are shown in the shaded regions.**

**Table 6** Percentage ice volume loss, relative to the initial volume (ΔV) and ice loss in mm of Sea Level Equivalent (SLE) for the end of the century (2097). Percentage volume loses are shown for low, medium and high elevation ranges as well as for all elevations. The data shows the multi-model mean ± one standard deviation. The conversion of volume to SLE assumes an ocean area of 3.618 x $10^8$ km$^2$. The initial area and volume from the Randolph Glacier Inventory Version 6 is listed in columns 1 and 2.

| | Area | Volume | ΔV 0m-9000m | ΔV 0m-2000m | ΔV 2250m-4000m | ΔV 4250m-8000m | SLE |
|---|---|---|---|---|---|---|---|
| | km$^2$ | km$^3$ | % | % | % | % | mmSLE |
| Alaska | 86,616 | 19,743 | -89±2 | -93±1 | -55±9 | 408±18 | 44.6±1.1 |
| Western Canada and US | 14,357 | 1,070 | -100±0 | -100±0 | -99±0 | 684±136 | 2.8±0.0 |
| Arctic Canada North | 104,920 | 32,376 | -47±3 | -43±4 | 40±1 | - | 35.8±3.0 |
| Arctic Canada South | 40,861 | 9,780 | -74±8 | -72±9 | - | - | 18.1±2.1 |
| Greenland | 126,143 | 29,856 | -31±5 | -31±6 | 37±3 | - | 20.1±4.4 |
| Iceland | 11,052 | 3,722 | -98±3 | -98±3 | - | - | 9.3±0.3 |
| Svalbard | 33,932 | 10,112 | -68±16 | -65±18 | 608±158 | - | 17.0±4.6 |
| Scandinavia | 2,948 | 244 | -98±3 | -97±3 | -92±17 | - | 0.6±0.0 |
| Russian Arctic | 51,552 | 16,908 | -79±10 | -77±11 | - | - | 33.3±4.8 |
| North Asia | 2,400 | 156 | -71±5 | -97±2 | -52±8 | 220±41 | 0.3±0.0 |
| Central Europe | 2,091 | 127 | -99±0 | -100±0 | -99±0 | -77±24 | 0.3±0.0 |
| Caucasus & Middle East | 1,305 | 71 | -100±0 | -100±0 | -100±0 | -99±0 | 0.2±0.0 |
| Central Asia | 48,415 | 3,849 | -80±7 | - | -100±0 | -74±9 | 8.0±0.7 |
| South Asia West | 29,561 | 3,180 | -98±1 | - | -100±0 | -98±1 | 8.1±0.1 |
| South Asia East | 11,148 | 773 | -95±2 | - | -100±0 | -95±2 | 1.9±0.0 |
| Low Latitudes | 2,341 | 88 | -100±0 | -100±0 | -100±0 | -100±0 | 0.2±0.0 |
| Southern Andes | 29,369 | 5,701 | -98±1 | -99±1 | -74±14 | -57±12 | 14.4±0.1 |
| New Zealand | 1,161 | 65 | -88±5 | -100±0 | 71±62 | - | 0.1±0.0 |
| Global | 600,172 | 137,821 | -64±5 | -61±6 | -36±3 | -84±5 | 215.2±21.3 |

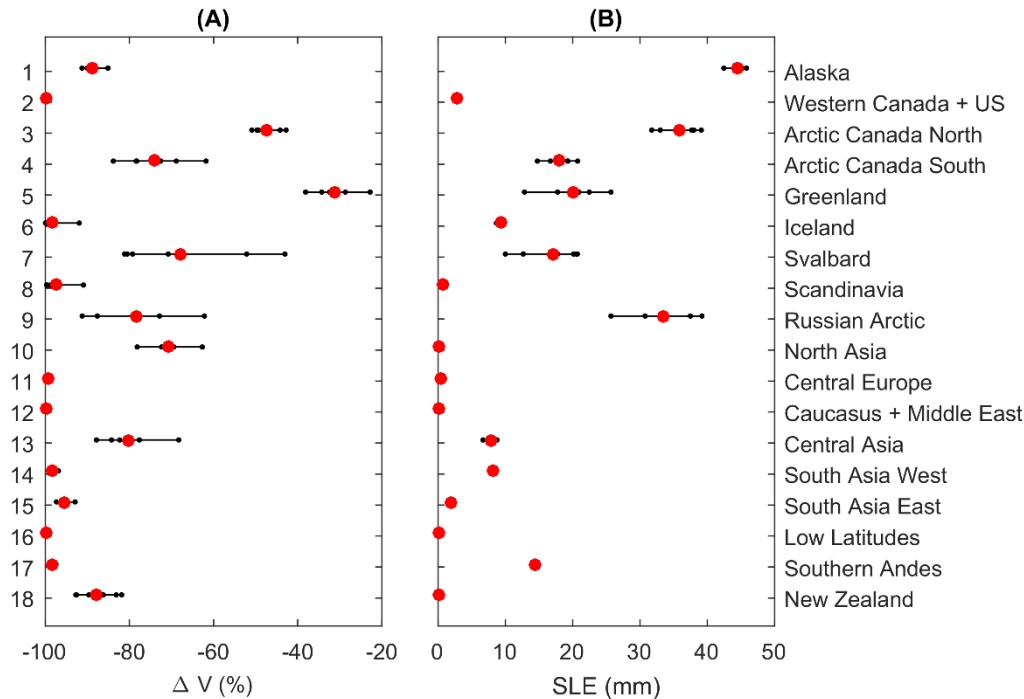

**Figure 10 (A)** Regional percentage volume losses at the end of the century (2097), relative to the initial volume and (B) volume losses expressed in sea-level equivalent contributions. The large red dots represent the multi-model mean and the small black dots are the individual HadGEM3-A model runs.

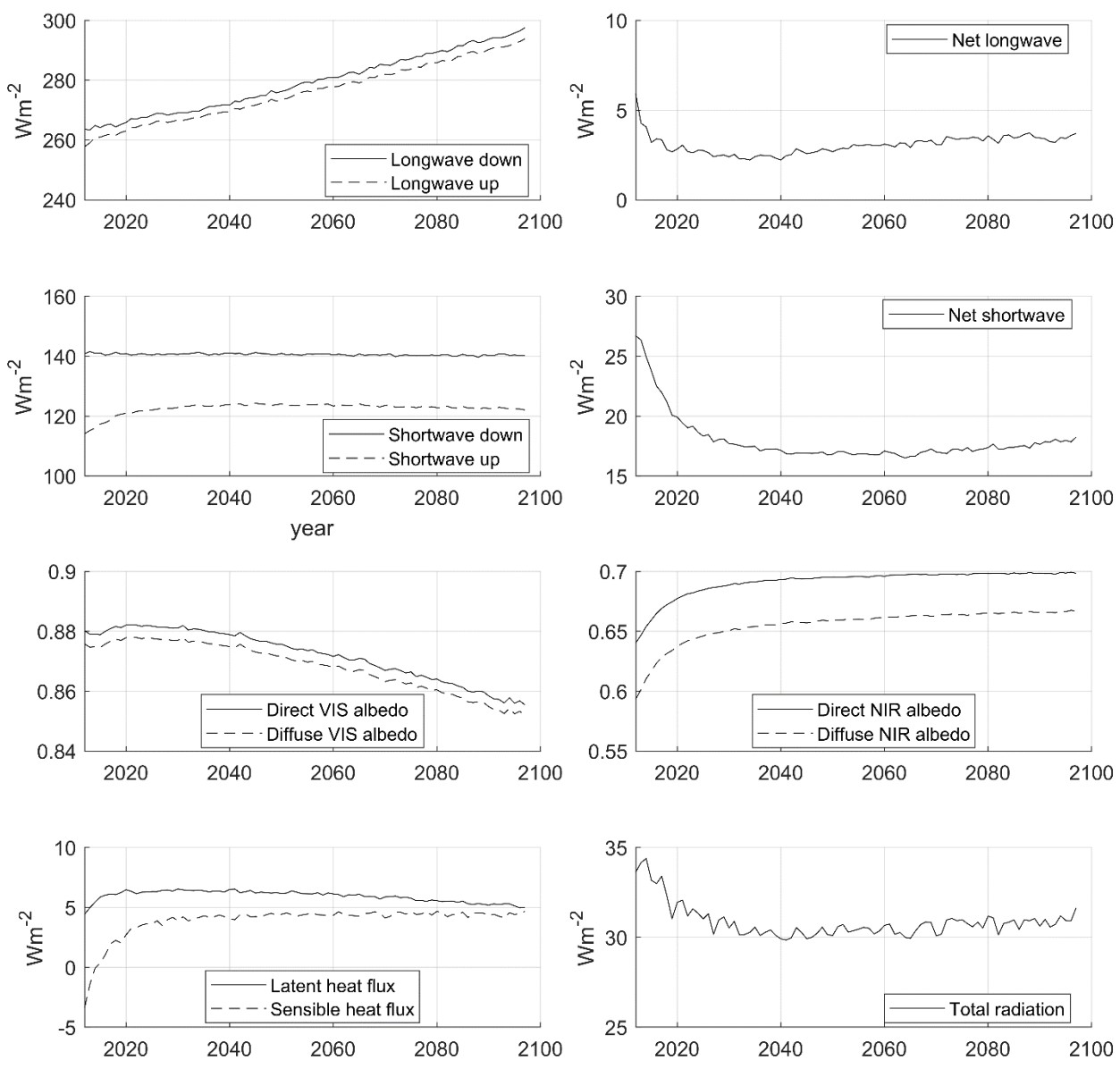

**Figure 11 Ensemble mean energy balance components averaged over all glaciated regions and all elevation bands when the model is forced with HadGEM3-A data.**

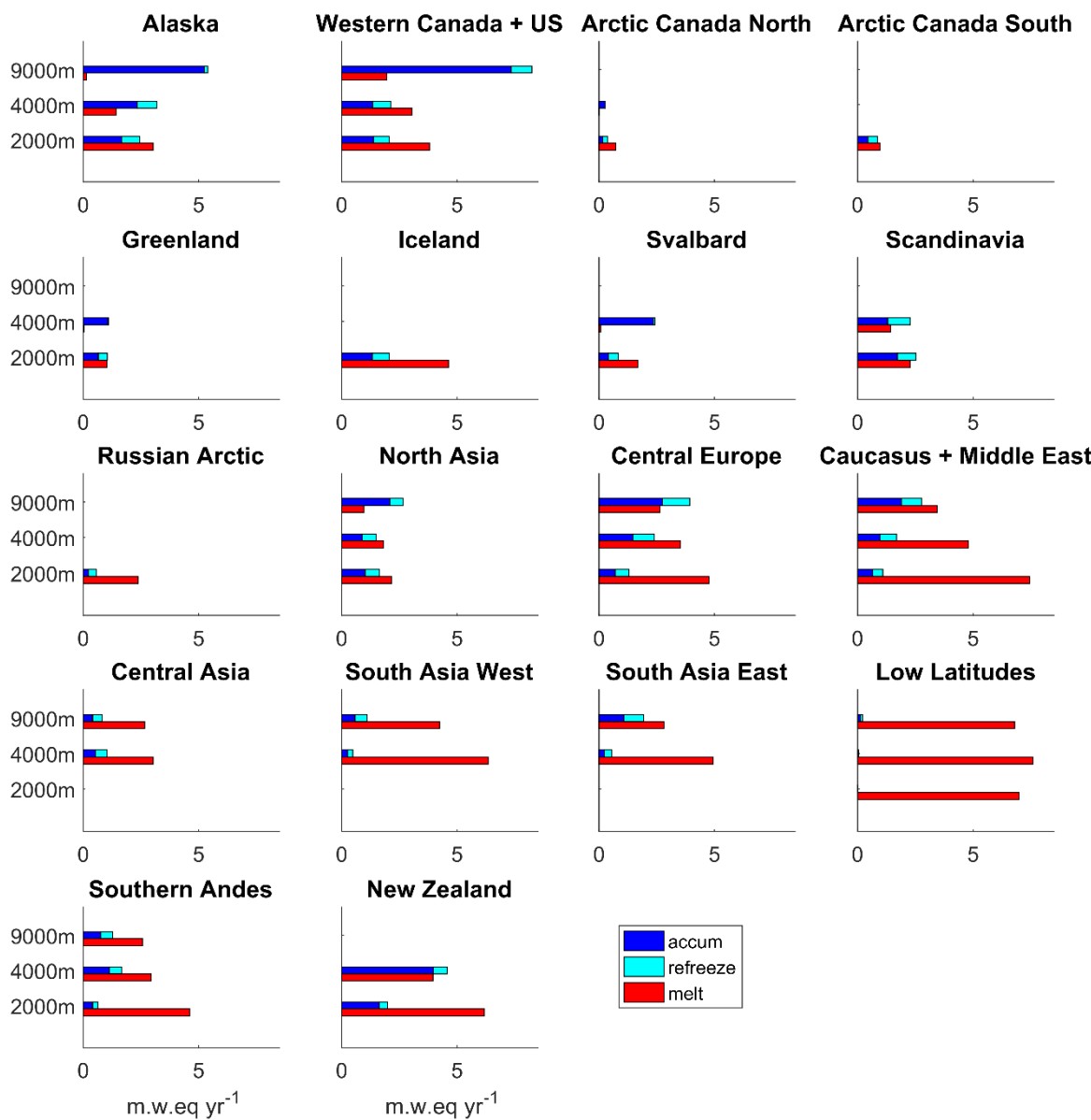

**Figure 12 Modelled annual surface mass balance components; accumulation, refreezing and melting for the period 1980-2000 for RGI6 regions. To make the figure easier to read melting is given as positive sign and sublimation is excluded because its contribution is very small. Mass balance components are averaged over low (0-2000m), medium (2250m-4000m) and high (4250-9000m) elevation ranges.**

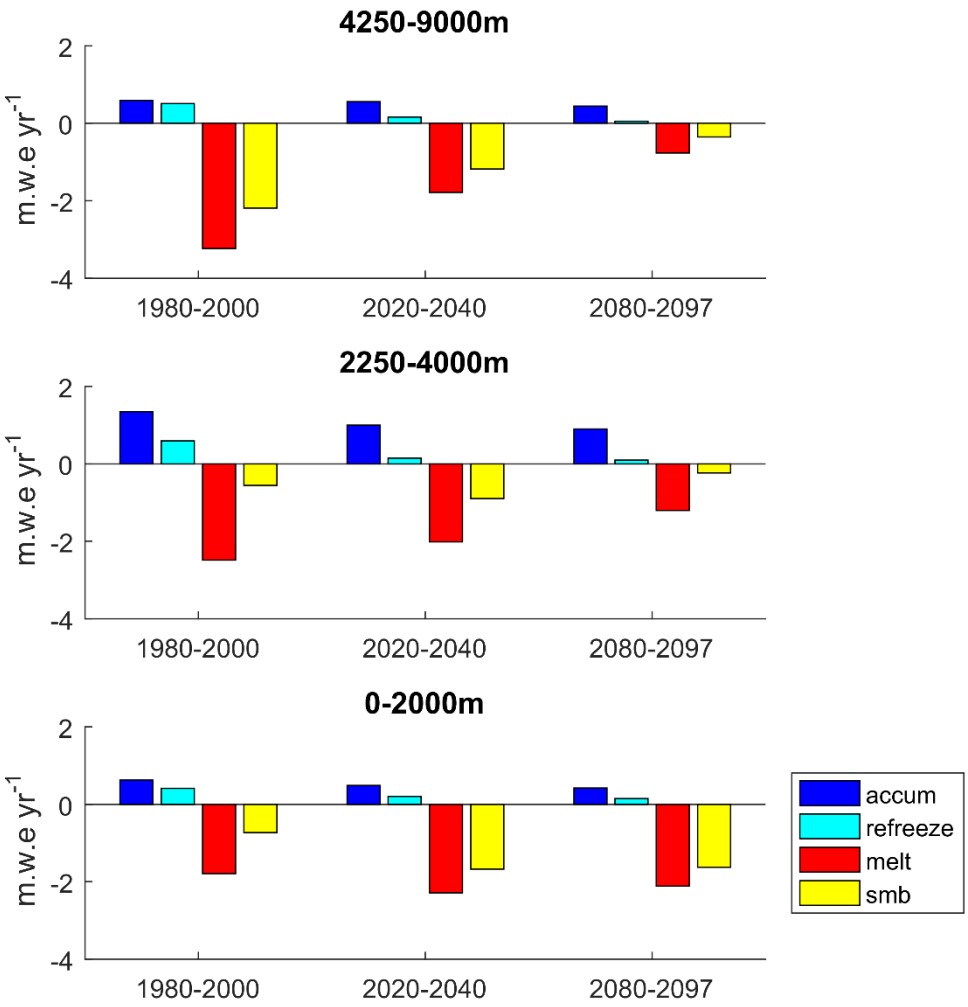

**Figure 13 Global mass balance components for three elevation ranges. The historical period is calculated using the WDFEI data and the future period is the multi-model means of all GCMs. The bars show the averages over the time periods for accumulation, refreezing, melt and mass balance rates.**

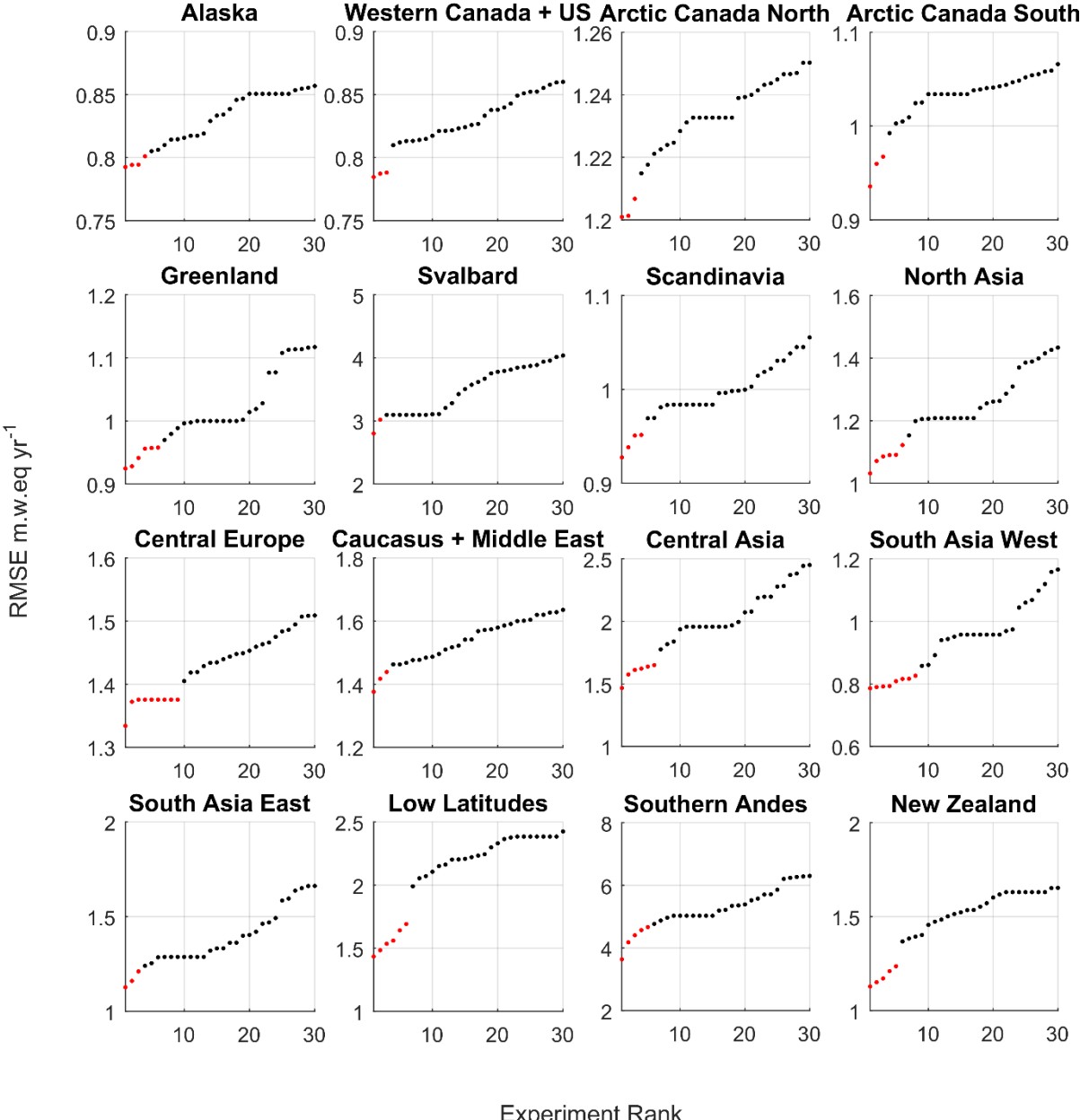

**Figure 14 Calibration experiments raked according the root mean square error between simulated and observed mass balance profiles for RGI6 regions. There are 198 experiments but only the top 30 have been plotted to make the figure easier to read. The red dots indicate experiments that perform equally well.**

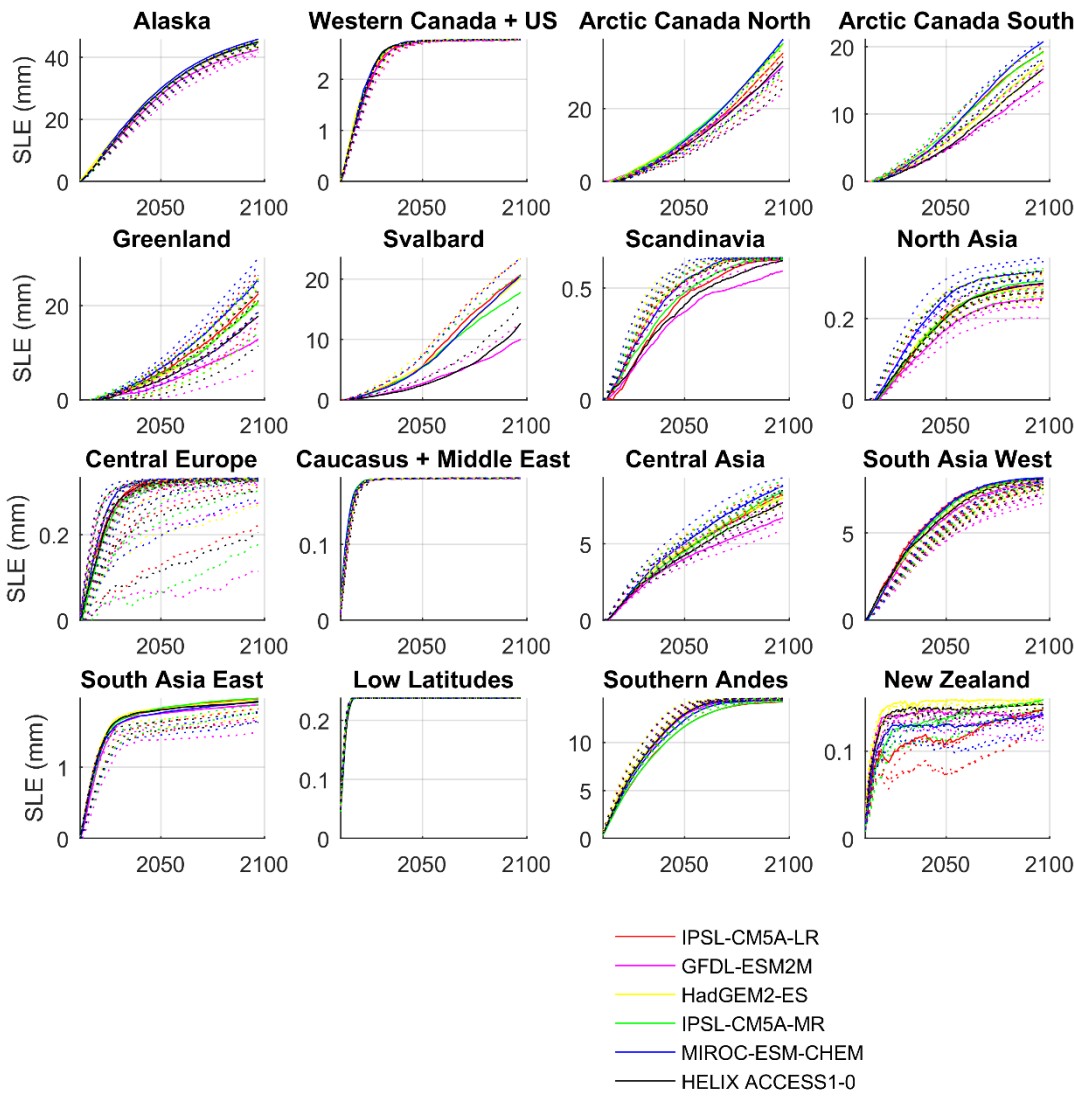

**Figure 15 Regional volume losses expressed in sea-level equivalent including parametric uncertainty in mass balance parameters. The solid lines show the volume loss for each downscaled CMIP5 GCM using the optimum parameter sets. The dashed lines are for runs which use other equally 'good' parameter sets based on the RMSE.**

**Table 7 Regional ensemble mean, minimum and maximum volume losses for 2097 in sea-level equivalent (mm) when the present-day mass balance is calibrated in different ways. Columns 1-3; mass balance is calibrated by minimising the RMSE. Columns 4-6; mass balance is calibrated using an ensemble of equally plausible RMSE values. Columns 7-9; mass balance is calibrated by minimising the RMSE, minimising the bias and maximising the correlation coefficient.**

| | Optimum parameter | | | Equally plausible RMSE | | | Extra performance metrics | | |
|---|---|---|---|---|---|---|---|---|---|
| | $SLE_{mean}$ | $SLE_{min}$ | $SLE_{max}$ | $SLE_{mean}$ | $SLE_{min}$ | $SLE_{max}$ | $SLE_{mean}$ | $SLE_{min}$ | $SLE_{max}$ |
| Alaska | 44.6 | 42.5 | 45.8 | 43.8 | 40.5 | 45.8 | 43.6 | 38.2 | 46.3 |
| Western Canada and US | 2.8 | 2.8 | 2.8 | 2.8 | 2.8 | 2.8 | 2.8 | 2.8 | 2.8 |
| Arctic Canada North | 35.8 | 31.8 | 39.1 | 32.8 | 24.3 | 39.1 | 37.2 | 22.3 | 61.8 |
| Arctic Canada South | 18.1 | 14.8 | 20.8 | 17.9 | 13.7 | 21.1 | 20.3 | 14.8 | 24.1 |
| Greenland | 20.1 | 12.9 | 25.7 | 20.4 | 6.7 | 30.2 | 23.5 | 14.0 | 31.8 |
| Iceland | 9.3 | 8.7 | 9.5 | 9.3 | 8.7 | 9.5 | 9.4 | 8.5 | 9.5 |
| Svalbard | 17.0 | 10.0 | 20.7 | 18.4 | 10.0 | 23.6 | 19.7 | 10.0 | 25.8 |
| Scandinavia | 0.6 | 0.6 | 0.6 | 0.6 | 0.6 | 0.6 | 0.6 | 0.6 | 0.6 |
| Russian Arctic | 33.3 | 25.7 | 39.2 | 33.3 | 25.7 | 39.2 | 36.6 | 25.1 | 42.8 |
| North Asia | 0.3 | 0.3 | 0.3 | 0.3 | 0.2 | 0.4 | 0.3 | 0.3 | 0.4 |
| Central Europe | 0.3 | 0.3 | 0.3 | 0.3 | 0.1 | 0.3 | 0.3 | 0.2 | 0.3 |
| Caucasus and Middle East | 0.2 | 0.2 | 0.2 | 0.2 | 0.2 | 0.2 | 0.2 | 0.2 | 0.2 |
| Central Asia | 8.0 | 6.7 | 8.8 | 8.0 | 5.9 | 9.3 | 8.1 | 5.9 | 9.5 |
| South Asia West | 8.1 | 8.0 | 8.2 | 7.8 | 6.7 | 8.2 | 7.9 | 7.1 | 8.2 |
| South Asia East | 1.9 | 1.9 | 2.0 | 1.8 | 1.5 | 2.0 | 1.8 | 1.6 | 2.0 |
| Low Latitudes | 0.2 | 0.2 | 0.2 | 0.2 | 0.2 | 0.2 | 0.2 | 0.2 | 0.2 |
| Southern Andes | 14.4 | 14.2 | 14.5 | 14.4 | 14.2 | 14.6 | 14.4 | 14.2 | 14.6 |
| New Zealand | 0.1 | 0.1 | 0.2 | 0.1 | 0.1 | 0.2 | 0.1 | 0.1 | 0.2 |
| Global | 215.2 | 181.5 | 238.9 | 212.6 | 162.2 | 247.3 | 227.1 | 166.1 | 281.1 |
| Global $SLE_{max}$ - $SLE_{min}$ | 57.3 | | | 85.1 | | | 115.0 | | |

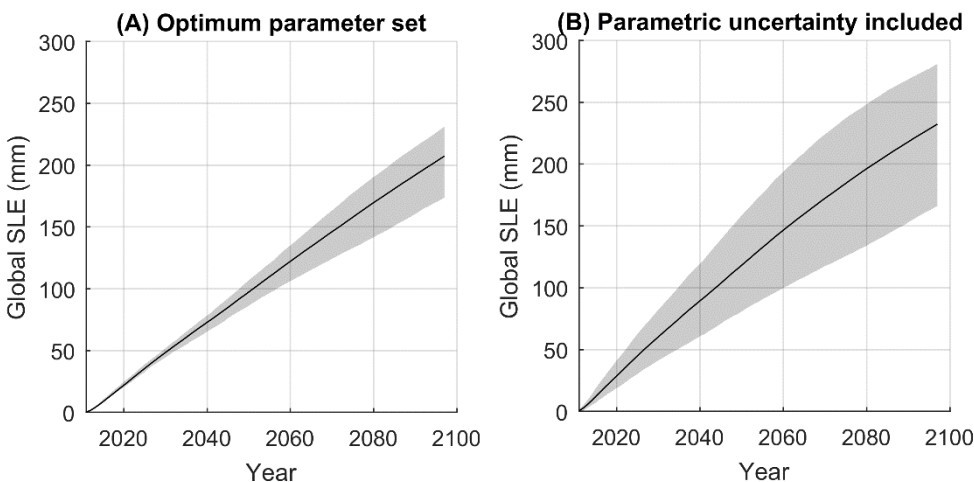

**Figure 16 Multi-model mean (black line) and ensemble spread (shaded) global volume loss in sea-level equivalent. (A) is the volume loss when optimum parameters sets are selected by minimising the RMSE and (B) is volume loss when optimum parameters sets are selected using additional performance metrics (minimising RMSE, minimising the bias and maximising the correlation coefficient).**

## 5 Code Availability

The glacier scheme is included in JULES v4.7. The source code can be downloaded by accessing the Met Office Science Repository Service (MOSRS) (requires registration): https://code.metoffice.gov.uk/ The code used for this study is in

https://code.metoffice.gov.uk/svn/jules/main/branches/dev/sarahshannon/vn4.7_va_scaling

**Acknowledgements**

This research was funded by the European Union Seventh Framework Programme FP7/2007-2013 under grant agreement n° 603864. We would like to thank the Natural Environment Research Council (NERC) for the use of the Joint Analysis System Meeting Infrastructure Needs (JASMIN) super computer cluster.

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
