# Peer review of "Global glacier volume projections under high-end climate change scenarios"

_The Cryosphere, 2018_

## Referee Comment (RC1) · B. Marzeion (Referee) · 23 Apr 2018

The model presented here is a very timely and relevant contribution to the growing group of global glacier models since it is, to my knowledge, the only global model that has been developed in the framework of a land surface model. It thus follows a concept that is very different from most other glacier models, adding important diversity and opening a potential path to coupled modeling of glaciers (i.e., allowing interactions between glaciers and ocean/atmosphere, e.g. through the changed freshwater balance), as well as a more integrated perspective on hydrologic impacts of glacier mass loss.

In principle, I therefore think the authors present a very valuable contribution. However,

there are numerous results (particularly concerning calibration and validation) that require much more in-depth analysis and discussion than presented in the manuscript. This concerns particularly (i) the substantial (and very consistently negative) bias; (ii) modelled negative winter and positive summer mass balances that are currently uncommented; (iii) the lack of a representation of the terminus altitude-mass balance feedback which might also contribute to stronger mass loss projections, and (iv) the Nash-Sutcliffe coefficients that are included in the tables, are mostly negative in the validation, but not at all explained or discussed in the text.

If these issues – outlined below in more detail – are appropriately addressed, I would be able to recommend the manuscript for publication.

Specific comments:

Abstract and Introduction: I understand the desire to express the considered CMIP5 projections in terms of temperature above pre-industrial, which has become a quite common measure with the formulation of the goals in the Paris Agreement. However, most readers will be familiar with the RCP scenarios. I was wondering until page 10 whether a "mixed" scenario was used, e.g. selecting CMIP runs based on warming. A quick statement in the abstract and the introduction that RCP8.5 is used will make things a lot clearer.

P2 L5: In some regions this is true, but since glacier water release is not causally related to demand for water, this sentence should be reformulated.

P2 L7: Most of the authors are native speakers, which makes me doubt myself – but I learned that when used as a compound adjective, sea level should be hyphenated (as in "sea-level rise")?

P3 L14ff: It would be nice if the authors could comment on how strongly this limitation affects the usability of JULES. I.e., is it realistic that the new glacier scheme would be used in a default setup of JULES, or would the limitations inflicted on the other surface

classes be too strong?

P4 L5f: If I understand correctly, this implies that the negative feedback between terminus elevation and mass balance is missing, and that the only way for a melting glacier to reach equilibrium with climate is by melting completely (similar to Slangen & van de Wal, 2011, https://doi.org/10.5194/tc-5-673-2011). This is a major limitation that should be discussed in greater depth. E.g., Marzeion et al. (2014, https://doi.org/10.5194/tc-8-59-2014) find that depending on scenario and accumulated mass loss, this may contribute a few tens of mm SLE (their Figs. 9 to 11). The differences in Tab. S3 could be plausibly explained by this alone. I also think that the discussion of the lack of a parameterization of ice dynamics (end of Sec. 4.4) is flawed with respect to this feedback.

P4 L19ff: Units of the constants are partly missing or wrong.

P4 L28: Goff-Gratch (typo), and Landolt-Bordstein 1987 is incomplete in the references, and probably should be Landolt-Börnstein.

P5 L25: It would be better to say that the energy balance calculated in JULES includes the sensible heat flux, since the snow melt is not a direct (or separate) consequence of the sensible heat flux alone.

Eq. 13: I think this equation is a bit problematic if justified by katabatic winds, since the katabatic winds should not be expected to be proportionate to the large-scale wind field.

P6 L25: "at the beginning" (typo)

P9 L 11: In Marzeion et al. (2012), it is 3 %/100 m.

Tab. 2: Since the Nash-Sutcliffe efficiency coefficient is included, it should be discussed in the text. It would be particularly good to address the reasons for the numerous negative values – are they caused by bias, too great/small variance, etc.?

Tab. 3: Please add global mean values.

[Figure]

P10 L1f: The tropical glaciers are really small; there are probably numerous more likely explanations for a warm bias than glaciers lacking in the model.

Figs. 3 and 4, and discussion around them (also P10 L14f): I'm not convinced the Pyrenean glacier is to blame for the low correlation. How much does the correlation change if you exclude it? I'm also wondering why the point cloud for Central Europe in Fig. 3 looks different from the one in Fig. 4?

Tab. 4: Please add global mean values.

Tab. 4: Again, it is necessary to discuss the negative NS-values. There is only one positive value in the table, which indicates that only for summer mass balance in Scandinavia, the model has better predictive skill than taking the mean of the observations. Also, given that all the biases are negative (with the exception of one that is close to zero), the implications for the projections need to be addressed. E.g., if the bias was compensated for in the projections: how would that change the results? Could the differences to previously published projections be explained by the global mean bias?

Fig. 5: the RMSE mentioned in the caption is missing in the panels.

Fig. 6: It is surprising that the model produces many substantially negative winter mass balances (and not quite as many positive summer mass balances). This behavior should be looked into and discussed in the text.

Caption of Fig. 6: "number of glaciers" – isn't that the number of grid boxes?

P10 L23: "sensitivities" (typo)

P11 L12: "Arctic" (typo)

Fig S7: It is hard to see anything here; perhaps just leave out the East African and Indonesian glaciers (which is sad for sentimental reasons, but I think they are mostly irrelevant for sea level and water availability).

P11 L16: Figs. S1 to S7.
Sec. 4.3: It is good to see that the model results appear to be robust; on the other hand, this may indicate the negative bias may be hard to overcome. In the calibration, minimizing the RMSE was used for identifying the best parameter set(s). Another way of looking into "parametric" uncertainty (in a wider sense) would be to minimize the bias, or to maximize the correlation or the NS coefficient. These experiments might give valuable insights into the causes of the sometimes problematic model performance, which needs to be better explained.

P13 L2: "equivalent" (typo)

P14 L27: Reduction in glacier mass, not necessarily in mass balance.

P15 L18: "periphery" (typo)

---

## Referee Comment (RC2) · Anonymous Referee #2 · 8 Jun 2018

**1   General comments**

The proposed manuscript presents new estimates of future sea-level rise obtained with a new model of global glacier evolution. The model is quite original because it is integrated in a land surface model. Also, the mass-balance is computed from the surface energy balance, which could lead to different and more contrasted results than classical "temperature-index" mass-balance models. For these reasons, it is a useful addition to the literature.

That said, I think that the paper needs serious revisions before being considered for publication. I have some major concerns listed below. My recommendation would be to:

[Figure]

- use additional datasets for model validation

- put less focus on the validation with point mass-balance data

- spend more time on the energy balance instead

**1.1 Model set-up**

I found it difficult to understand some aspects of the model set-up, and I believe the text could be meliorated by reorganizing some of its sections. In particular:

- the decription of the snowpack/ice meld model is scattered around Section 2/ Intro, and page 7.

- the description of how and where glaciers lose mass is unclear to me (and to the other reviewer as well)

The manuscript would also benefit from a clear discussion and listing of the particularites (strengths/weaknesses) of this model in comparison to previous modelling attempts. This could be a way to highlight the strengths of your model (see "Energy balance" below).

**1.2 Calibration/validation**

This was already mentioned by reviewer 1: some model results are poorly explained or not discussed at all.

Validating and calibrating against in-situ elevation dependant MB data is really hard. Point mass-balance observations reflect a number of local and glacier specific factors, and there is a risk that these local factors affect parameter sets chosen for entire RGI

regions (this is particularly true for regions with few observations). I am not asking to change you procedure as this will likely represent too much work, but I'm allowing myself to provide a couple of suggestions:

- I would personnally recommend to use one global set of parameters instead of the regional ones, unless there is a compelling reason not to do so (a good candidate would be a physical explanation of the regional parameter sets). The resulting parameter sets would be more robust and would allow statistical scrutinizing using cross-validation (or more advanced) methods. This is even more relevant for physically based approaches like yours.

- Having some kind of independant validation would consideralby strengthen the readers' confidence in your results. Albeit not without their own problems, regional geodetic mass-balance estimates could be useful to at least get a quantitative estimate of the model performance at the regional scale.

**1.3 Energy balance**

The real strength of the model is its use of an energy balance model instead of a temperature index model as the majority of the other global models. Whether or not this increase in complexity is actually leading to better results remains (and will remain) a controversial topic, but this study should make use of this novel approach. In particular, I would find it very interesting to see plots of the energy balance components as a function of altitude, and how these energy balance components change in the future. This is interesting because energy balance models are likely to be less sensitive to temperature change and incorporate other processes instead.

I would also welcome new analyses of not only the total volume change, but the volume changes per elevation band.

**1.4 Code availability**

Please add a statement about where and how people can access your code and that of JULES.

**2 Specific comments**

**P3 L15** why is this limitation about the partial coverage necessary? In view of the objective of developping a fully coupled model, it would be good to overcome this limitation one day.

**P3 L23** $0.5°$ and 46 elevation bands: What motivated the choice of these resolutions? Can this be changed at whish?

**P4 L5** Snowpack: do I get this right that there is no distinction between ice and snow in the snowpack model? What are typical values for ice density in the model? How much time does it need to transform snow to ice?

**P4 L10** I must admit that I dislike the current approach to temperature downscaling, which in my opinion is an unhealthy mix of thermodynamics and tuning. I'm not asking to change it, but the fact that the lapse-rate is tuned in non-saturated conditions (the rate changes quite a lot according to table 2) but not in saturated conditions is likely to create odd non-linearities in the model's response to certain forcings.

**P5 L1** *"we only tune the dry adiabatic lapse-rate"*. Here and throughout the rest of the manuscript: do not use the term "dry adiabatic lapse-rate". The dry adiabatic lapse-rate **is** the dry adiabatic lapse-rate and is 9.8K per 1000m. What you are tuning though is the near-surface temperature lapse-rate, which might vary according to surface conditions and moisture content.

**P5 L22** wet-bulb temperature is a much better indicator for solid precipitation than regular dry-bulb temperature. This could mitigate parts of the dramatic changes in snowfall projected by your scenarios.

**P5 L27** here and at some other places in the manuscript, the missing katabatic flow is given a high prominence in the list of missing processes to be addressed. This might be the case (also not the most prominent on my list), but the proposed solution (scaling the synoptic wind field) does not sound really physical to me. The katabatic flow is notoriously decoupled from the synoptic conditions and is likely to be strongests when the synoptic flow is weak. What the scaling of the modelled wind achieves, though, is an increase of the turbulent fluxes: I would be very interested in seing more discussion about why this is necessary (see major point 1.3 above).

**P7 L1-2** If I get this right, the glacier tiles are able to lose mass per elevation band, right? The information is scattered in the manuscript (P7 L10, P7 L24 . . . ) and should be clarified much earlier to avoid confusion (see also the comment from reviewer 1 who seems to have understood something different than me).

**P7 L10** I don't understand this part. Can you be more specific about how glaciers grow/shrink and lose/gain mass in the model?

**P7 L25** What happens at the end of the initialisation? Setting 500 m of ice everywhere is maybe ok for a spin-up, but what happens next, or at the start of the 2011-2100 simulation?

**P9 -L27** I don't understand the statement "with the notable exception of the low latitude and Central European regions where melting is over estimated". According to Table 3 and the BIAS measure, melt is over-estimated in 9 regions with Svalbard, Southern Andes and New Zealand striking out with more than 1 m negative bias.

Central Europe even has a positive bias. A quick look through the table indicates a general negative bias.

**P10 L3 and Figure 4** : the explanation about the Maladeta is irrelevant. Figure 4 shows that there are other pink dots around the Maladeta starts, and there is no need for a case by case explanation here.

**Section 3.2** This does not represent an independant validation because the same data was used for calibration also. What is striking in Table 4 is that all regions now have a significant negative bias both in winter and summer. How can this be explained?

**P10 L21** I would like to see more explainations about the "downscaling" procedure. If only SST and Sea-ice are used, this sounds a lot more like a full atmosphere GCM simulation to me than a "downscaling" of a GCM product. In particular, what happens to the land-surface components in HadGEM3? What is actually left from the original GCM signal after "downscaling"?

**P11 L2** remove "which is suitable for capturing precipitation variability over complex topography"

**Section 4.3** Parametric uncertainty analysis is one aspect of parameter uncertainty. A further uncertainty would be revealed by doing data-denial experiments (cross-validation) and assessing the sensitivity of your calibration procedure to those.

**Section 4.4 Comparison with other studies** I would welcome future studies based on this model to use the same forcing data and same conventions as other global glacier models where possible in order to facilitate model intercomparisons.

**P13 L25** My understanding is that your study is using the global volume estimates provided by Matthias Huss.

**Supplementary material**  I suggest to invert the color scale: red seems more intuitive for mass loss.

---

## Author Comment (AC1) · 7 Sep 2018

*We would like to thank the reviewer very much for their feedback. In response to your comments we made the following changes to the manuscript: Added a discussion about the limitation of having a fixed terminus elevation and cited Marzeion et al 2014. A discussion on the negative Nash-Sutcliffe numbers and possible explanation for the model bias in the seasonal bass balance validation. We explored the impact of bias correcting the present-day annual mass balance, on future volume loss projections. We also re-ran the future simulations when the model is calibrated by minimising the bias and RMSE and maximising the correlation coefficient. Please see our detailed relies to your comments below.*

**Reviewer comment: The model presented here is a very timely and relevant contribution to the growing group of global glacier models since it is, to my knowledge, the only global model that has been developed in the framework of a land surface model. It thus follows a concept that is very different from most other glacier models, adding important diversity and opening a potential path to coupled modelling of glaciers (i.e., allowing interactions between glaciers and ocean/atmosphere, e.g. through the changed freshwater balance), as well as a more integrated perspective on hydrologic impacts of glacier mass loss.**

**In principle, I therefore think the authors present a very valuable contribution. However, there are numerous results (particularly concerning calibration and validation) that require much more in-depth analysis and discussion than presented in the manuscript.**

**This concerns particularly (i) the substantial (and very consistently negative) bias; (ii) modelled negative winter and positive summer mass balances that are currently uncommented; (iii) the lack of a representation of the terminus altitude-mass balance feedback which might also contribute to stronger mass loss projections, and (iv) the Nash-Sutcliffe coefficients that are included in the tables, are mostly negative in the validation, but not at all explained or discussed in the text. If these issues – outlined below in more detail – are appropriately addressed, I would be able to recommend the manuscript for publication.**

**Specific comments:**
**Abstract and Introduction: I understand the desire to express the considered CMIP5 projections in terms of temperature above pre-industrial, which has become a quite common measure with the formulation of the goals in the Paris Agreement. However, most readers will be familiar with the RCP scenarios. I was wondering until page 10 whether a "mixed" scenario was used, e.g. selecting CMIP runs based on warming. A quick statement in the abstract and the introduction that RCP8.5 is used will make things a lot clearer.**

*Changes to manuscript: In the abstract we included: The CMIP5 models use the RCP8.5 climate change scenario and were selected on the criteria of passing $2^oC$ global average warming during this century.*

*Changes to manuscript: In the introduction we included: "The CMIP5 models use the Representative Concentration Pathways (RCP) RCP8.5 climate change scenario for high greenhouse gas emissions. "*

**Reviewer comment: P2 L5: In some regions this is true, but since glacier water release is not causally related to demand for water, this sentence should be reformulated.**

*Reply: We removed the part "when demand for water is high"*

**Reviewer comment: P2 L7: Most of the authors are native speakers, which makes me doubt myself – but I learned that when used as a compound adjective, sea level should be hyphenated (as in "sea-level rise")?**

*Changes to manuscript: We have hyphenated instances of the adjective "sea-level rise"*

**Reviewer comment: P3 L14ff: It would be nice if the authors could comment on how strongly this limitation affects the usability of JULES. I.e., is it realistic that the new glacier scheme would be used in a default setup of JULES, or would the limitations inflicted on the other surface classes be too strong?**

*Reply: It is feasible to use the glacier scheme in the default configuration of JULES with a caveat. The model is set up so that glacier surfaces cannot share a gridbox with other surface types. This means that running the model with glaciers switched on, should not affect the other surface types. The caveat is this: in order to get sufficient accumulation in the mass balance profile, we had to lapse rate correct the precipitation. This is a standard procedure in mass balance modelling, but the consequence is that gridbox mean precipitation over glacier gridboxes is not conserved. We tested scaling the precipitation, while conserving the gridbox mean i.e. reducing the precipitation near the surface and increasing it at height, but this did not yield enough precipitation to get a good agreement the mass balance profiles. If you wanted to run JULES with the glacier model switched on, then it is worth bearing in mind that water will no longer be conserved. If the model is being used to simulate river discharge in glaciated catchments, then the precipitation lapse rate could be used as a parameter to calibrate the discharge.*

*Changes to manuscript: We have added this point to the model description section 2.2.3 in case the reader also has the same question.*

**P4 L5f: If I understand correctly, this implies that the negative feedback between terminus elevation and mass balance is missing, and that the only way for a melting glacier to reach equilibrium with climate is by melting completely (similar to Slangen & van de Wal, 2011, https://doi.org/10.5194/tc-5-673-2011). This is a major limitation that should be discussed in greater depth. E.g., Marzeion et al. (2014, https://doi.org/10.5194/tc-8-59-2014) find that depending on scenario and accumulated mass loss, this may contribute a few tens of mm SLE (their Figs. 9 to 11). The differences in Tab. S3 could be**
**plausibly explained by this alone.**

**I also think that the discussion of the lack of a parameterization of ice dynamics (end of Sec. 4.4) is flawed with respect to this feedback.**

*Reply: Yes, you understand correctly here. The negative feedback between terminus elevation and mass balance is missing and the only way for a melting glacier to reach equilibrium with climate is by melting completely.*

*Changes to manuscript: We have added the following text to the results section.*

*"Another explanation why our model predicts more volume loss than Radic et al (2014) and Huss and Hock (2015) is because there is no retreat of the glacier terminus represented in the model. The only way for glaciers to reach equilibrium with climate is by melting completely. A study by Marzeion et al (2014) showed that models predict more mass loss when the terminus elevation is fixed, than when it*

*is allowed to vary. This is because when the terminus retreats, the area available for melting is reduced, leading to less mass loss. Marzeion et al (2014) found that neglecting terminus elevation changes resulted in an extra few tens of mm SLE depending on RCP scenario."*

*Changes to manuscript: We removed the flawed section regarding the lack of ice dynamics (end of Sec. 4.4)*

**Reviewer comment: P4 L19ff: Units of the constants are partly missing or wrong.**
*Changes to manuscript: Units corrected and added.*

**Reviewer comment: P4 L28: Goff-Gratch (typo), and Landolt-Bordstein 1987 is incomplete in the references, and probably should be Landolt-Börnstein.**
*Changes to manuscript: Corrected*

**Reviewer comment: P5 L25: It would be better to say that the energy balance calculated in JULES includes the sensible heat flux, since the snow melt is not a direct (or separate) consequence of the sensible heat flux alone.**
*Changes to manuscript: Corrected –Instead we say "A component of the energy for melting the snowpack in JULES comes from the sensible heat flux"*

**Reviewer comment: Eq. 13: I think this equation is a bit problematic if justified by katabatic winds, since the katabatic winds should not be expected to be proportionate to the large-scale wind field.**

*Reply: We agree. Reviewer #2 also identified this as an unrealistic way to represent katabatic winds (Please see our response to this). We scaled the wind speed to increase the sensible heat flux compared to observations. Although our approach is not a physically realistic, we thought it was better to include a simple way to increase wind speed over glacier gridboxes, than excluding this.*

**Reviewer comment: P6 L25: "at the beginning" (typo)**
                            **P9 L 11: In Marzeion et al. (2012), it is 3 %/100 m.**

*Changes to manuscript: Corrected*

**Reviewer comment: Tab. 2: Since the Nash-Sutcliffe efficiency coefficient is included, it should be discussed in the text. It would be particularly good to address the reasons for the numerous negative values – are they caused by bias, too great/small variance, etc.?**

*Reply: We calculate the Nash-Sutcliffe efficiency coefficient equation using the following equation*

$$NS = 1 - \frac{\sum_{i=1}^{n}\left(Y_i^{obs} - Y_i^{model}\right)}{\sum_{i=1}^{n}\left(Y_i^{obs} - Y_i^{mean\,obs}\right)}$$

*where the numerator is the mean square error (or the bias) and the dominator is the variance. The negative numbers arise because the bias is greater than the variance of the observations. The negative bias indicates that melting is overestimated in the summer and accumulation is underestimated during the winter (Table 4).*

*Changes to manuscript: We added the following text in the model validation section 3.2 to explain the possible reasons for the bias.*

*"For all regions, except Scandinavia in the summer, negative Nash-Sutcliff numbers are calculated for winter and summer elevation-dependent mass balance (Table 4). The negative numbers arise because the bias in the model is larger than the variance of the observations. There are negative biases for nearly all regions implying that melting is overestimated in the summer and accumulation is underestimated in the winter.*
*Some, but not all, of the bias is due to the partitioning of rain and snow based on an air temperature threshold of 0°C. The 0°C threshold is likely too low, resulting in an underestimate of snowfall. When precipitation falls as rain or snow it adds liquid water or ice to the snowpack. The specific heat capacity of the snowpack is a function of the liquid water ($W_k$) and ice content ($I_k$) in each layer (k)*

$$C_k = I_k C_{ice} + W_k C_{water} \qquad\qquad (17)$$

*where $C_{ice}$ = 2100 $JK^{-1}kg^{-1}$ and $C_{water}$ = 4100 $JK^{-1}kg^{-1}$.*
*The liquid water content is limited by the available pore space in the snowpack, therefore changes in the ice content control the overall heat capacity. The underestimate in the ice content reduces the heat capacity which causes more melting than observed.*
*Other modelling studies have used higher air temperature thresholds; 1.5°C (Huss and Hock 2015, Giesen and Oerlemans 2012), 2°C (Hirabayashi et al 2010) and 3°C (Marzeion et al 2012). An improved approach would use the wet-bulb temperature to partition rain and snow which would include the effects of humidity on temperature. Alternatively, a spatially varying threshold based on precipitation observations could be used. Jennings et al (2018) showed by analysing precipitation observations, that the temperature threshold varies spatially. Jennings et al (2018) showed by analysing precipitation observations, that the temperature threshold varies spatially and generally higher for continental climates than maritime climates.*

*Increasing the temperature threshold only reduces the bias slightly, therefore another explanation is that the precipitation in the WFDEI data is too low. Although we have included the variation in precipitation with height, if the gridbox mean precipitation is too low then snowfall on the elevated tiles will be underestimated. We did not bias correct the precipitation before applying the lapse rate correction unlike other studies do (Marzeion et al. 2012, Huss and Hock 2015). The quality of the WFDEI precipitation maybe poor because the data is constrained by rain gauge observations which are sparse in high mountains regions and often biased towards low elevation levels. Even when observations are available snowfall at higher altitudes is often difficult to accurately measure and susceptible to undercatch by 20–50% (Rasmussen et al. 2012). The biases listed in Table 4 are larger in the summer than in the winter. It is likely that the simple albedo scheme, which relates albedo to the density of the snowpack surface, does not perform particularly well in the ablation zone. "*

**Reviewer comment: Tab. 3: Please add global mean values.**
*Changes to manuscript: Global values are added. Also, we noticed the number of observations listed in column 5 was incorrect, so we updated this.*

**Reviewer comment: P10 L1f: The tropical glaciers are really small; there are probably numerous more likely explanations for a warm bias than glaciers lacking in the model.**

**Figs. 3 and 4, and discussion around them (also P10 L14f): I'm not convinced the Pyrenean glacier is to blame for the low correlation. How much does the correlation change if you exclude it? I'm also wondering why the point cloud for Central Europe in Fig. 3 looks different from the one in Fig. 4?**

*Reply: There was an error in Figure 4 which we have corrected, and the point cloud now matches the data for Central Europe in Figure 3.*

*Changes to manuscript: We added the following "In Central Europe some of the poor correlation with observations is caused by the Maladeta glacier in the Pyrenees (Fig. 4) which is a small glacier with an area of 0.52 km² WGMS (2017). When this glacier is excluded from the analysis the correlation coefficient increases from 0.26 to 0.35 and the RMSE decreases from 1.99 to 1.73 meters of water equivalent per year."*

**Reviewer comment: Tab. 4: Please add global mean values.**
*Changes to manuscript: Values are added*

**Reviewer comment: Tab. 4: Again, it is necessary to discuss the negative NS-values. There is only one positive value in the table, which indicates that only for summer mass balance in Scandinavia, the model has better predictive skill than taking the mean of the observations. Also, given that all the biases are negative (with the exception of one that is close to zero), the implications for the projections need to be addressed. E.g., if the bias was compensated for in the projections: how would that change the results? Could the differences to previously published projections be explained by the global mean bias?**

*Reply: Please see above for a discussion on the negative NS-values.*

*We explored the impact of correcting the bias in the annual mean mass balance on the volume projections. The differences to previously published projections cannot be explained by the bias alone, but it does account for why we have larger volume losses in the Southern Andes, where the bias was particularly large.*

*Changes to manuscript: The text below has been added to section 4.5 (Comparison with other studies).*

*"We estimate the end of century global sea-level contribution, excluding Antarctic glaciers, to be 215 ± 20mm which is higher than 188mm (Radic et al. 2014) and 136±23mm (Huss and Hock 2015) caused mainly by greater contributions from Alaska, Southern Andes and the Russian Arctic. These three regions are discussed in turn.*

*For the Southern Andes our estimates are approximately double (14.4mm) that of the other studies (5.8mm (Huss and Hock 2015), 8.5mm (Radic et al. 2014)). This region has the largest negative bias in the calibrated present-day mass balance (-2.87 m.w.eq.yr⁻¹ see Table 3). To explore the effects of correcting the calibration bias on the ice volume projections, we subtract the bias values listed in Table 3 from the future annual mass balance rates. Each gridbox is assumed to have the same regional mass balance bias. The bias corrected volume losses are listed in Table S3 in the supplementary material. For the Southern Andes, the volume losses are much closer to the other studies (7.6mm) when the bias is corrected. The impact is less for the other regions where the biases are smaller. For the Russian Arctic our volume losses are higher than the other studies but that should be interpreted with caution because there were no observations available in this region to get a tuned parameter set (global mean*

*parameters where used instead). In Alaska the bias in annual mass balance is small (0.06 m.w.eq.yr$^{-1}$) so correcting the bias has little effect on the volume loss projection for this region. Applying the bias correction increases the global volume loss from 215 ± 20mm to 222.5±20.1mm, therefore the difference between our model and the other studies cannot be explained by the bias in the calibration."*

**Reviewer comment: Fig. 5: the RMSE mentioned in the caption is missing in the panels.**

*Changes to manuscript: The caption has been modified so it no longer says the RMSE error values are shown on the plots. The RMSE values are listed in Table 4.*

**Reviewer comment: Fig. 6: It is surprising that the model produces many substantially negative winter mass balances (and not quite as many positive summer mass balances). This behaviour should be looked into and discussed in the text.**

*Reply: This suggests there is a warm bias in the winter but no equivalent cold bias in the summer. We think the presence of a winter warm bias in the model should not necessarily mean there is an equivalent cold bias in the summer.*

**Reviewer comment: Caption of Fig. 6: "number of glaciers" – isn't that the number of grid boxes?**

*Reply: This is the number of glaciers for which there are observations.*

**Reviewer comment:**
**P10 L23: "sensitivities" (typo)**
**P11 L12: "Arctic" (typo)**

*Changes to manuscript: Corrected*

**Reviewer comment:**
**Fig S7: It is hard to see anything here; perhaps just leave out the East African and Indonesian glaciers (which is sad for sentimental reasons, but I think they are mostly irrelevant for sea level and water availability).**
*Changes to manuscript: The figure is modified to only show South and Central America.*

**Reviewer comment: P11 L16:**
*Changes to manuscript: Figs. S1 to S7.*

**Reviewer comment: Sec. 4.3: It is good to see that the model results appear to be robust; on the other hand, this may indicate the negative bias may be hard to overcome. In the calibration, minimizing the RMSE was used for identifying the best parameter set(s). Another way of looking into "parametric" uncertainty (in a wider sense) would be to minimize the bias, or to maximize the correlation or the NS coefficient. These experiments might give valuable insights into the causes of the sometimes problematic model performance, which needs to be better explained.**

*Reply: We investigated the effect of using multiple performance metrics on the calibrated parameters and glacier volume projections*

*"Another way to explore the uncertainty in the volume projections caused the calibration procedure, is to use different performance metrics to identify best parameters sets. In addition to using RMSE, we calculate best parameter sets by (1) minimising the absolute value of the bias and (2) maximizing the correlation coefficient. The best regional parameter sets are different depending on the choice of performance metric used (See Tables S2 and S3 in the Supplementary material). For twelve regions, minimising the bias results in higher precipitation lapse rates, than when RMSE values are used to select parameters. This suggests the bias in many regions is caused by underestimating the precipitation lapse rates. As discussed above, this could be due to the fact the gridbox mean WFDEI precipitation was not bias corrected. Glacier volume projections are generated by repeating the simulations using these two additional performance metrics to identify best parameter sets. The uncertainty in the global volume loss when the extra performance metrics are used, is approximately double the uncertainty arising from the different climate forcings (Fig. 16, Table 7). When extra performance metrics are used, the upper bound volume loss increases to 281.1 mm sea-level equivalent by the end of the century."*

*Changes to manuscript: We changed Figure 16 to show the large spread in the global volume loss when extra performance metrics are used in the calibration.*

*Extra columns are added to Table 7 to list the regional volume losses when we minimise the bias and RMSE and maximise the correlation coefficient.*

*Tables are added to the Supplementary Information listing the best parameter selected by minimising the bias (Table S2) and maximining the correlation coefficient (Table S3).*

**Reviewer comment:**

**P13 L2: "equivalent" (typo)**
**P14 L27: Reduction in glacier mass, not necessarily in mass balance.**
**P15 L18: "periphery" (typo)**

*Changes to manuscript: Typos corrected*

Giesen, R. H. & J. Oerlemans (2012) Calibration of a surface mass balance model for global-scale applications. *Cryosphere,* 6**,** 1463-1481.

Hirabayashi, Y., P. Doll & S. Kanae (2010) Global-scale modeling of glacier mass balances for water resources assessments: Glacier mass changes between 1948 and 2006. *Journal of Hydrology,* 390**,** 245-256.

Huss, M. & R. Hock (2015) A new model for global glacier change and sea-level rise. *Frontiers in Earth Science,* 3.

Jennings, K. S., T. S. Winchell, B. Livneh & N. P. Molotch (2018) Spatial variation of the rain–snow temperature threshold across the Northern Hemisphere. *Nature Communications,* 9.

Marzeion, B., A. H. Jarosch & M. Hofer (2012) Past and future sea-level change from the surface mass balance of glaciers. *The Cryosphere Discuss.,* 6**,** 3177-3241.

---

## Author Comment (AC2) · 7 Sep 2018

*We thank the reviewer very much for their feedback on the manuscript. These are the main changes we made in response to your comments; Added a validation of energy balance components to show the reason we scaled the wind speed was to increase the sensible heat flux. Added a discussion of the key strengths and weaknesses of the model (including refence to previous work to implement a glacier scheme into the REMO model). Added a discussion on the drawbacks of our calibration approach. Added a section with figures on mass balance components to show how these vary with height and how these may change in the future. Included percentage volume change projections for elevation levels. Added a discussion on the reasons for the model bias in the seasonal mass balance validation (also commented on by reviewer #1). Please see the detailed replies to your comments below.*

**Reviewer comment: The proposed manuscript presents new estimates of future sea-level rise obtained with a new model of global glacier evolution. The model is quite original because it is integrated in a land surface model. Also, the mass-balance is computed from the surface energy balance, which could lead to different and more contrasted results than classical "temperature-index" mass-balance models. For these reasons, it is a useful addition to the literature. That said, I think that the paper needs serious revisions before being considered for publication. I have some major concerns listed below.**

**My recommendation would be to:**

 **• use additional datasets for model validation**

**• put less focus on the validation with point mass-balance data**

**• spend more time on the energy balance instead**

*Reply: To focus more on the energy balance and to justify why we give a high prominence to adjusting wind speed (your point below) we have added a comparison of the modelled energy balance components with observations from the Pasterze glacier in the Alps (Greuell and Smeets 2001). We found that the model underestimated the sensible heat flux by an order of magnitude and the wind speed was four times lower than the observations. The sensible heat flux is underestimated because the surface exchange coefficient (used to calculate the turbulent heat flux) is proportional to the wind speed.*

*You make the point (further down) that the wind speed scaling does not seem physical because the katabatic wind is decoupled from synoptic conditions. We agree that our approach is rather crude and could certainly be improved. However, we thought it was better to include a crude adjustment for wind speed rather than neglecting this process altogether.*

*Change to manuscript: We have added the following to model description of wind speed section 2.2.4*

*"Although our approach is rather crude, we found that scaling the wind speed was necessary to get reasonable values for the sensible heat flux. This is seen when we compare the modelled energy balance components to observations from the Pasterze glacier in the Alps (Greuell and Smeets 2001). The measurements consist of incoming and outgoing short and long wave radiation, albedo, temperature, wind speed and roughness length at five heights between 2205m-3325m meters above sea level on the glacier. Table S6 in the Supplementary Material lists the observed and modelled energy balance components and meteorological data, for experiments with and without wind speed scaling. The comparison shows that JULES underestimates the sensible heat flux by at least one order of magnitude and the modelled wind speed is four times lower than the observations. When we increase the wind speed to match the observations there is a better agreement with the observed sensible heat*

*flux. This is because the surface exchange coefficient, which is used to calculate the sensible heat flux, is a function of the wind speed in the model. "*

**Reviewer comment: Model set-up I found it difficult to understand some aspects of the model set-up, and I believe the text could be meliorated by reorganizing some of its sections.**

**In particular:**

**• the description of the snowpack/ice meld model is scattered around Section 2/ Intro, and page 7.**

*Reply: We preferred to keep the description of the snowpack initialisation (page 7) separate to the snowpack description (Section 2). We also kept the description of the multi-level snowpack scheme brief because it is described in detail in Best et al 2011.*

**Reviewer comment: the description of how and where glaciers lose mass is unclear to me (and to the other reviewer as well)**

*Reply: We have added the following text early in the model description section to help clarify how and where glaciers loose mass*

*Change to manuscript: "Each elevated glacier tile has a snowpack which can gain mass through accumulation and freezing of water and lose mass through sublimation and melting. JULES has a full energy balance multi-level snowpack scheme which splits the snowpack into layers each having a thickness, temperature, density, grain size (used to determine albedo), and solid ice and liquid water contents. The initialisation of the snowpack properties and the distribution of the glacier tiles as a function of height is described in section 2.3"*

**Reviewer comment: The manuscript would also benefit from a clear discussion and listing of the particularities (strengths/weaknesses) of this model in comparison to previous modelling attempts.**

**This could be a way to highlight the strengths of your model (see "Energy balance" below).**

*Reply: We modified the discussion section to list the strengths and weaknesses of the model.*

*Change to manuscript: "There are three key strengths to the JULES glacier model. Firstly, we include variations in orography within a climate gridbox which is important to calculate elevation-dependent glacier mass balance. Kotlarski et al (2010) developed a glacier scheme for the REMO regional climate model by lumping glaciers into 0-5-degree gridboxes in a similar approach to us, but they did not have a representation of subgrid orography. Instead glacier gridboxes received double the gridbox mean snowfall, glacier ice had a fixed albedo and a constant lapse rate was applied to adjust temperatures. They concluded that to reproduce mass balance trends over the Alps, the scheme needed to include subgrid variability of atmospheric parameters within a gridbox.*

*Secondly, the model uses a full energy balance scheme to calculate glacier melting. This is a more physically based approach than the widely used temperature index models, which relate melting to temperature using a degree day factor (DDF). The DDF lumps all the energy balance components into a single number meaning that the effects of changing wind speed, cloudiness and radiation on melt rates cannot be considered. Changes in solar radiation can be an important driver of melting. Huss et al (2009) studied long term mass balance trends for a site in the Alps and showed that melting was stronger during the 1940's than in recent years despite more warming. This was because summer solar radiation was higher during the 1940s. Moreover, temperature index models have been found to be less accurate with increasing temporal resolution (for example on daily time steps) (Hock 2005).*

*Finally, the glacier scheme is coupled to a land surface model, which presents opportunities for further studies. For instance, the model could be used to investigate the impact of climate change on river discharge in glaciated catchments in Asia, South America or the Arctic."*

*One of the major shortcomings of the model is that glacier dynamics is not included (glacier area does not vary). The model does not simulate the retreat of the glacier terminus which results in an overestimate of mass loss. Neither does the model simulate the transport of ice from higher elevations to lower elevations.*

*An additional drawback of the model is the coarse resolution of the gridboxes which make it unfeasible to include some process which affect local mass balance such as hillside shading, avalanching, blowing snow and calving. The model could, however, be run on a finer resolution using higher resolution climate forcing data."*

**Reviewer comment: Calibration/validation**

**Validating and calibrating against in-situ elevation dependant MB data is really hard. Point mass-balance observations reflect a number of local and glacier specific factors, and there is a risk that these local factors affect parameter sets chosen for entire RGI C2 regions (this is particularly true for regions with few observations). I am not asking to change your procedure as this will likely represent too much work, but I'm allowing myself to provide a couple of suggestions:**

**I would personally recommend to use one global set of parameters instead of the regional ones, unless there is a compelling reason not to do so (a good candidate would be a physical explanation of the regional parameter sets). The resulting parameter sets would be more robust and would allow statistical scrutinizing using cross-validation (or more advanced) methods. This is even more relevant for physically based approaches like yours.**

*Reply: It is not clear why a single global parameter set would be more robust than regional parameters sets. For example, in Table 2 we show that the optimal value for the fresh snow albedo in the visible range is 0.83 in Central Europe and 0.97 in Western Canada and the US. A single parameter set would not capture this regional variation. Similarly, the wind speed scaling varies between the regions, 1.83 in Europe and 2.29 in Western Canada and the US.*

**Reviewer comment: Having some kind of independent validation would considerably strengthen the readers' confidence in your results. Albeit not without their own problems, regional geodetic mass-balance estimates could be useful to at least get a quantitative estimate of the model performance at the regional scale.**

*Reply: We agree with the reviewer that the calibration approach could be improved so we have added a section on this in the discussion.*

*Change to manuscript: "The robustness of the glacier projections depends on how well the model can reproduce present-day glacier mass balance. One of the main shortcomings of the calibration and validation of mass balance is that only a single type of observations is used. This data was used because we wanted to ensure the model could reproduce variations in accumulation and ablation with height when the elevated tiling scheme was introduced. Point mass balance observations are affected by local factors such as aspect, avalanching, debris cover and there is a possibility that these local factors affect*

*parameter sets chosen for entire RGI region. This could be improved by using observations from satellite gravimetry and altimetry, such as that described by Gardner et al (2013) to get a quantitative estimate of the model performance at the regional scales. "*

**Reviewer comment: Energy balance**

**The real strength of the model is its use of an energy balance model instead of a temperature index model as the majority of the other global models. Whether or not this increase in complexity is actually leading to better results remains (and will remain) a controversial topic, but this study should make use of this novel approach. In particular, I would find it very interesting to see plots of the energy balance components as a function of altitude, and how these energy balance components change in the future. This is interesting because energy balance models are likely to be less sensitive to temperature change and incorporate other processes instead. I would also welcome new analyses of not only the total volume change, but the volume changes per elevation band.**

*Reply: This extra analysis is added to the paper*

*Change to manuscript: We added percentage volume changes for low, medium and high elevation ranges to Table 6 and added the following text to section 4.2*

*"The percentage volume changes for three different elevation ranges; low (0-2000m), medium (2250m-4000m) and high (4250m-9000m) are listed in Table 6. Some of the high latitude regions particularly Alaska, Western Canada & US, Svalbard and North Asia experience very large volume increases at their upper elevation ranges. This would be reduced if the model included glacier dynamics, because ice would be transported from higher elevations to lower elevations."*

*Change to manuscript:*

*"**4.3 Mass Balance Components***

*In this section we examine how the surface mass balance components vary with height and how this will change in the future. Fig. 12 shows the accumulation, refreezing and melting contributions to mass balance averaged over low, medium and high elevations ranges for the period 1980-2000. Sublimation is excluded because its contribution to mass balance is relatively small. As expected there is more melting in the lower elevation ranges and more accumulation at the higher elevation ranges. The refreezing component, which includes refreezing of melt water and elevated adjusted rainfall, shows no clear variation with height. This is because the refreezing component can both increase and decrease with height. Refreezing can increases towards lever elevations because there is more rain and melted water. It can also decrease if the snowpack is depleted or if there is not enough pore space to hold water because previous refreezing episodes have converted the firn into solid ice. The largest accumulation rates occur in Alaska (5.3 m.w.eq.yr-1) and Western Canada and US (7.3 m.w.eq.yr-1) between 4250m-9000m and the largest melt rates are found in the Caucasus and Middle East (-7.4 m.w.eq.yr-1) and the Low Latitudes (-7.6 m.w.eq.yr-1).*

*Fig. 13 shows how the global annual mass balance components vary with time for low, medium and high elevations ranges. At the high and medium elevations accumulation, refreezing and melting decrease leading to a reduction in mass loss as glaciers disappear towards the end of the century. At high elevations mass balance is reduced from -2.2 m.w.eq.yr$^{-1}$ (-177 Gtyr-1) during the historical period (1980-2000) to -0.35 m.w.eq. yr-1 (-28 Gtyr$^{-1}$) by the end of the century (2080-2097). Similarly, for*

*the medium elevation ranges mass balance reduces from -0.56 m.w.eq.yr$^{-1}$ (-26 Gtyr$^{-1}$) to -0.24 m.w.eq.yr$^{-1}$ (-11 Gtyr$^{-1}$)."*

**Reviewer comment: Code availability Please add a statement about where and how people can access your code and that of JULES.**

*Reply: We added a section on code availability at the end of the paper.*

*Change to manuscript:*

***Code availability***

*The glacier scheme is included in JULES v4.7. The source code can be downloaded by accessing the Met Office Science Repository Service (MOSRS) (requires registration): https://code.metoffice.gov.uk/ The code used for this study is in* [https://code.metoffice.gov.uk/svn/jules/main/branches/dev/sarahshannon/vn4.7_va_scaling](https://code.metoffice.gov.uk/svn/jules/main/branches/dev/sarahshannon/vn4.7_va_scaling)

**Reviewer comment: Specific comments**

**P3 L15 why is this limitation about the partial coverage necessary? In view of the objective of developing a fully coupled model, it would be good to overcome this limitation one day.**

*Reply: We agree with the reviewer that it would be preferable to be able to mix the elevated tiles and vegetated tile schemes within gridboxes. There are a number of structural difficulties in the JULES code that make it very difficult to do this unfortunately, well outside the scope of even the significant code development that was done for the elevated tiles used in this study.*

*One of the main difficulties is that the primary soil model in JULES (which is essential for the vegetation, but incompatible with glaciated tiles) is fundamentally structured to run with one set of variables and parameters for each gridbox. Thus, for each gridbox one must choose to have either a soil or an ice subsurface, and all the tiles above for that gridbox must fit exclusively into one of those two categories. Work has been ongoing for a number of years in the JULES development groups to allow the subsurfaces (like soil) to be tiled like the surfaces, but this is a major undertaking which touches almost every part of the codebase and as yet there is no firm timescale for this work to be completed. In addition, the surface tiles in JULES can change area fraction during the course of a run, as climate favours different vegetation types or a glacier changes in volume, but extending this essential functionality to the tiled soils brings non-trivial challenges in carbon and water conservation within the model that have yet to be addressed.*

**Reviewer comment: P3 L23 0.5° and 46 elevation bands: What motivated the choice of these resolutions? Can this be changed at wish?**

*Reply: Yes, these resolutions can be changed. The 0.5° resolution was chosen because the climate data (historical and future) is on this resolution. The vertical resolution of 250m was chosen for computational cost. This could be increased to a finer vertical resolution, but a new tile fraction ancillary and initial snowpack depth would need to be generated from the 50m glacier hypsometry data in the RGI6.*

*Change to manuscript: Added text to the model description: "The horizontal resolution of 0.5-degree is used because it matches the forcing data used to drive the model. The vertical resolution of 250m was used based on computational cost. The vertical and horizontal resolutions of the model can be modified for any setup."*

**Reviewer comment: P4 L5 Snowpack: do I get this right that there is no distinction between ice and snow in the snowpack model? What are typical values for ice density in the model? How much time does it need to transform snow to ice?**

*Reply: There are some distinctions between snow and ice in the snowpack. Snow and ice have different densities, ice and liquid water content, grain size (which determines albedo) and temperature. Fresh snow at the top of the snowpack has a typical density $250 kgm^{-3}$ and ice has a density $917 kgm^{-3}$. The albedo is treated differently for ice and snow. The new albedo scheme, which scales the albedo as a function of the snowpack surface density, is activated when the firn density is greater than $550 kgm^{-3}$ Below this threshold the snow aging scheme is used.*

*We have not estimated how long it would take for snow to convert to ice, because in our model setup we always prescribe the bottom of the snowpack with solid ice. However, theoretically the change in density and grain size with time is described in Best et al 2011 (Equations 21 and 39 respectively). It would be interesting to explore whether the model can grow realistic glaciers from scratch and if so, how long this would take.*

**Reviewer comment: P4 L10 I must admit that I dislike the current approach to temperature downscaling, which in my opinion is an unhealthy mix of thermodynamics and tuning. I'm not asking to change it, but the fact that the lapse-rate is tuned in non-saturated conditions (the rate changes quite a lot according to table 2) but not in saturated conditions is likely to create odd non-linearities in the model's response to certain forcings. P5 L1 "we only tune the dry adiabatic lapse-rate". Here and throughout the rest of the manuscript: do not use the term "dry adiabatic lapse-rate". The dry adiabatic lapse-rate is the dry adiabatic lapse-rate and is 9.8K per 1000m. What you are tuning though is the near-surface temperature lapse-rate, which might vary according to surface conditions and moisture content.**

*Change to manuscript: We removed "P5 L1 "we only tune the dry adiabatic lapse-rate" and replaced instances of "dry adiabatic lapse-rate" with "lapse rate"*

**Reviewer comment: P5 L22 wet-bulb temperature is a much better indicator for solid precipitation than regular dry-bulb temperature. This could mitigate parts of the dramatic changes in snowfall projected by your scenarios.**

*Reply: We agree with the reviewer that using wet-bulb temperature to partition rain and snow is better than dry-bulb temperature. We did preliminary sensitivities studies which showed that the mass balance was more sensitive to the precipitation lapse rate than the temperature for partitioning rain and snow. However, one of the reasons we have a negative bias in the seasonal mass balance (an overestimate in melting and an underestimate in accumulation) is because we use a dry bulb temperature of $^0C$ which is likely too low for partitioning rain and snow. We have added the following text to the model validation section 3.2*

*Change to manuscript: "Some, but not all, of the bias is due to the partitioning of rain and snow based on an air temperature threshold of $0^oC$. The $0^oC$ threshold is likely too low, resulting in an underestimate of snowfall. When precipitation falls as rain or snow it adds liquid water or ice to the snowpack. The specific heat capacity of the snowpack is a function of the liquid water ($W_k$) and ice content ($I_k$) in each layer*

$$C_k = I_k C_{ice} + W_k C_{water} \tag{17}$$

where $C_{ice}$ = 2100 JK$^{-1}$kg$^{-1}$ and $C_{water}$ = 4100 JK$^{-1}$kg$^{-1}$.  The liquid water content is limited by the available pore space in the snowpack, therefore changes in the ice content control the overall heat capacity. The underestimate in the ice content reduces the heat capacity which causes more melting than observed.

Other modelling studies have used higher air temperature thresholds; 1.5$^o$C (Huss and Hock 2015, Giesen and Oerlemans 2012), 2$^o$C (Hirabayashi et al 2010) and 3$^o$C (Marzeion et al 2012).  An improved approach would use the wet-bulb temperature to partition rain and snow which would include the effects of humidity on temperature.  Alternatively, a spatially varying threshold based on precipitation observations could be used. Jennings et al (2018) showed by analysing precipitation observations, that the temperature threshold varies spatially and generally higher for continental climates than maritime climates.

**Reviewer comment: P5 L27 here and at some other places in the manuscript, the missing katabatic flow is given a high prominence in the list of missing processes to be addressed. This might be the case (also not the most prominent on my list), but the proposed solution (scaling the synoptic wind field) does not sound really physical to me. The katabatic flow is notoriously decoupled from the synoptic conditions and is likely to be strongest when the synoptic flow is weak. What the scaling of the modelled wind achieves, though, is an increase of the turbulent fluxes: I would be very interested in seeing more discussion about why this is necessary (see major point 1.3 above).**

*Reply: The reason we include katabatic winds was to increase the modelled sensible heat flux compared to observations.  Please see our first point (above) responding to this.*

**Reviewer comment: P7 L1-2 If I get this right, the glacier tiles are able to lose mass per elevation band, right? The information is scattered in the manuscript (P7 L10, P7 L24 . . . ) and should be clarified much earlier to avoid confusion (see also the comment from reviewer 1 who seems to have understood something different than me).**

*Reply: Yes, you understand correct.*

*Change to manuscript: We add this line early in the model description at page 3 to help clarify "This allows glacier tiles to gain or lose mass at elevation bands"*

**Reviewer comment: P7 L10 I don't understand this part. Can you be more specific about how glaciers grow/shrink and lose/gain mass in the model?**

*Reply: In this line we refer to the glacier tile fraction which does not grow/shrink because the glacier area is fixed. The tile fraction (i.e. the fraction of a gridbox covered in ice at elevation levels) is calculated from the glacier area. Glaciers gain/lose mass though melting, sublimation, accumulation and refreezing in the snowpack but the area (tile fraction) remains fixed.*

**Reviewer comment: P7 L25 What happens at the end of the initialisation? Setting 500 m of ice everywhere is maybe ok for a spin-up, but what happens next, or at the start of the 2011-2100 simulation?**

*Reply: For the future simulations the depth of the bottom level of the snowpack comes from the RGI6 thickness which is based on the thickness inversion Huss and Farinotti (2012).*

*Change to manuscript: Added this line to the section 2.3.2 "*

*For the future simulations the thickness and ice mass at the bottom of the snowpack comes from thickness and volume data in the RGI6. The data is based on thickness inversion calculations from Huss and Farinotti (2012) for individual glaciers which are consolidated onto 0.5-degree gridboxes.*

*Reply: We noticed a typo in the model description which we have corrected. For the calibration period the initial ice thickness is 1000m not 500m. The spin-up period is 10 years not 1 year. Our description of the spin-up/initialisation is not very clear so we have modified it.*

*Change to manuscript: "The snowpack temperature profile is calculated by spinning the model up for 10 years for the calibration period and 1 year for the future simulations. The temperature at the top layer of the snowpack is set to the January mean temperature and the bottom layer and subsurface temperature is set to the annual mean temperature. For the calibration period the monthly and annual temperature comes from the last year of the spin-up. Setting the snowpack temperature this way gives a profile of warming towards the bottom of the snowpack representative of geothermal warming from the underlying soil. The initial temperature of the bedrock before the spin up is set to 0°C but this adjusts to the climate when the model spins up. We use these prescribed snowpack properties as the initial state for the calibration and future runs. "*

**Reviewer comment: P9 -L27 I don't understand the statement "with the notable exception of the low latitude and Central European regions where melting is over estimated". According to Table 3 and the BIAS measure, melt is over-estimated in 9 regions with Svalbard, Southern Andes and New Zealand striking out with more than 1 m negative bias.**

**C5 Central Europe even has a positive bias. A quick look through the table indicates a general negative bias.**

*Reply: The reviewer is correct*

*Change to manuscript: We removed this line and added "Nine out of the sixteen regions have a negative bias in the annual mass balance. Notably Svalbard, Southern Andes and New Zealand underestimate mass balance by 1 m.w.eq.yr$^{-1}$. "*

*Change to manuscript: Also, we noticed that the column containing the number of observations did not match those labelled in Figure 3 so we have corrected this.*

**Reviewer comment: P10 L3 and Figure 4: the explanation about the Maladeta is irrelevant. Figure 4 shows that there are other pink dots around the Maladeta starts, and there is no need for a case by case explanation here.**

*Reply: There was a mistake in this figure, so we have updated it. The new figure shows our point that the model performs particularly badly for the Maladeta glacier where melting is overestimated.*

*Change to manuscript: Figure 4 changed.*

**Reviewer comment: Section 3.2 This does not represent an independent validation because the same data was used for calibration also. What is striking in Table 4 is that all regions now have a significant negative bias both in winter and summer. How can this be explained?**

*Reply: The fact we are using similar data to validate the model is certainly a weakness of the calibration and we added this point to the discussion (see above). The negative bias (model overestimates melting in the summer and underestimates accumulation in the winter) was also noted by Reviewer #1 and is likely caused by not bias correcting the gridbox mean precipitation before applying the lapse rate adjustment. This causes an underestimate in the snowfall. (Please see our response to reviewer #1 about this).*

**Reviewer comment: P10 L21 I would like to see more explanations about the "downscaling" procedure. If only SST and sea-ice are used, this sounds a lot more like a full atmosphere GCM simulation to me than a "downscaling" of a GCM product. In particular, what happens to the land-surface components in HadGEM3? What is actually left from the original GCM signal after "downscaling"?**

*Reply: The reference to downscaling refers to the use of a higher resolution atmosphere model (HadGEM3 GA6.0) to produce new projections consistent with the CMIP5 SST and sea ice projections used to drive these simulations. A more typical usage of the term downscaling (in dynamical terms) might involve the running of a higher resolution limited area regional climate model with boundary conditions supplied by a GCM.*

*HadGEM3 benefits from an increased horizontal and vertical resolution over the CMIP5 HadGEM2-ES model, and also has substantial changes to the model dynamics. The version of HadGEM3 used here represents a transition between the CMIP5 and CMIP6 versions of the Hadley Centre climate model. The use of a sub-set of CMIP5 model SST and sea ice as drivers allows an exploration of uncertainties in the regional impacts of climate change, but consistent with the CMIP5 simulations.*

**Reviewer comment: P11 L2 remove "which is suitable for capturing precipitation variability over complex topography"**

*Reply: We removed this as suggested.*

**Section 4.3 Parametric uncertainty analysis is one aspect of parameter uncertainty. A further uncertainty would be revealed by doing data-denial experiments (crossvalidation) and assessing the sensitivity of your calibration procedure to those.**

*Reply: We agree that data-denial experiments would be valuable to assess the calibration uncertainty. We explored the calibration uncertainty in a slightly different way in response in reviewer #1 comments. We selected best parameters from our Latin Hype Ensemble by minimising the bias and RMSE and maximizing the correlation coefficient. The uncertainty in the global volume loss when the extra performance metrics are used to calibrate present-day mass balance, is approximately double the uncertainty arising from the different climate forcings (Fig. 16, Table 7). This shows the calibration approach has a large impact of the results.*

**Reviewer comment: Section 4.4 Comparison with other studies I would welcome future studies based on this model to use the same forcing data and same conventions as other global glacier models where possible in order to facilitate model intercomparisons.**

*Reply: We hope that the model can participate in the Glacier Model Inter-Comparison Project (glacierMIP http://www.climate-cryosphere.org/activities/targeted/glaciermip). This project will compare glacier volume projections for a range of global glacier models each using the same climate forcing and initial ice volumes (RGI6).*

**Reviewer comment: P13 L25 My understanding is that your study is using the global volume estimates provided by Matthias Huss.**

*Reply: Yes, that is correct.*

**C6 Supplementary material I suggest to invert the color scale: red seems more intuitive for mass loss.**

*Change to manuscript: We reversed the colour scale as suggested.*

Best, M. J., M. Pryor, D. B. Clark, G. G. Rooney, R. L. H. Essery, C. B. Ménard, J. M. Edwards, M. A. Hendry, A. Porson, N. Gedney, L. M. Mercado, S. Sitch, E. Blyth, O. Boucher, P. M. Cox, C. S. B. Grimmond & R. J. Harding (2011) The Joint UK Land Environment Simulator (JULES), model description – Part 1: Energy and water fluxes. *Geosci. Model Dev.,* 4**,** 677-699.

Gardner, A. S., G. Moholdt, J. G. Cogley, B. Wouters, A. Arendt, J. A.Wahr, E. Berthier, R. Hock, W. T. Pfeffer, G. Kaser, S. R. M. Ligtenberg, T. Bolch, M. J. Sharp, J. O. Hagen, M. R. van den Broeke & F. Paul (2013) A Reconciled Estimate of Glacier Contributions to Sea Level Rise: 2003 to 2009. *Science,* 340**,** 852-857.

Greuell, W. & P. Smeets (2001) Variations with elevation in the surface energy balance on the Pasterze (Austria). *Journal of Geophysical Research-Atmospheres,* 106**,** 31717-31727.

Huss, M. & D. Farinotti (2012) Distributed ice thickness and volume of all glaciers around the globe. *Journal of Geophysical Research,* 117**,** F04010.

Rye, C. J., I. C. Willis, N. S. Arnold & J. Kohler (2012) On the need for automated multiobjective optimization and uncertainty estimation of glacier mass balance models. *Journal of Geophysical Research-Earth Surface,* 117.

---

## Referee Report (RR1)

I have read the paper a second time after the author's revision. The authors invested a lot of energy in the revision, which is recommendable. Some of my comments have been addressed, other have been ignored or implemented differently. Some model results are still dubious to me (mostly: the surprising seasonal mass-balances and the mass fluxes per elevation band). Overall I am still convinced that there is a potential for model improvements, but the current version of the paper discusses the model limitations in an appropriate way.

I have a few minor comments listed below and would like to take the opportunity to reply to three topics raised in the interactive discussion.

**Points raised in the discussion**

From the three points below, only point 1 needs concrete action in the manuscript. Points 2 and 3 are here just for the sake of scientific debate.

**1. Elevation feedback**

From the author's response to Reviewer #1 and myself I found two contradicting statements:

To reviewer #1 you wrote: "*The negative feedback between terminus elevation and mass balance is missing and the only way for a melting glacier to reach equilibrium with climate is by melting completely.*" And to my question about whether glaciers can melt on elevation bands you write: "*This allows glacier tiles to gain or lose mass at elevation bands*". I still have trouble to understand what you actually mean in your answer to reviewer #1 and in the text: if you loose mass at an elevation band you could include an elevation feedback by letting the band's elevation decrease until the bedrock is reached (which you probably won't do because of obvious complications in the code). However, you are able to stop the melt when an elevation band is melted completely. So some negative feedback should already be included in your model, and you might revise the answer to reviewer #1 by saying that the *area* is left unchanged, which is better than leaving the entire elevation band after it has melted.

**2. Regional parameter sets.**

To my comment about regional calibration, you write: "*It is not clear why a single global parameter set would be more robust than regional parameters sets.*"

Let me make an example based on JULES. How would it be if the model parameters for, say, "clay porosity" or "tree leave albedo" would be different between England and Wales? The equations of wind motion or ice melt do not follow arbitrary frontiers. I might be wrong, but this glacier module is probably the first module in JULES to use regional parameter sets.

Don't get me wrong: I understand that parameters need to be tuned, especially in a "physically based model" with many parameters. I just say that using parameters based on RGI regions is suboptimal, for several reasons:
- it creates unphysical differences between neighboring regions (such as 13, 14, 15 in High Asia)
- it hides model deficiencies (or errors in forcing data) by tuning the model on a smaller set of observations (sometimes only one or two glaciers per region)
- in a global model like JULES, it will hinder the acceptance by the wider community and the module will have more difficulties to enter the main codebase

**3. Energy balance**

In the revised version you added analyses of mass fluxes (which can be done by more simple models like degree-day models as well), but not of energy fluxes. I believe this is a missed opportunity.

**Detailed comments**

P6 L26-27: remove "this is because..."
P9 L6: add Marzeion et al., 2012 to the references list.
P10 L23: please add reasons for the negative bias. In linear a model with enough degrees of freedom, minimizing RMSD will always minimize the bias too. So the first thing that comes to mind is stat systematic problems in the model and/or the forcing data are preventing this bias minimization (confirmed by the supplementary analyses). In short: there seems to be a structural problem in either the model or the forcing data.
P10 L29: "Our mass balance model does include sublimation". I am curious: since you have a latent heat flux, why don't you simply convert it to a mass loss? This is the typical way to compute sublimation in glacier energy and mass balance models.
Figure 4: you might consider add Maladeta to Figure 3 and spare a figure.
Table 6: consider making a bar-plot out of it for more readability
P13 L30: here you talk about sublimation. This contradicts your statement above.
Fig. 12: to make the figure more readable you could remove the x and y axis labels for the interior plots, since they are the same for each plot.
Figure 13 and corresponding analysis in the text: I have trouble to understand why the upper elevations see a reduction in melt while the lower parts do not? The provided explanation ("reduction in mass loss as glaciers disappear towards the end of the century" holds even more true for lower elevations. Or is this due to regional differences, the high latitude arctic having more mass below 2000 m a.s.l? This needs more explanation in the text.
P16-L16: about elevation feedback - see main comment above.
P18 L20: "*Changes in solar radiation can be an important driver of melting.*": It is a bit sad that you didn't take my advice about analysing the energy fluxes...

---

## Author Response (AR2)

**Dear Sarah Shannon and co-authors**

**Thank you for re-submitting this manuscript to TC. I appreciate all the work you have put into the revision in order to address the reviewer comments, which really improved the manuscript.**

**In general, I agree that this is an important contribution to the literature as it reports a novel approach to glacier modelling inside a land surface scheme. I agree with the reviewers that this is more a report about the current progress in this field which should also identify possibilities for improvement.**

**Please, address carefully the second round of reviewer comments and improve the manuscript accordingly. I fully agree with the reviewer comments and will check if these have been addressed thoroughly prior to any final publication. My own concerns are about the strong model biases and negative NS coefficients in winter and summer: How reliable are the future projections in light of these biases? What are the major improvements that you recommend for future studies?**

**I am very much looking forward to a detailed point-by-point reply to the reviewer comments and an improved version of the manuscript.**

**Kind regards**

**Christian Beer**

Dear Christian Beer,

Thank you for your feedback on the manuscript. Please find the responses to your comments and the reviewers comments below.

In response to your comments:

**My own concerns are about the strong model biases and negative NS coefficients in winter and summer:**

1. **How reliable are the future projections in light of these biases?**

*We are aware that the present-day seasonal mass balance contains a negative bias suggesting that melting is overestimated in the summer and accumulation is underestimated in the winter. The bias may result in an overestimate in future volume loss, although this is uncertain given other factors may have a larger effect in the future, for example uncertainty in the climate forcing.*

*The bias is caused by an underestimate in snowfall due to the simple approach to correcting coarse scale gridded precipitation for orographic effects and lapse rate. This problem is not unique to our approach but affects many attempts to use global climate model information to make projections of glacier mass balance. A more nuanced approach could be to use a more detailed regional climate model data that explicitly represent orographic effects. Alternative approaches effectively add a scalar on precipitation to capture orographic enhancement and assume this to be constant under a future climate then as a second step lapse the precipitation across elevation bands. Standing alone these projections may contain larger uncertainties than other global glaciers model, due to bias in the calibration, the coarse nature of JULES and missing processes. The model could however, contribute to the aim of generating reliable future projections, by contributing an alternative type of global glacier model to a glacier model intercomparison exercise.*

*To ensure we make the point that the bias might cause an overestimate in the volume loss projection we add the following to the top of the discussion*

"Our calibrated seasonal mass balance contains a negative bias (accumulation is underestimated, and melting is overestimated) which suggests that the volume loss projections might be overestimated"

**2. What are the major improvements that you recommend for future studies?**

*Firstly, we think calibration approach could be improved. This is certainly the most challenging part of this work. Future work would test if the negative bias in the seasonal mas balance could be reduced or eliminated by scaling the gridbox mean precipitation prior to applying the lapse rate correction. We would also consider extrapolating the precipitation lapse rates to other regions using an empirical relationship which categorises the climate, similar to the work of Radic et al 2014., Also, we would use observations from satellite gravimetry and altimetry, such as that described by Gardner et al (2013) to get a quantitative estimate of the model performance at the regional scales (included in the discussion section P17 Line 12).*

*Secondly, we would add a volume-area scaling scheme to remove the limitation of fixed glaciated area (included in the discussion section P17 line 25). We would consider evaluating the scheme over the Alps, because the region is data rich in comparison to other locations and a comparison could be made to the glacier scheme in the REMO regional climate model (Kotlarski et al. 2010).*

**I have read the paper a second time after the author's revision. The authors invested a lot of energy in the revision, which is recommendable. Some of my comments have been addressed, other have been ignored or implemented differently. Some model results are still dubious to me (mostly: the surprising seasonal mass-balances and the mass fluxes per elevation band). Overall I am still convinced that there is a potential for model improvements, but the current version of the paper discusses the model limitations in an appropriate way. I have a few minor comments listed below and would like to take the opportunity to reply to three topics raised in the interactive discussion. Points raised in the discussion**

**From the three points below, only point 1 needs concrete action in the manuscript. Points 2 and 3 are here just for the sake of scientific debate.**

1. Elevation feedback: From the author's response to Reviewer #1 and myself I found two contradicting statements: To reviewer #1 you wrote: "The negative feedback between terminus elevation and mass balance is missing and the only way for a melting glacier to reach equilibrium with climate is by melting completely." And to my question about whether glaciers can melt on elevation bands you write: "This allows glacier tiles to gain or lose mass at elevation bands". I still have trouble to understand what you actually mean in your answer to reviewer #1 and in the text: if you loose mass at an elevation band you could include an elevation feedback by letting the band's elevation decrease until the bedrock is reached (which you probably won't do because of obvious complications in the code). However, you are able to stop the melt when an elevation band is melted completely. So some negative feedback should already be included in your model, and you might revise the answer to reviewer #1 by saying that the area is left unchanged, which is better than leaving the entire elevation band after it has melted.

*You are correct here, there is an elevation feedback, so we will revise the answer to reviewer #1. Part of the reason for the confusion is that the elevation feedback is different to our classical understanding of elevation feedback. For example, in a classical case, if a glacier thickens then the top of the glacier*

*will experience cooler temperatures.  In our case if the snowpack thickens the top of the snowpack doesn't feel that cooler temperature but it experiences the temperature of the elevation band. The elevation feedback in our model is simpler than the classical elevation feedback mechanism.*

*Changes to manuscript:*

*We added the following to the model description section (P4 L13)*

*"The scheme assumes that the snowpack can grow or shrink at elevation bands depending on the mass balance, but that tile fraction (derived from the glacier area) is static with time. The ability to grow or shrink the snowpack at elevation levels means that the model includes a simple elevation feedback mechanism.  If the snowpack shrinks to zero at an elevation band, then the terminus of the glacier moves to the next level above. On the other hand, if the snowpack grows at an elevation band it just continues to grow and there is no process to move the ice from higher elevations to lower elevations. Typically, in an elevation feedback, when a glacier grows the surface of the glacier will experience a cooler temperature, however in this case, the snowpack surface experiences the temperature of the elevation band."*

*We removed the following text P16 L18 and the citation of Marzeion et al (2014) in the reference list.*

*Another explanation why we predict more volume loss than Radic et al. (2014) and Huss and Hock (2015) is because there is no retreat of the glacier terminus represented in the model. The only way for glaciers to reach equilibrium with climate is by melting completely. A study by Marzeion et al. (2014) showed that models predict more mass loss when the terminus elevation is fixed than when it is allowed to vary. This is because when the terminus is allowed to retreat, there will be less area available to melt.  Marzeion et al (2014) found that neglecting terminus elevation changes resulted in an extra few tens of mm SLE depending on RCP scenario. Lastly, some of the differences between our study and other published projections could be due models using different initial ice volumes and glaciated areas.*

**2. Regional parameter sets. To my comment about regional calibration, you write: "It is not clear why a single global parameter set would be more robust than regional parameters sets." Let me make an example based on JULES. How would it be if the model parameters for, say, "clay porosity" or "tree leave albedo" would be different between England and Wales? The equations of wind motion or ice melt do not follow arbitrary frontiers. I might be wrong, but this glacier module is probably the first module in JULES to use regional parameter sets. Don't get me wrong: I understand that parameters need to be tuned, especially in a "physically based model" with many parameters. I just say that using parameters based on RGI regions is suboptimal, for several reasons: - it creates unphysical differences between neighboring regions (such as 13, 14, 15 in High Asia) - it hides model deficiencies (or errors in forcing data) by tuning the model on a smaller set of observations (sometimes only one or two glaciers per region) - in a global model like JULES, it will hinder the acceptance by the wider community and the module will have more difficulties to enter the main codebase**

*We agree with your point that this approach will result in arbitrary thresholds between adjacent regions. As you know the motivation for using regional parameter sets is because different regions have different process (for example debris cover is more prevalent in Himalayan glaciers than Alpine glaciers). If the model included more processes, for instances avalanching, blowing snow etc..) then there would be no need to have regional parameters.*

*There are examples in JULES where regional parameter sets are used. For example*

- Later configurations which use spatially varying albedo properties based on satellite ancillaries

- Different fresh snow density in the UK configuration compared to the global

*In terms of JULES, there are no barriers for the inclusion of the module into the code base, however, the way in which the glacier scheme is used in specific model configurations is likely to require more work.*

**2. Energy balance**

**In the revised version you added analyses of mass fluxes (which can be done by more simple models like degree-day models as well), but not of energy fluxes. I believe this is a missed opportunity.**

*Apologies, I miss-interpreted your request as asking for mass fluxes per elevation, hence the extra analysis of this. We have added some extra material on the future global energy fluxes. Since the manuscript is already quiet long, we have kept the extra material brief. An analysis of how the fluxes vary regionally and with height would be interesting for a follow up paper.*

*Changes to manuscript:*

*We added an extra figure (Fig 11) showing the ensemble mean energy balance components for all glaciated regions averaged over all elevation levels.*

*We also added the following text to section 4.2 Regional glacier volume projections 2011-2097*

*"To investigate which parts of the energy balance are driving the future melt rates, we show the energy balance components averaged over all regions and all elevation levels in **Error! Reference source not found.**. Future melting is caused by a positive net radiation of approximately 30 Wm$^{-2}$ that is sustained throughout the century. This is comprised of 18 Wm$^{-2}$ net shortwave, 3 Wm$^{-2}$ net longwave, 5 Wm$^{-2}$ latent heat flux and 4 Wm$^{-2}$ sensible heat flux. The largest component of the radiation for melting comes from the net shortwave radiation. The upward shortwave radiation comprises of direct and diffuse components in the visible and near infrared wavelengths. The visible albedo deceases because melting causes the ice surface to darken. In contrast, the near infrared albedo increases because the ice is heating up emitting radiation in the infrared part of the spectrum. The downward and upward longwave radiation are increasing in future however, the net longwave radiation contribution to the melting is small. The downward longwave radiation increases because of the T$^4$ relationship with air temperature, whereas the upward longwave radiation increases because the glacier surface is warming. The latent heat flux from refreezing of melt water and the sensible heat from surface warming are also small components of the net radiation balance."*

**Detailed comments P6 L26-27: remove "this is because..."**

*Deleted "this is because..."*

**P9 L6: add Marzeion et al., 2012 to the references list.**

*Reference added*

**P10 L23: please add reasons for the negative bias. In linear a model with enough degrees of freedom, minimizing RMSD will always minimize the bias too. So the first thing that comes to mind is stat systematic problems in the model and/or the forcing data are preventing this bias minimization**

**(confirmed by the supplementary analyses). In short: there seems to be a structural problem in either the model or the forcing data.**

*We agree that minimising the RMSE should also minimise the bias unless there is some structural problem preventing this. Perhaps increasing the upper bound for the tuneable precipitation gradient might help this, since we found that minimising the bias resulted in a preference for higher precipitation gradients. Our upper bound is 25%/100m but the work of Rye (2012) found optimum values as high as 45% to 51%/100m for glaciers in Svalbard. If this work were to be repeated, we would recommend improving the way the precipitation lapse rate is implemented in the model.*

*The explanation for the negative biases is included in Section 3.2 so instead of repeating the material we add*

*Changes to manuscript:*

*" The negative bias is also seen in the summer and winter mass balance and discussed in Section 3.2"*

*We also updated section 3.2 to answer a comment by Reviewer #1 as to why the model predicts some negative winter mass balance and some positive summer mass balance. Please see our reply to this below.*

**P10 L29: "Our mass balance model does include sublimation". I am curious: since you have a latent heat flux, why don't you simply convert it to a mass loss? This is the typical way to compute sublimation in glacier energy and mass balance models.**

*Perhaps there is a mis-communication here. We write that our model 'does include' sublimation. Sublimation is calculated by the surface exchange module in JULES. To avoid confusion, we change this to 'our model **includes** sublimation'*

**Figure 4: you might consider add Maladeta to Figure 3 and spare a figure.**

*Changes to manuscript: We deleted Figure 4 and included it in Figure 3 (see black circles)*

**Table 6: consider making a bar-plot out of it for more readability**

*We did not make a bar-chart of Table 6 because we included an extra figure in the supplementary material to show the future cumulated mass balances at elevation levels for each RGI6 region (Fig. S8). The volume change per height in Fig S8 is more meaningful than the percentage changes listed in columns 5-7 of Table 6. For example, in Alaska for elevation ranges 4250m-8000m there is 408 % volume increase, but this does not translate to a large volume increase because there is very little glaciated area at these elevation bands.*

Changes to manuscript: None

**P13 L30: here you talk about sublimation. This contradicts your statement above.**

*Please see our reply above confirming that sublimation **is** included in the model.*

**Fig. 12: to make the figure more readable you could remove the x and y axis labels for the interior plots, since they are the same for each plot.**

*Changes to manuscript: The x and y axis labels for the interior plots are removed.*

**Figure 13 and corresponding analysis in the text: I have trouble to understand why the upper elevations see a reduction in melt while the lower parts do not? The provided explanation**

**("reduction in mass loss as glaciers disappear towards the end of the century" holds even more true for lower elevations.**

**Or is this due to regional differences, the high latitude arctic having more mass below 2000 m a.s.l? This needs more explanation in the text.**

*Yes, you are correct. The reason there is no reduction in melt rates below 2000m is because there are still large amounts of ice available to melt at high latitudes.*

*Changes to manuscript: We added the text below and Figure S8 to the supplementary material to show the future cumulated mass balances as a function of height.*

*"Figure 12 shows how the global annual mass balance components vary with time for low, medium and high elevations ranges. There is a reduction in accumulation and refreezing at all elevation ranges towards the end of the century. Melt rates decreases at medium and high elevation ranges because glaciers mass is lost at these altitudes, therefore less ice is available to melt (see Fig. S8 for the future cumulated mass balances as a function of height). Melt rates are constant at the low elevation ranges because there remains substantial quantities of ice available to melt at the end of the century in Greenland, Arctic Canada North and South, Svalbard, Russian Arctic.*

**P16-L16: about elevation feedback - see main comment above.**

*We removed this text. Please see our reply to this above.*

**P18 L20: "Changes in solar radiation can be an important driver of melting.": It is a bit sad that you didn't take my advice about analysing the energy fluxes...**

*Please see our reply to this above and the following changes to the manuscript:*

*We added the following to the discussion: "In this paper, we present a brief analysis of the future global energy balance fluxes, but how the fluxes vary for individual regions and elevation levels could be investigated further. "*

**I would like to thank the authors for the thoughtful, and in most cases, satisfactory responses to my review comments. In particular, I want to compliment them on performing a substantial amount of additional runs, including alternative optimization approaches. Those additional analyses and results make the manuscript a lot stronger and help to better understand and evaluate the results. I hope the authors would agree that there are still a number of issues with the model that will eventually need to be addressed, but given that this is a very new approach to modeling glaciers on the global scale (and one has to start somewhere), I think the comprehensive presentation of the validation results, and the discussion of the (in some cases problematic) model behaviour warrants a publication in The Cryosphere.**

**There are just two minor comments that should be addressed prior to publication, and I have one suggestion that the author may consider for the further development of the model:**

**- My comment on P10 L1f does not seem to have been addressed.**

*Your original comment was "The tropical glaciers are really small; there are probably numerous more likely explanations for a warm bias than glaciers lacking in the model.*

*Change to manuscript: We removed the text "It is possible that the ECMWF model does not include glacier ice in tropical regions. The absence of ice to cool the lower atmosphere would make the grid box mean temperature too warm".*

**- With my comment on Fig 6 and 7 (negative winter MBs and positive summer MBs) I did not want to imply that from a warm bias in winter follows a cold bias in summer (sorry for the confusion), but I wanted to suggest that it should be discussed why there are so many negative winter MB values (as well as quite a few positive summer MB values). Typically, winter and summer values are well separated by the zero line (see, e.g., Fig. 3 in Radic et al., 2014).**

*The negative mass balance in winter and positive mass balance in summer at some sites is caused by the simple treatment of the precipitation lapse rate in the model.*

*We added the following text by way of explanation to the model validation section 3.2.  An extra two figures have been added to the Supplementary material to show the mass balance components for two glaciers, one which negative winter mass balance and another which has positive summer mass balance (Fig. S9 and S10).*

*Changes to manuscript:*

*The reason for the negative bias is because the model underestimates the precipitation and therefore the accumulation part of the mass balance is underestimated.   This is because our approach to correcting the coarse scale gridded precipitation for orographic effects is simple. We use a single precipitation gradient for each RGI6 region and do not apply a bias correction. A bias correction is often recommended because precipitation is underestimated in coarse resolution datasets. Gauging observations are sparse in high mountains regions and snowfall observations can be susceptible to undercatch by 20–50% (Rasmussen et al. 2012).  Our precipitation rates are generally too low because we do not bias correct the precipitation.*

*Other studies use a bias correction that varies regionally (Radic and Hock 2011, Radic et al. 2014, Bliss et al. 2014).  In those studies, the precipitation at the top of the glacier was estimated using a bias correction factor $k_p$. The decrease in precipitation from the top of the glacier to the snout was calculated using a precipitation gradient. To account for the fact that the mass balance of maritime and continental glaciers respond differently to precipitation changes $k_p$ was related to a continentality index. Our motivation for using a single precipitation gradient for each RGI6 region, and no bias correction was to test the simplest approach first, however the resulting biases suggest that this approach could be improved.*

*The impact of underestimating the precipitation is that we simulate negative mass balance in winter at some observational sites (Fig 5(A) and Fig 4.). To demonstrate this, we compare the mass balance components for two glaciers; the Leviy Aktru in the Russian Altai Mountains which has negative mass balance in the winter and Kozelskiy glacier in North Eastern Russia which has no negative mass balance in the winter (See Fig. S9).  Both glaciers are in the North Asia RGI6 region, so have the same tuned parameters for mass balance. The simulated winter accumulation rates are much lower at Leviy Aktru glacier than Kozelskiy glacier leading to negative mass balance at the lowest 3 model levels below 2750m.*

*The simplistic treatment of the precipitation lapse rate also leads to instances where the model simulates positive mass balance in the summer at some locations (Fig 6 (A) and Fig 4.). We show the summer mass balance components for the same two glaciers in Fig S10. Positive mass balance is simulated at Kozelskiy glacier because accumulation exceeds the melting. This suggests that the precipitation gradient (19% per 100m for North Asia) is overly steep in the summer at this location.*

**Finally, just an idea to consider, concerning the issue of water mass conservation being affected by the precipitation lapse rate (page 6, line 3): The authors correctly point out that a precipitation lapse rate is standard in glacier modeling (effectively increasing precipitation to the glacier surface compared to the data set used as boundary condition), and it makes sense because wind-blown snow and avalanching are important components of the mass balance of many glaciers – i.e., there is in fact horizontal redistribution of precipitation to the glaciers. One way to conserve mass would be to reduce the precipitation onto the non-glaciated area of the grid cell, which would be conceptually consistent with horizontal mass movement within the grid box.**

*Thank you for this suggestion. We added this to the discussion*

*Changes to manuscript:*

*Added this text P17 L24*

*"Another limitation of the model, which may be problematic for same applications, is that the gridbox mean precipitation is not conserved when precipitation is adjusted for elevation. This correction was necessary to get enough accumulation in the mass balance at high elevations. One way to conserve water mass would be to reduce the precipitation onto the non-glaciated area of the grid cell. This would represent horizontal mass movement within the grid box from wind-blown snow and avalanching."*

Kotlarski, S., D. Jacob, R. Podzun & F. Paul (2010) Representing glaciers in a regional climate model. *Climate Dynamics,* 34**,** 27-46.

Marzeion, B., A. H. Jarosch & J. M. Gregory (2014) Feedbacks and mechanisms affecting the global sensitivity of glaciers to climate change. *Cryosphere,* 8**,** 59-71.

Radic, V., A. Bliss, A. C. Beedlow, R. Hock, E. Miles & J. G. Cogley (2014) Regional and global projections of twenty-first century glacier mass changes in response to climate scenarios from global climate models. *Climate Dynamics,* 42**,** 37-58.

Rye, C. J., I. C. Willis, N. S. Arnold & J. Kohler (2012) On the need for automated multiobjective optimization and uncertainty estimation of glacier mass balance models. *Journal of Geophysical Research-Earth Surface,* 117.

[revised manuscript text omitted]
 100km2,300km2 and 500km2 are applied to the validation. The model predicts negative mass balance in winter and positive mass balance in summer at several sites (also seen in Fig. 4).

The negative mass balance in winter happens because the accumulation is underestimated and the positive mass balance in summer is when accumulation is overestimated.

To explore why we get negative winter mass balances we compare the mass balance components for two glaciers; the Leviy Aktru in the Russian Altai Mountains which has some melting in the winter and Kozelskiy glacier in North Eastern Russia which has no melting in the winter (See Fig. S9).

Both glaciers are in the North Asia RGI6 region, so have the same tuned parameters for mass balance; precipitation and temperature lapse rates, wind speed scaling and visible and near-infra albedos for snow and ice. The simulated accumulation rates are much lower at Leviy Aktru glacier than Kozelskiy glacier leading to negative mass balance at the lowest 3 model levels.

Our approach to the precipitation correction for height is very simple in contrast to other studies (Radic and Hock 2011, Radic et al. 2014, Bliss, Hock and Radic 2014). In those studies, the precipitation at the top of the glacier was estimated using a bias correction factor kp. The decrease in precipitation from the top of the glacier to the snout was calculated using a precipitation gradient. To account the fact the mass balance of maritime and continental glaciers respond differently to precipitation changes kp was related to a continentality index. This regional variation in the precipitation lapse as a function of climate is not included in our model. Neither do we include a bias correction factor meaning hat our acculination artes are too low.

[revised manuscript text omitted]

*Figure S8 Cumulative mass balances as a function of height for the middle (2020-2040) and end of the century (2080-2087). The projections are the mean of the HadGEM3-A ensemble.*

[Figure]

*Figure S9 Simulated winter mass balance, snow melt and accumulation for the year 1989 when the model is forced with WFDEI data. The black stars are the elevation-dependant specific mass balance observations.*

[Figure]

*Figure S10 Simulated summer mass balance, snow melt and accumulation for the year 1992 when the model is forced with WFDEI data. The black stars are the elevation-dependant specific mass balance observations.*

*Table S1 Equally plausible parameter sets derived from the root mean square error for each RGI6 region. Best parameter sets are shown in bold.*

[revised manuscript text omitted]

*Table S1 Equally plausible parameter sets for each RGI6 region.*

*Table S2 Optimum regional parameter sets when we maximise the correlation coefficient between the simulated and observed specific mass balance.*

| Region | $\alpha_{vis,snow}$ | $\alpha_{nir,snow}$ | $\alpha_{vis,ice}$ | $\alpha_{nir,ice}$ | $\gamma_{temp}$ ($^{\circ}$K km$^{-1}$) | $\gamma_{precip}$ (%/100m) | $\gamma_{wind}$ |
|---|---|---|---|---|---|---|---|
| Alaska | 0.96 | 0.74 | 0.67 | 0.30 | 9.20 | 25.00 | 1.68 |
| Western Canada and US | 0.85 | 0.61 | 0.36 | 0.17 | 9.60 | 12.00 | 3.71 |
| Arctic Canada North | 0.99 | 0.74 | 0.64 | 0.24 | 7.40 | 22.00 | 1.46 |
| Arctic Canada South | 0.97 | 0.68 | 0.65 | 0.52 | 9.40 | 23.00 | 3.48 |
| Greenland | 0.99 | 0.71 | 0.32 | 0.15 | 9.20 | 20.00 | 2.12 |
| Iceland | 0.95 | 0.69 | 0.48 | 0.25 | 8.67 | 19.31 | 2.79 |
| Svalbard | 0.98 | 0.76 | 0.58 | 0.34 | 9.00 | 9.00 | 3.69 |
| Scandinavia | 0.96 | 0.74 | 0.67 | 0.30 | 9.20 | 25.00 | 1.68 |
| Russian Arctic | 0.95 | 0.69 | 0.48 | 0.25 | 8.67 | 19.31 | 2.79 |
| North Asia | 0.98 | 0.66 | 0.25 | 0.18 | 9.60 | 21.00 | 3.12 |
| Central Europe | 0.94 | 0.61 | 0.23 | 0.16 | 9.10 | 22.00 | 1.49 |
| Caucasus and Middle East | 0.98 | 0.66 | 0.25 | 0.18 | 9.60 | 21.00 | 3.12 |
| Central Asia | 0.98 | 0.66 | 0.25 | 0.18 | 9.60 | 21.00 | 3.12 |
| South Asia West | 0.97 | 0.75 | 0.44 | 0.12 | 7.70 | 17.00 | 3.77 |
| South Asia East | 0.86 | 0.61 | 0.57 | 0.28 | 8.70 | 22.00 | 3.95 |
| Low Latitudes | 0.97 | 0.68 | 0.65 | 0.52 | 9.40 | 23.00 | 3.48 |
| Southern Andes | 0.86 | 0.69 | 0.68 | 0.32 | 4.30 | 9.00 | 1.02 |
| New Zealand | 0.97 | 0.75 | 0.44 | 0.12 | 7.70 | 17.00 | 3.77 |
| Mean | 0.95 | 0.69 | 0.48 | 0.25 | 8.67 | 19.31 | 2.79 |

Table S3 Optimum regional parameter sets when we minimise the bias between the simulated and observed specific mass balance.

| Region | $\alpha_{vis,snow}$ | $\alpha_{nir,snow}$ | $\alpha_{vis,ice}$ | $\alpha_{nir,ice}$ | $\gamma_{temp}$ (°K km$^{-1}$) | $\gamma_{precip}$ (%/100m) | $\gamma_{wind}$ |
|---|---|---|---|---|---|---|---|
| Alaska | 0.88 | 0.63 | 0.42 | 0.24 | 8.20 | 19.00 | 1.21 |
| Western Canada + US | 0.87 | 0.72 | 0.56 | 0.33 | 6.00 | 10.00 | 2.25 |
| Arctic Canada North | 0.83 | 0.66 | 0.20 | 0.14 | 5.30 | 7.00 | 3.21 |
| Arctic Canada South | 0.82 | 0.66 | 0.16 | 0.12 | 9.30 | 21.00 | 1.74 |
| Greenland | 0.84 | 0.67 | 0.64 | 0.29 | 5.80 | 17.00 | 1.42 |
| Iceland | 0.90 | 0.71 | 0.52 | 0.27 | 7.53 | 17.38 | 1.84 |
| Svalbard | 0.95 | 0.76 | 0.54 | 0.35 | 9.00 | 14.00 | 1.02 |
| Scandinavia | 0.92 | 0.68 | 0.52 | 0.25 | 5.40 | 20.00 | 1.48 |
| Russian Arctic | 0.90 | 0.71 | 0.52 | 0.27 | 7.53 | 17.38 | 1.84 |
| North Asia | 0.94 | 0.74 | 0.69 | 0.50 | 8.10 | 19.00 | 1.40 |
| Central Europe | 0.76 | 0.51 | 0.48 | 0.19 | 4.90 | 24.00 | 3.73 |
| Caucasus and Middle East | 0.90 | 0.75 | 0.41 | 0.11 | 6.90 | 5.00 | 3.22 |
| Central Asia | 0.96 | 0.74 | 0.67 | 0.30 | 9.20 | 25.00 | 1.68 |
| South Asia West | 0.95 | 0.76 | 0.54 | 0.35 | 9.00 | 14.00 | 1.02 |
| South Asia East | 0.94 | 0.78 | 0.60 | 0.23 | 5.90 | 19.00 | 1.76 |
| Low Latitudes | 0.96 | 0.74 | 0.67 | 0.30 | 9.20 | 25.00 | 1.68 |
| Southern Andes | 0.95 | 0.76 | 0.54 | 0.35 | 9.00 | 14.00 | 1.02 |
| New Zealand | 0.96 | 0.74 | 0.67 | 0.30 | 9.20 | 25.00 | 1.68 |
| Mean | 0.90 | 0.71 | 0.52 | 0.27 | 7.53 | 17.38 | 1.84 |

*Table S4 Comparison of percentage volume change relative to initial volume, from this study with Huss and Hock (2015)*

| | This study ΔV % 2097-2011 | Huss & Hock (2015) ΔV % 2100-2010 | This study minus Huss & Hock (2015) |
|---|---|---|---|
| Alaska | -89±2 | -58±14 | -30 |
| Western Canada and US | -100±0 | -95±5 | -5 |
| Arctic Canada North | -47±3 | -30±12 | -9 |
| Arctic Canada South | -74±8 | -52±14 | -22 |
| Greenland | -31±5 | -52±13 | 20 |
| Iceland | -98±3 | -62±18 | -36 |
| Svalbard | -68±16 | -82±18 | 14 |
| Scandinavia | -98±3 | -96±4 | -2 |
| Russian Arctic | -79±10 | -70±19 | -7 |
| North Asia | -71±5 | -81±7 | 10 |
| Central Europe | -99±0 | -98±2 | -1 |
| Caucasus and Middle East | -100±0 | -96±3 | -4 |
| Central Asia | -80±7 | -88±7 | 8 |
| South Asia West | -98±1 | -87±9 | -11 |
| South Asia East | -95±2 | -92±5 | -3 |
| Low Latitudes | -100±0 | -98±0 | -2 |
| Southern Andes | -98±1 | -44±14 | -54 |

| New Zealand | -88±5 | -82±8 | 4 |
|---|---|---|---|

*Table S2 Comparison of percentage volume change relative to initial volume, from this study with Huss and Hock (2015)*

| | This study SLE mm 2097-2011 | Huss & Hock (2015) SLE mm 2100-2010 | Radic et al.,(2014) SLE mm 2100-2006 | This study minus Huss & Hock (2015 | This study minus Radic et al.,(2014) |
|---|---|---|---|---|---|
| Alaska | 44.6±1.1 | 24.9±6.3 | 25.4 | 23.1 | 22.6 |
| Western Canada and US | 2.8±0.0 | 2.2±0.1 | 2.6 | 0.7 | 0.3 |
| Arctic Canada North | 35.8±3.0 | 19.7±7.8 | 42.2 | 15.1 | -7.4 |
| Arctic Canada South | 18.1±2.1 | 9.9±2.8 | 15.0 | 10.1 | 5.0 |
| Greenland | 20.1±4.4 | 17.7±4.6 | 20.4 | 9.1 | 6.4 |
| Iceland | 9.3±0.3 | 4.7±1.7 | 4.9 | 5.4 | 5.2 |
| Svalbard | 17.0±4.6 | 13.9±3.1 | 15.8 | 5.0 | 3.1 |
| Scandinavia | 0.6±0.0 | 0.3±0.0 | 0.5 | 0.4 | 0.2 |
| Russian Arctic | 33.3±4.8 | 18.1±5.5 | 28.3 | 18.0 | 7.8 |
| North Asia | 0.3±0.0 | 0.2±0.0 | 0.6 | 0.1 | -0.3 |
| Central Europe | 0.3±0.0 | 0.3±0.0 | 0.3 | 0.0 | 0.0 |
| Caucasus and Middle East | 0.2±0.0 | 0.1±0.0 | 0.2 | 0.1 | 0.0 |
| Central Asia | 8.0±0.7 | 9.2±1.1 | 11.9 | -0.7 | -3.4 |
| South Asia West | 8.1±0.1 | 6.2±1.0 | 7.1 | 2.5 | 1.6 |
| South Asia East | 1.9±0.0 | 2.4±0.7 | 3.5 | -0.4 | -1.4 |
| Low Latitudes | 0.2±0.0 | 0.2±0.0 | 0.5 | 0.0 | -0.3 |
| Southern Andes | 14.4±0.1 | 5.8±1.8 | 8.5 | 9.7 | 7.0 |
| New Zealand | 0.1±0.0 | 0.1±0.0 | 0.1 | 0.0 | 0.0 |
| Global | 215.2±21.3 | 135.9±13.0 | 187.9 | 98.3 | 46.4 |

*Table S5 Comparison of volume losses for this study with two other studies (Huss and Hock 2015, Radic et al. 2014). Volume loss is expressed in terms of sea level equivalent (mm). The last column lists volume losses when we correct the bias in the calibrated mass balance. There were no observations available to calibrate the present-day mass balance for Iceland and the Russian Arctic, so there is no bias corrected volume loss for these regions.*

| | This study 2097-2011 | Huss 2100-2010 | Radic 2100-2006 | This study minus Hock | This study minus Radic | Bias corrected |
|---|---|---|---|---|---|---|
| Alaska | 44.6±1.1 | 24.9±6.3 | 25.4 | 23.1 | 22.6 | 45.0±1.1 |
| Western Canada and US | 2.8±0.0 | 2.2±0.1 | 2.6 | 0.7 | 0.3 | 2.2±0.0 |
| Arctic Canada North | 35.8±3.0 | 19.7±7.8 | 42.2 | 15.1 | -7.4 | 37.8±2.9 |
| Arctic Canada South | 18.1±2.1 | 9.9±2.8 | 15.0 | 10.1 | 5.0 | 18.3±2.2 |
| Greenland | 20.1±4.4 | 17.7±4.6 | 20.4 | 9.1 | 6.4 | 21.9±4.3 |
| Iceland | 9.3±0.3 | 4.7±1.7 | 4.9 | 5.4 | 5.2 | - |

| | | | | | |
|---|---|---|---|---|---|
| Svalbard | 17.0±4.6 | 13.9±3.1 | 15.8 | 5.0 | 3.1 | 15.6±4.6 |
| Scandinavia | 0.6±0.0 | 0.3±0.0 | 0.5 | 0.4 | 0.2 | 0.8±0.0 |
| Russian Arctic | 33.3±4.8 | 18.1±5.5 | 28.3 | 18.0 | 7.8 | - |
| North Asia | 0.3±0.0 | 0.2±0.0 | 0.6 | 0.1 | -0.3 | 0.2±0.0 |
| Central Europe | 0.3±0.0 | 0.3±0.0 | 0.3 | 0.0 | 0.0 | 0.4±0.0 |
| Caucasus and Middle East | 0.2±0.0 | 0.1±0.0 | 0.2 | 0.1 | 0.0 | 0.2±0.0 |
| Central Asia | 8.0±0.7 | 9.2±1.1 | 11.9 | -0.7 | -3.4 | 6.0±0.8 |
| South Asia West | 8.1±0.1 | 6.2±1.0 | 7.1 | 2.5 | 1.6 | 7.9±0.1 |
| South Asia East | 1.9±0.0 | 2.4±0.7 | 3.5 | -0.4 | -1.4 | 1.7±0.0 |
| Low Latitudes | 0.2±0.0 | 0.2±0.0 | 0.5 | 0.0 | -0.3 | 0.2±0.0 |
| Southern Andes | 14.4±0.1 | 5.8±1.8 | 8.5 | 9.7 | 7.0 | 7.6±0.3 |
| New Zealand | 0.1±0.0 | 0.1±0.0 | 0.1 | 0.0 | 0.0 | 0.01±0.0 |
| Global | 215.2±21.3 | 135.9±13.0 | 187.9 | 98.3 | 46.4 | 222.5±20.1 |

*Table S3 Comparison of volume losses for this study with (Huss and Hock 2015, Radic et al. 2014). Volume loss is expressed in terms of sea level equivalent (mm)*

Table S6 Comparison of modelled and observed energy balance components at five elevation levels on the Pasterze glacier, Austria. Information about the observations are detailed in Greuell and Smeets (2001). The table contains data from two experiments, the first where wind speed in not tuned ($\gamma_{wind}$ = 1) and the second where wind speed is increased to four times the surface wind speed ($\gamma_{wind}$ = 4).

| Elevation | | Observations | $\gamma_{wind}$ = 1 | $\gamma_{wind}$ = 4 |
|---|---|---|---|---|
| 2205m | Incoming short-wave radiation (Wm$^{-2}$) | 256.0 | 262.9 | 262.9 |
| | Albedo (visible) | 0.2 | 0.1 | 0.2 |
| | Incoming long-wave radiation (Wm$^{-2}$) | 299.0 | 295.7 | 295.7 |
| | Outgoing long-wave radiation (Wm$^{-2}$) | 315.0 | 305.8 | 309.5 |
| | Elevated air temperature (°C) | 6.8 | 9.1 | 9.1 |
| | Surface temperature (°C) | -0.1 | -1.5 | -0.7 |
| | Roughness length (mm) | 2.6 | 3.0 | 3.0 |
| | **Sensible heat flux (Wm$^{-2}$)** | **48.0** | **0.2** | **22.2** |
| | Latent heat flux (Wm$^{-2}$) | 10.0 | 0.2 | 13.0 |
| | Net radiation on tiles (Wm$^{-2}$) | 282.2 | 223.7 | 219.1 |
| | **Wind speed (ms$^{-1}$)** | **4.1** | **1.1** | **4.3** |
| | Ablation rate (mm water equivalent day$^{-1}$) | 61.3 | 52.7 | 58.4 |
| 2310m | Incoming short-wave radiation (Wm$^{-2}$) | 272.0 | 262.9 | 262.9 |
| | Albedo (visible) | 0.3 | 0.1 | 0.2 |
| | Incoming long-wave radiation (Wm$^{-2}$) | 299.0 | 295.7 | 295.7 |
| | Outgoing long-wave radiation (Wm$^{-2}$) | 315.0 | 305.8 | 309.6 |
| | Elevated air temperature (°C) | 6.4 | 8.4 | 8.4 |
| | Surface temperature (°C) | -0.1 | -1.5 | -0.6 |
| | Roughness length (mm) | 1.2 | 3.0 | 3.0 |
| | **Sensible heat flux (Wm$^{-2}$)** | **53.0** | **0.2** | **21.5** |
| | Latent heat flux (Wm$^{-2}$) | 11.0 | 0.2 | 12.7 |
| | Net radiation on tiles (Wm$^{-2}$) | 283.1 | 223.4 | 218.5 |
| | **Wind speed (ms$^{-1}$)** | **4.5** | **1.1** | **4.3** |
| | Ablation rate (mm water equivalent day$^{-1}$) | 63.1 | 52.6 | 57.9 |
| 2420m | Incoming short-wave radiation (Wm$^{-2}$) | 278.0 | 262.9 | 262.9 |
| | Albedo (visible) | 0.3 | 0.2 | 0.2 |
| | Incoming long-wave radiation (Wm$^{-2}$) | 296.0 | 295.7 | 295.7 |

| | | | | |
|---|---|---|---|---|
| | Outgoing long-wave radiation (Wm$^{-2}$) | 315.0 | 305.8 | 309.7 |
| | Elevated air temperature (°C) | 7.1 | 7.6 | 7.6 |
| | Surface temperature (°C) | -0.1 | -1.5 | -0.6 |
| | Roughness length (mm) | 5.8 | 3.0 | 3.0 |
| | **Sensible heat flux (Wm$^{-2}$)** | **63.0** | **0.1** | **20.7** |
| | Latent heat flux (Wm$^{-2}$) | 10.0 | 0.1 | 12.1 |
| | Net radiation on tiles (Wm$^{-2}$) | 315.5 | 218.2 | 212.7 |
| | **Wind speed (ms$^{-1}$)** | **4.6** | **1.1** | **4.3** |
| | Ablation rate (mm water equivalent day$^{-1}$) | 66.9 | 51.3 | 56.3 |
| 2945m | Incoming short-wave radiation (Wm$^{-2}$) | 307.0 | 262.9 | 262.9 |
| | Albedo (visible) | 0.6 | 0.4 | 0.4 |
| | Incoming long-wave radiation (Wm$^{-2}$) | 282.0 | 295.7 | 295.7 |
| | Outgoing long-wave radiation (Wm$^{-2}$) | 314.0 | 307.7 | 311.0 |
| | Elevated air temperature (°C) | 3.5 | 4.2 | 4.2 |
| | Surface temperature (°C) | -0.3 | -1.1 | -0.3 |
| | Roughness length (mm) | 1.3 | 3.0 | 3.0 |
| | **Sensible heat flux (Wm$^{-2}$)** | **23.0** | **1.3** | **15.9** |
| | Latent heat flux (Wm$^{-2}$) | 5.0 | 1.2 | 9.1 |
| | Net radiation on tiles (Wm$^{-2}$) | 139.9 | 183.4 | 178.3 |
| | **Wind speed (ms$^{-1}$)** | **4.3** | **1.1** | **4.3** |
| | Ablation rate (mm water equivalent day$^{-1}$) | 32.1 | 46.0 | 50.7 |
| 3225m | Incoming short-wave radiation (Wm$^{-2}$) | 286.0 | 262.9 | 262.9 |
| | Albedo (visible) | 0.6 | 0.5 | 0.5 |
| | Incoming long-wave radiation (Wm$^{-2}$) | 274.0 | 295.7 | 295.7 |
| | Outgoing long-wave radiation (Wm$^{-2}$) | 313.0 | 308.0 | 310.7 |
| | Elevated air temperature (°C) | 3.2 | 2.6 | 2.6 |
| | Surface temperature (°C) | -0.7 | -1.0 | -0.4 |
| | Roughness length (mm) | 2.0 | 3.0 | 3.0 |
| | **Sensible heat flux (Wm$^{-2}$)** | **20.0** | **2.4** | **11.5** |
| | Latent heat flux (Wm$^{-2}$) | 1.0 | 2.2 | 6.9 |
| | Net radiation on tiles (Wm$^{-2}$) | 115.4 | 169.8 | 165.4 |
| | **Wind speed (ms$^{-1}$)** | **4.4** | **1.1** | **4.3** |
| | Ablation rate (mm water equivalent day$^{-1}$) | 24.0 | 43.5 | 47.3 |

Greuell, W. & P. Smeets (2001) Variations with elevation in the surface energy balance on the Pasterze (Austria). *Journal of Geophysical Research-Atmospheres,* 106, 31717-31727.

Huss, M. & R. Hock (2015) A new model for global glacier change and sea-level rise. *Frontiers in Earth Science,* 3.

Radic, V., A. Bliss, A. C. Beedlow, R. Hock, E. Miles & J. G. Cogley (2014) Regional and global projections of twenty-first century glacier mass changes in response to climate scenarios from global climate models. *Climate Dynamics,* 42**,** 37-58.